# ZO-1 interacts with YB-1 in endothelial cells to regulate stress granule formation during angiogenesis

Yassine El Bakkouri [1,11], Rony Chidiac [1,2,11], Chantal Delisle[1], Jeanne Corriveau[1], Gael Cagnone [3], Vanda Gaonac'h-Lovejoy[1], Ashley Chin[4], Éric Lécuyer[4,5], Stephane Angers [2], Jean-Sébastien Joyal[3,6], Ivan Topisirovic [7], Laura Hulea [5,8], Alexandre Dubrac [3,9] & Jean-Philippe Gratton [1,10] ✉

Zonula occludens-1 (ZO-1) is involved in the regulation of cell-cell junctions between endothelial cells (ECs). Here we identify the ZO-1 protein interactome and uncover ZO-1 interactions with RNA-binding proteins that are part of stress granules (SGs). Downregulation of ZO-1 increased SG formation in response to stress and protected ECs from cellular insults. The ZO-1 interactome uncovered an association between ZO-1 and Y-box binding protein 1 (YB-1), a constituent of SGs. Arsenite treatment of ECs decreased the interaction between ZO-1 and YB-1, and drove SG assembly. YB-1 expression is essential for SG formation and for the cytoprotective effects induced by ZO-1 downregulation. In the developing retinal vascular plexus of newborn mice, ECs at the front of growing vessels express less ZO-1 but display more YB-1-positive granules than ECs located in the vascular plexus. Endothelial-specific deletion of ZO-1 in mice at post-natal day 7 markedly increased the presence of YB-1-positive granules in ECs of retinal blood vessels, altered tip EC morphology and vascular patterning, resulting in aberrant endothelial proliferation, and arrest in the expansion of the retinal vasculature. Our findings suggest that, through its interaction with YB-1, ZO-1 controls SG formation and the response of ECs to stress during angiogenesis.

Cell-cell junctions between endothelial cells (ECs) are constantly remodeled during the formation of new blood vessels[1,2]. Intercellular junctions between ECs are generally formed by homotypic interactions between transmembrane proteins connected to a network of intracellular cytosolic proteins that integrate signaling events essential for angiogenesis[3]. Genetic deletion of the cell junction proteins VE-cadherin, β-catenin, or zonula occludens-1 (ZO-1) in murine ECs leads to embryonic lethality as a result of vascular defects[4–6]. In particular,

[1]Department of Pharmacology and Physiology, Faculty of Medicine, Université de Montréal, Montreal, Quebec, Canada. [2]Donnelly Centre, University of Toronto, Toronto, ON, Canada. [3]Centre Hospitalier Universitaire Sainte-Justine Research Center, Montreal, QC, Canada. [4]Institut de recherches cliniques de Montréal (IRCM), Montreal, QC, Canada. [5]Department of Biochemistry and Molecular Medicine, Université de Montréal, Montreal, QC, Canada. [6]Department of Pediatrics, Faculty of Medicine, Université de Montréal, Montreal, QC, Canada. [7]Lady Davis Institute for Medical Research, Sir Mortimer B. Davis-Jewish General Hospital, Montreal, Quebec, Canada and Gerald Bronfman Department of Oncology, Faculty of Medicine, McGill University, Montreal, QC, Canada. [8]Maisonneuve-Rosemont Hospital Research Centre, Montréal, Quebec, Canada and Department of Medicine, Faculty of Medicine, Université de Montréal, Montreal, QC, Canada. [9]Department of Pathology and Cell Biology, Faculty of Medicine, Université de Montréal, Montreal, QC, Canada. [10]Centre d'Innovation Biomédicale (CIB), Faculty of Medicine, Université de Montréal, Montreal, QC, Canada. [11]These authors contributed equally: Yassine El Bakkouri, Rony Chidiac. ✉e-mail: jean-philippe.gratton@umontreal.ca

ZO-1 deficiency in mice leads to defective angiogenesis in the yolk sac, although EC differentiation appears to be normal[6]. ZO-1 functions as a major cytoskeletal organizer in ECs as it orchestrates adherens junctions to control barrier function, cell migration, and angiogenesis[7,8]. It has been previously reported that vascular endothelial growth factor (VEGF) induces phosphorylation of both ZO-1 and the plasma membrane protein occludin, resulting in the disruption of the localization of ZO-1 at cell junctions and increased permeability of EC monolayers[9,10]. Furthermore, we previously identified ZO-1 as being at the center of a signaling nexus essential for the VEGF-induced proliferation of ECs during angiogenesis[11]. Angiogenesis requires the coordination of differentiation, proliferation, polarization, migration, junction rearrangements, and morphological changes of ECs to ensure the growth and integrity of the vascular network[12,13]. In the context of angiogenic sprouting, as observed during vascular development of the mouse retina, ECs differentiate into tip and stalk cells, a process regulated by Notch-Dll4-VEGF pathways[14,15]. The highly motile tip cells extend filopodia and branch out in a directional manner from existing vessels, whereas the stalk cells remain behind, maintaining cell-cell contacts during sprouting to preserve vessel integrity and establish lumens for blood perfusion[16]. During this collective cell migration process, tip cells at the front have to adopt a polarized morphology while cell-cell junctions at the back of endothelial sprouts should be maintained[17].

Zonula occludens (ZO) proteins (ZO-1, ZO-2, and ZO-3) are part of the membrane-associated guanylate kinase (MAGUK) family of proteins and comprise three PDZ domains, a SH3 domain, and a guanylate kinase (GUK) domain[18]. They interact with each other through the binding of their corresponding PDZ-2 domains[19,20]. ZO-1 also interacts with several membrane proteins, including claudins and JAM-A via the PDZ-1 and PDZ-2 domains, respectively, and occludin through the GUK domain[21–23]. Moreover, ZO-1 interacts with cytoplasmic proteins such as the actin-binding proteins cortactin, α-catenin, and the Ras target AF6/afadin as well as the actin- and myosin-binding proteins cingulin and Shroom[8,18,24]. In addition to its roles at cell junctions, ZO-1 is involved in transcriptional modulation through its association with the Y-box binding protein 3 (YB-3; aka ZO-1-associated nucleic acid-binding protein, ZONAB) to influence cell proliferation[25–27]. Once released from ZO-1, YB-3 can accumulate in the nucleus, where it associates with cell division kinase (CDK) 4 and promotes cell proliferation[26].

ECs are exposed to numerous stresses emanating from their environment; these include reactive oxygen species (ROS), oxidized low-density lipoprotein (LDL), hypoxic conditions, and turbulent blood flow. The disorganized vasculature in solid tumors leads to hypoxia, which in turn can increase cellular ROS and cause endoplasmic reticulum (ER) stress[28–30]. One of the hallmarks of cellular adaptation to stress is the suppression of global protein synthesis[31]. Stress granules (SGs) are cytoplasmic membraneless ribonucleoprotein (RNP) conglomerates composed of the 40 S small ribosomal subunit, translation initiation factors, various RNA-binding proteins (RBPs), and translationally stalled mRNAs. Emerging evidence suggests that SGs are dynamic structures that form in response to acute or chronic stress. Their composition undergoes rapid remodeling, influencing cell fate during periods of stress[32]. The stress-induced phosphorylation of the eukaryotic translation initiation factor 2α (eIF2α) attenuates global mRNA translation while promoting both acute and chronic SG assembly[32,33]. SGs are generally thought to function as sites of mRNA triage, wherein individual mRNAs are dynamically sorted for storage during stress[34]. Furthermore, recent studies suggest that SGs function as modulators of the innate immune response, highlighting the multifunctional nature of these conglomerates[35]. SGs play a major role in homeostatic cellular stress responses, whereas alterations in their assembly and/or clearance are associated with various human pathologies including neurodegenerative and neoplastic diseases and

viral infections[36]. SG-associated proteins have also been reported to exert physiological functions in other types of RNP complexes[37–39]. SGs contain a compendium of RNA-binding proteins (RBPs), including Ras-GTPase-activating protein-binding protein 1 (G3BP1), T-cell-restricted intracellular antigen-1 (TIA-1), and Y-box binding protein 1 (YB-1)[40–43]. YB-1 is a DNA/RNA-binding protein that regulates transcription, translation, DNA repair, and SG formation in part by translationally activating G3BP1[40].

The function of ZO-1 is governed by its association with numerous transmembrane, cytoplasmic, and cytoskeletal proteins. Herein, we identified the VEGF-modulated ZO-1 protein interactome, which revealed previously unknown interactions between ZO-1 and RBPs that are known components of SGs, including YB-1. We found that the downregulation of ZO-1 in ECs and in the developing retinal vasculature of mice enhances SG formation and is protective against cellular stresses during angiogenesis and that these effects occur in a YB-1-dependent manner. Our results provide compelling evidence that, in addition to its function at cell junctions, ZO-1 plays a crucial role in regulating SG formation in ECs during angiogenesis. This pivotal function is facilitated by the interaction of ZO-1 with YB-1, therefore establishing a paradigm where a key junctional protein and an RBP communicate to modulate the formation and function of these subcellular structures.

## Results

### Proteomic profiling of the ZO-1 protein interactome

To identify the protein interactome of ZO-1 in ECs, bovine aortic endothelial cells (BAECs) were treated with VEGF or a vehicle, followed by ZO-1 immunoprecipitation (IP) and liquid chromatography-tandem mass spectrometry (LC-MS/MS) analysis. ZO-1 interactors were determined through statistical analyses comparing the IPs of the bait protein (ZO-1) for each condition with IPs of an IgG control (Supplemental Fig. S1a). In total, we identified 131 proteins as putative ZO-1-interacting proteins; of these, 63 were affected by the VEGF treatment (Fig. 1a). In particular, VEGF treatment enhanced the interaction of ZO-1 with 62 proteins but hindered its interaction with only one protein (BTF3) (Fig. 1a, Supplementary Data 1, Supplemental Fig. S1b). A protein-protein interaction network was generated, and proteins were grouped based on their function as determined by the UniProt database or by Gene Ontology (GO) enrichment analyses (Fig. 1b). As expected, we found that ZO-1 (*TJP1*) interacts with proteins involved in the cell-cell junction and cytoskeleton organization, including ZO-2 (*TJP2*), gap junction protein-1 (*GJA1*), desmoplakin (*DSP*), cingulin (*CGNL1*), junction plakoglobin (*JUP*), drebrin-1 (*DBN1*), and myosins (*MYL1/6, MYH11*) (Fig. 1b)[24,44–47]. Even though many of the identified ZO-1 partners have not been previously reported, our MS results confirmed the interactions of ZO-1 with JUP, ZO-2, cingulin, and YB-3 (Fig. 1b)[22,26,48]. Furthermore, we confirmed the previously identified ZO-1/JUP and ZO-1/YB-3 interactions by IP and found that VEGF treatment increased the association between ZO-1 and JUP (Fig. 1b, c). However, VEGF hindered the ZO-1/YB-3 interaction, which was not detected by MS (Fig. 1b, Supplemental Fig. S1c).

### ZO-1 interacts with the RBPs involved in stress granule formation

GO term enrichment analyses of the proteins that interact with ZO-1 revealed that many are RBPs involved in ribonucleoprotein complex (GO:1990904), RNA binding (GO:0003723), RNA processing (GO:0006396), and translation (GO:0006412) (Fig. 1b, d, Supplemental Fig. S1d, e). We found 55 RBPs as ZO-1 interactors and VEGF treatment enhanced the interaction of ZO-1 with 25 of them (Supplementary Data 1). We then validated the interactions of ZO-1 with two of these RBPs (AIMP1 and RPL23A) by performing ZO-1 IPs (Fig. 1c). The unprecedented associations between ZO-1 and RBPs prompted us to compare the proteomics data of the ZO-1 interactome with those we

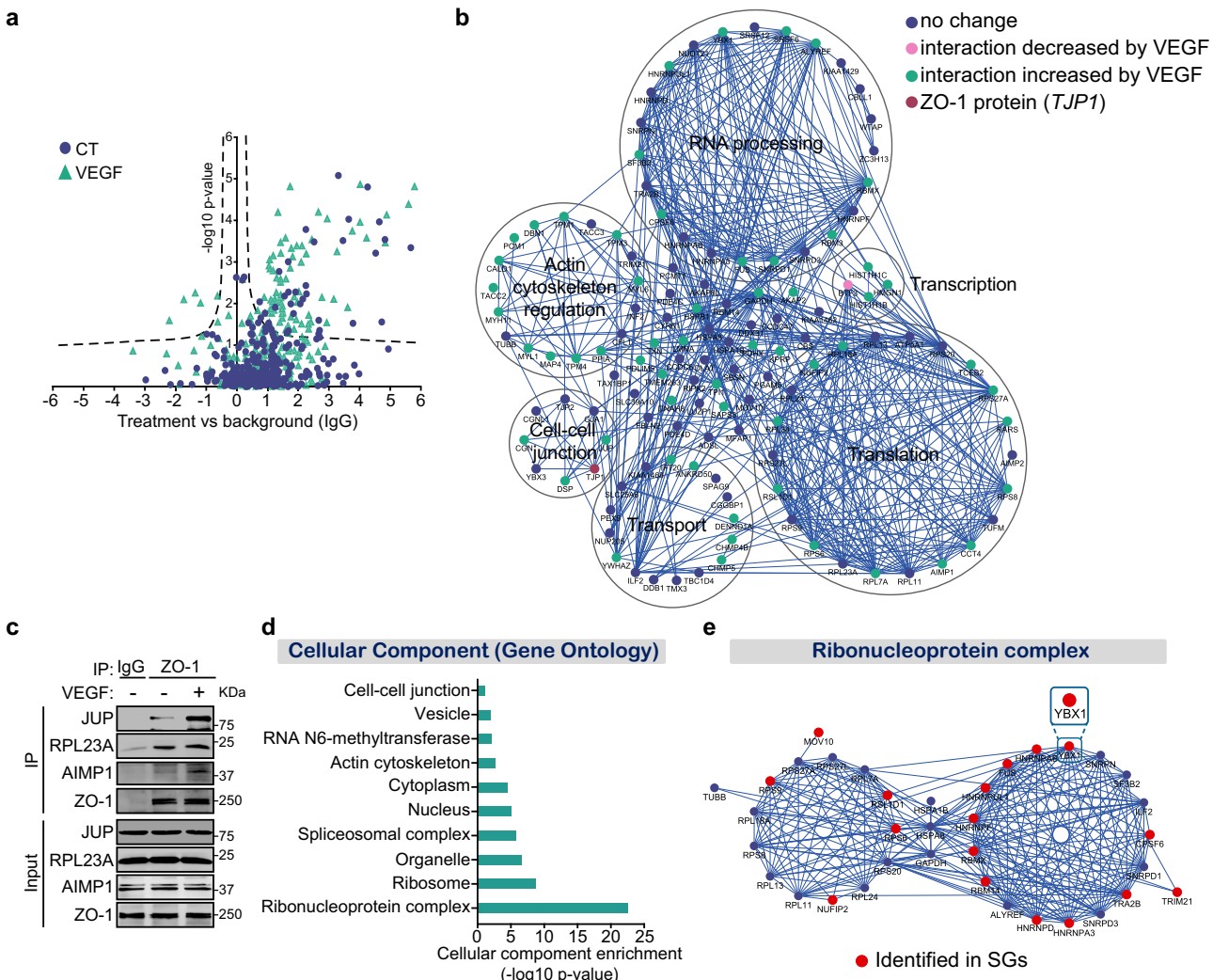

**Fig. 1 | Proteomic analysis of the ZO-1 interactome in endothelial cells.**
**a** Volcano plot showing the distribution of proteins abundance of control (CT) and VEGF treated conditions over IgG control (fold-change, x-axis) as a function of statistical significance (-log10 of p-value, y-axis). The hyperbolic curves indicate a false discovery rate (FDR) cutoff of 0.05 and separate specific from non-specific interactors. Every point represents one protein. **b** Protein interaction network of ZO-1 (*TJP1*, red) interacting proteins using STRING database and visualized by Cytoscape. The functional enrichment was manually annotated using GO terms of the biological process category and Uniprot database. Green corresponds to ZO-1-associated proteins that were increased by VEGF treatment. Pink corresponds to ZO-1-associated proteins that were decreased by VEGF treatment. **c** Validation of

the interaction between ZO-1 and interactors identified by mass spectrometry. Proteins were immunoprecipitated using ZO-1 antibody from whole lysates of BAEC treated or not with VEGF (40 ng/ml, 10 min). Then, interacting proteins were detected by immunoblot using specific antibodies against ZO-1, JUP, AIMP1 and RPL23A proteins. Non-immune IgG serves as a control for non-specific co-immunoprecipitation. Immunoblots were repeated twice with identical results.
**d** Enrichment analysis for GO cellular component terms of ZO-1 interacting proteins. **e** Ribonucleoprotein complex interaction network. Protein interaction network of ZO-1 interacting proteins part of the ribonucleoprotein complex GO term (GO:1990904). Red corresponds to proteins associated with stress granules (SGs).

---

had previously obtained for the interactome of β-catenin, another cytosolic junctional protein (Supplementary Data 2). Although, 42% of the 131 ZO-1-interacting proteins are RBPs, only 18% of the 62 β-catenin interactors are RBPs. Furthermore, only three RBPs are common to the β-catenin and ZO-1 interactomes (PPIA, RPL11, and RPS27A) (Supplementary Data 2). This low abundance of RBPs in the β-catenin interactome indicates that the identified interactions between ZO-1 and RBPs are specific.

Moreover, we noted that 22 RBPs that interact with ZO-1 are part of the ribonucleoprotein complex GO term (GO:1990904) and are also annotated in the RNA Granule Database (http://rnagranuledb.lunenfeld.ca/) as proteins associated with stress granules (SGs) (Fig. 1e)[49]. For instance, we identified YB-1 (*YBX1*) and FUS−known to be implicated in SG formation in cells under stress conditions−as ZO-1-associated proteins[40,50] (Fig. 1b, e). Hence, the ZO-1 protein

interactome revealed previously unknown associations between ZO-1 and a network of RBPs involved in SG formation.

## ZO-1 expression levels modulate stress granule formation in ECs

To investigate the link between ZO-1 and SG-related proteins, we performed siRNA-mediated depletion of ZO-1 and monitored SG formation following the exposure of ECs to stress. Human umbilical vein endothelial cells (HUVECs) or BAECs transfected with two distinct siRNAs targeting ZO-1 showed a marked increase in SG formation in response to arsenite treatment relative to control transfected cells (Fig. 2a, b, Supplemental Fig. S2a−c). Downregulation of ZO-1 increased both the percentage of ECs that exhibited G3BP1-positive SGs and the number of SGs per cell in response to the arsenite treatment (Fig. 2a, b, Supplemental Fig. S2b, c). Moreover, ZO-1 depletion did not induce SG formation in control non-stressed ECs (Fig. 2a,

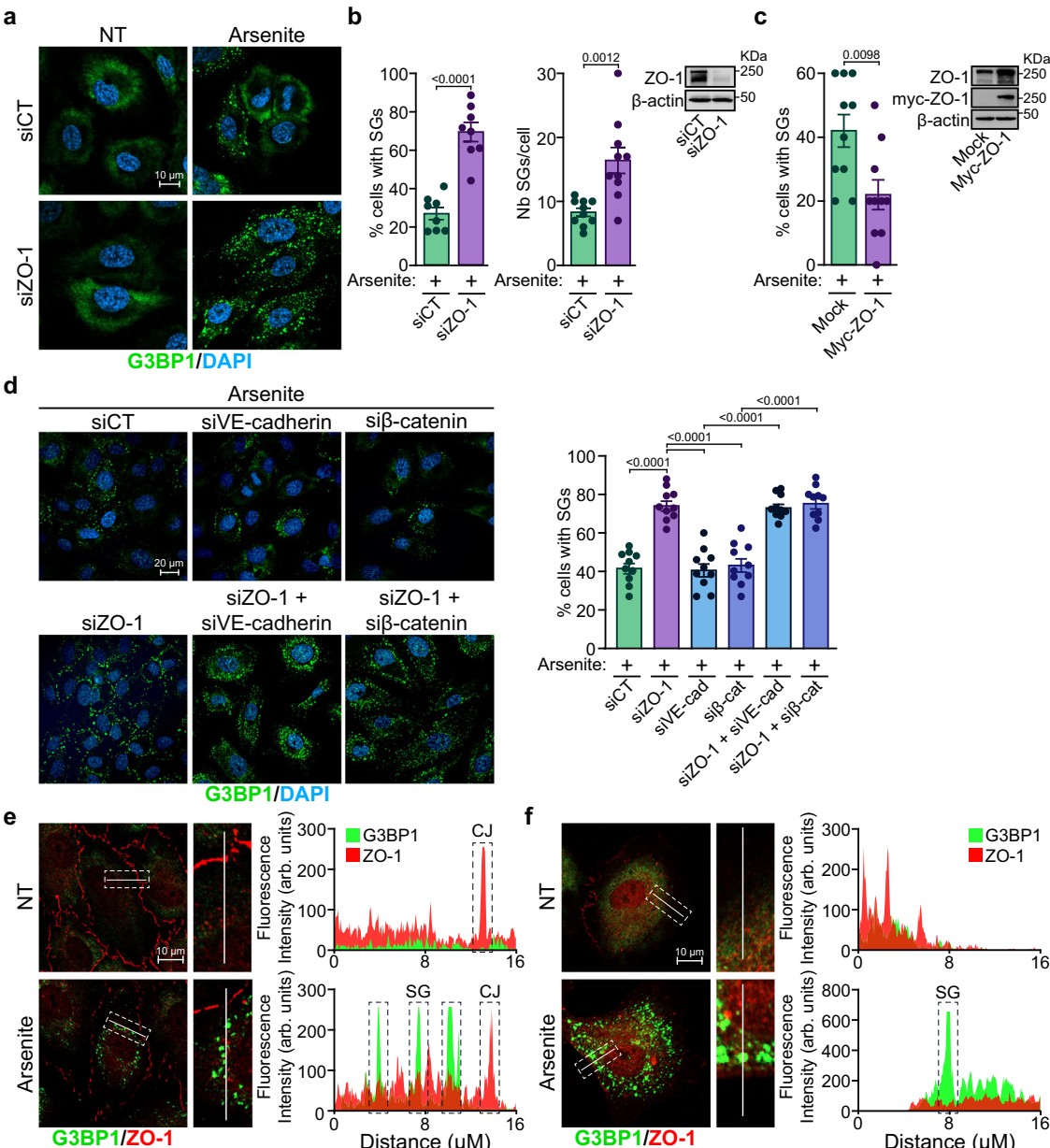

**Fig. 2 | ZO-1 depletion increases SG formation in ECs. a** Increased SG formation in ZO-1 depleted cells. Immunofluorescence staining of SGs using antibodies against G3BP1 in siCT and siZO-1 transfected HUVECs in absence (NT) or presence of sodium arsenite (500 μM for 30 min). Nuclei are stained with DAPI. **b** Quantification of G3BP1-positive SGs shown in (**a**). The percentage of cells with SGs (left; n = 8 fields of view per condition and at least 25 cells/field) and the number of SGs per cell (right; n = 10 fields of view per condition and at least 10 cells/field) are shown. Unpaired two-tailed Student's *t* test. The downregulation efficiency of siZO-1 was confirmed by immunoblotting. **c** Overexpression of ZO-1 decreases SG formation. BAECs expressing myc-ZO-1 were treated with arsenite (500 μM; 30 min) and the percentage of cells with SGs (G3BP1) was quantified (n = 10 fields of view per condition and at least five myc-positive cells/field). Unpaired two-tailed Student's *t* test. The expression of myc-ZO-1 plasmid was confirmed by immunoblotting.

**d** Increased SG formation is specific to the downregulation of ZO-1. (Left) Immunofluorescence staining of SGs in HUVECs transfected with siRNA against ZO-1, VE-cadherin, siβ-catenin or ZO-1 and VE-cadherin or ZO-1 and β-catenin and treated or not with arsenite. (Right) Quantification of the percentage of cells with G3BP1-positive SGs (n = 10 fields of view per condition and at least 25 cells/field). One-way ANOVA followed by Bonferroni's multiple comparison tests. Efficacy of siRNA-mediated knockdowns is shown in Supplemental Fig. S2g. ZO-1 does not localize in SGs of confluent and sparse ECs. Immunofluorescence staining of ZO-1 and G3BP1 in confluent (**e**) or sparsely (**f**) plated HUVECs treated or not with arsenite (500 μM; 30 min). The fluorescence intensities of ZO-1 (red) and G3BP1 (green) along the white lines are represented as arbitrary units (arb. units). Fluorescence intensities across stress granules (SG) and cell junctions (CJ) are indicated. Data are presented as mean values ± SEM. Source data are provided in the Source Data file.

Supplemental Fig. S2a, b). Similar to arsenite treatment, SG formation in response to heat shock (42 °C) was substantially increased in ZO-1-downregulated ECs (Supplemental Fig. S2d). On the other hand, SG formation was increased in arsenite-treated hTERT-immortalized human aortic ECs (TeloHAECs), where ZO-1 was deleted using CRISPR/Cas9, compared to wildtype TeloHAECs (Supplemental Fig. S2e). In contrast to the effects caused by the downregulation of ZO-1, we found

that the overexpression of ZO-1 in BAECs by transfection of myc-tagged ZO-1 reduced the percentage of cells with SGs in response to arsenite treatment compared to mock-transfected cells (Fig. 2c). We also observed that these effects appeared in cells other than ECs. For instance, the downregulation of ZO-1 in the human hepatoma cell line HepG2 resulted in increased SG formation following treatment with arsenite (Supplemental Fig. S2f). These results indicate that cellular

ZO-1 expression levels are negatively correlated with SG formation following exposure of ECs or other cell types to stress conditions.

Next, we investigated whether the downregulation of other cell junction proteins also resulted in increased SG formation. In contrast to the downregulation of ZO-1, siRNA-mediated depletion of VE-cadherin, β-catenin, ZO-2, or claudin-5 (a tight junction integral membrane protein associated with ZO-1) did not increase SG formation in response to arsenite (Fig. 2d, Supplemental Fig. S2g–i). Nonetheless, the knockdown of VE-cadherin or β-catenin combined with the knockdown of ZO-1 induced SG formation at a similar level as ZO-1 downregulation alone (Fig. 2d). This indicates that the downregulation of ZO-1, which increases the formation of SGs in response to stress, is independent of other cell junction proteins. Furthermore, the downregulation of ZO-1 in sparsely plated ECs was still able to boost SG formation in response to arsenite treatment, indicating that cell-cell contacts between ECs are unnecessary for inducing SG formation by ZO-1 depletion (Supplemental Fig. S2j).

### ZO-1 levels decrease upon exposure of ECs to stress

Our results revealed that ZO-1 interacts with a subset of RBPs associated with SGs; thus, we examined whether ZO-1 could be present in SGs and colocalize with G3BP1 in HUVECs treated with arsenite. Arsenite treatment was performed in confluent or sparsely plated HUVECs (Fig. 2e, f). In untreated confluent ECs, most ZO-1 was localized at cell-cell junctions and G3BP1 was diffusely present in the cytoplasm. By contrast, in arsenite-treated ECs, ZO-1 level at the cell junctions decreased but its presence in the cytoplasm slightly increased. As expected, arsenite induced the localization of G3BP1 to SGs (Fig. 2e). However, there was no detectable colocalization between ZO-1 and G3BP1 (Pearson's colocalization coefficient ZO-1/G3BP1 = 0.036) under either condition. Hence, in sparsely plated HUVECs, no colocalization was observed in untreated or arsenite-treated ECs in the absence of cell-cell junctions, even though ZO-1 was predominantly observed in the cytoplasm (Pearson's colocalization coefficient ZO-1/G3BP1 = 0.0068) (Fig. 2f).

Next, we examined the effect of arsenite-induced stress on ZO-1 localization at cell-cell junctions in ECs. Arsenite treatment of HUVECs resulted in a marked decrease in the ZO-1 fluorescence intensity at EC-junctions but no effect was observed on the VE-cadherin and β-catenin levels (Fig. 3a). The decrease in the cellular levels of ZO-1 in arsenite-treated ECs prompted us to quantify the effects of various stressors on the ZO-1 levels, using immunoblotting. ECs were treated in a dose-dependent manner with arsenite, hydrogen peroxide, or thapsigargin. We found that all three treatments markedly decreased ZO-1 expression but did not affect the VE-cadherin or β-catenin levels (Fig. 3b–d). In addition, the ZO-1 levels were significantly reduced after only 30 min of arsenite treatment (500 μM) (Fig. 3e). Because of the rapid effect of arsenite on ZO-1 levels, we investigated whether ZO-1 protein degradation is proteasome-mediated. As a result, we found that inhibiting the proteasome using MG132 attenuated the arsenite-induced decrease in ZO-1 protein levels (Fig. 3f). These findings suggest that exposure of ECs to stress provoked the rapid proteasome-mediated degradation of ZO-1. Next, we examined the presence of SGs and the levels of ZO-1 following a recovery period after 1 h of exposure to arsenite. Subsequent to arsenite removal, ZO-1 levels remained decreased for 1 h, and the number of ECs with SGs remained elevated (Fig. 3g, h). After 3 h of recovery, ZO-1 levels began to return to normal, coinciding with an absence of cells positive for SGs (Fig. 3g, h). This demonstrates that the stress-induced decrease in ZO-1 levels correlates with SG formation in ECs.

### Downregulation of ZO-1 protects ECs against stress conditions

SGs assemble during periods of stress to protect cells[51], and a decrease in ZO-1 levels boosts SG formation. Thus, we investigated whether the downregulation of ZO-1 is cytoprotective. We found that the

downregulation of ZO-1 protected ECs against cell death and hindered caspase-3 cleavage, both induced by arsenite treatment (Fig. 4a, b, Supplemental Fig. S3a). Notably, similar results were found in sparsely plated ECs, suggesting that these effects also occur in the absence of cell-cell junctions (Supplemental Fig. S3b). By contrast, the downregulation of VE-cadherin or β-catenin did not protect ECs from arsenite-induced death and did not prevent the cleavage of caspase-3 (Fig. 4a, b). Furthermore, the downregulation of VE-cadherin even resulted in a decrease in the viability of untreated ECs. Moreover, the downregulation of ZO-1 in VE-cadherin- or β-catenin-downregulated cells also protected them against arsenite-induced cell death (Supplemental Fig. S3c). Overall, these results indicate that a decrease in ZO-1 expression levels protects ECs against stress conditions.

We have previously demonstrated that the downregulation of ZO-1 increases the VEGF-stimulated proliferation of ECs[11]. To examine the effects of arsenite treatment on EC proliferation induced by ZO-1 downregulation, we counted the number of proliferating cells by measuring bromodeoxyuridine (BrdU) incorporation in control and ZO-1-downregulated ECs in the absence or presence of arsenite. While the downregulation of ZO-1 increased cell proliferation in the control (untreated ECs), arsenite treatment decreased the number of BrdU-positive cells to similar levels to those in siCT- and siZO-1-transfected cells (Supplemental Fig. S3d). These results indicate that arsenite-induced stress inhibits proliferation equally in ZO-1-downregulated and control ECs. Subsequently, we determined whether VEGF treatment influenced SG formation and ZO-1 levels in ECs. As previously reported, prolonged VEGF treatment of HUVECs lowered ZO-1 expression (Supplemental Fig. S3e)[52]. Furthermore, VEGF pre-treatment protected ECs against arsenite-induced cell death (Fig. 4c). However, in ZO-1-downregulated ECs, VEGF pre-treatment significantly decreased the arsenite-induced SG formation while maintaining cell viability (Fig. 4c, d). These results suggest that the antiapoptotic effects of VEGF on ECs do not involve the formation of SGs and differ from the protective effects of ZO-1 against stress, which involve SG formation.

Afterward, we examined whether stress-induced SG formation and ZO-1 levels were modulated during EC migration. Hence, we performed 2D wound-healing assays on confluent HUVEC monolayers transfected with control or ZO-1 siRNA. These experiments revealed that in migrating and arsenite-treated ECs, those at the leading edge have more SGs and express less ZO-1 protein than follower cells (Fig. 4e–g). The downregulation of ZO-1 in migrating and arsenite-treated ECs boosted the formation of SGs in follower cells without further increasing the number of SGs in cells located at the leading edge (Fig. 4e, f). Furthermore, no SGs were observed in migrating siCT- or siZO-1-transfected ECs not subjected to arsenite treatment (Fig. 4f).

Next, we monitored the effects of ZO-1 downregulation and arsenite treatment on basal and VEGF-stimulated EC migration. Downregulation of ZO-1 significantly increased the migration of untreated ECs compared to control-transfected cells (Fig. 4h, i). On the other hand, arsenite treatment of control-transfected ECs hampered their migration, which was restored in ZO-1-downregulated cells. Finally, VEGF treatment induced the migration of siCT-transfected ECs but did not lead to an additional increase in the migration of siZO-1-transfected ECs in the absence or presence of arsenite (Fig. 4h, i). These results indicate that the downregulation of ZO-1 promotes the formation of SGs, thereby facilitating cell migration in the presence of arsenite, and that the formation of SGs is not involved in VEGF-stimulated migration or antiapoptotic effects.

### Stress conditions disrupt the interaction between ZO-1 and YB-1

To identify the mechanism by which ZO-1 affects SG formation in ECs, we focused on YB-1, a component of SGs present in the ZO-1 interactome[40,43,53]. Notably, we found that the interaction between YB-1 and ZO-1 increased significantly in response to VEGF treatment

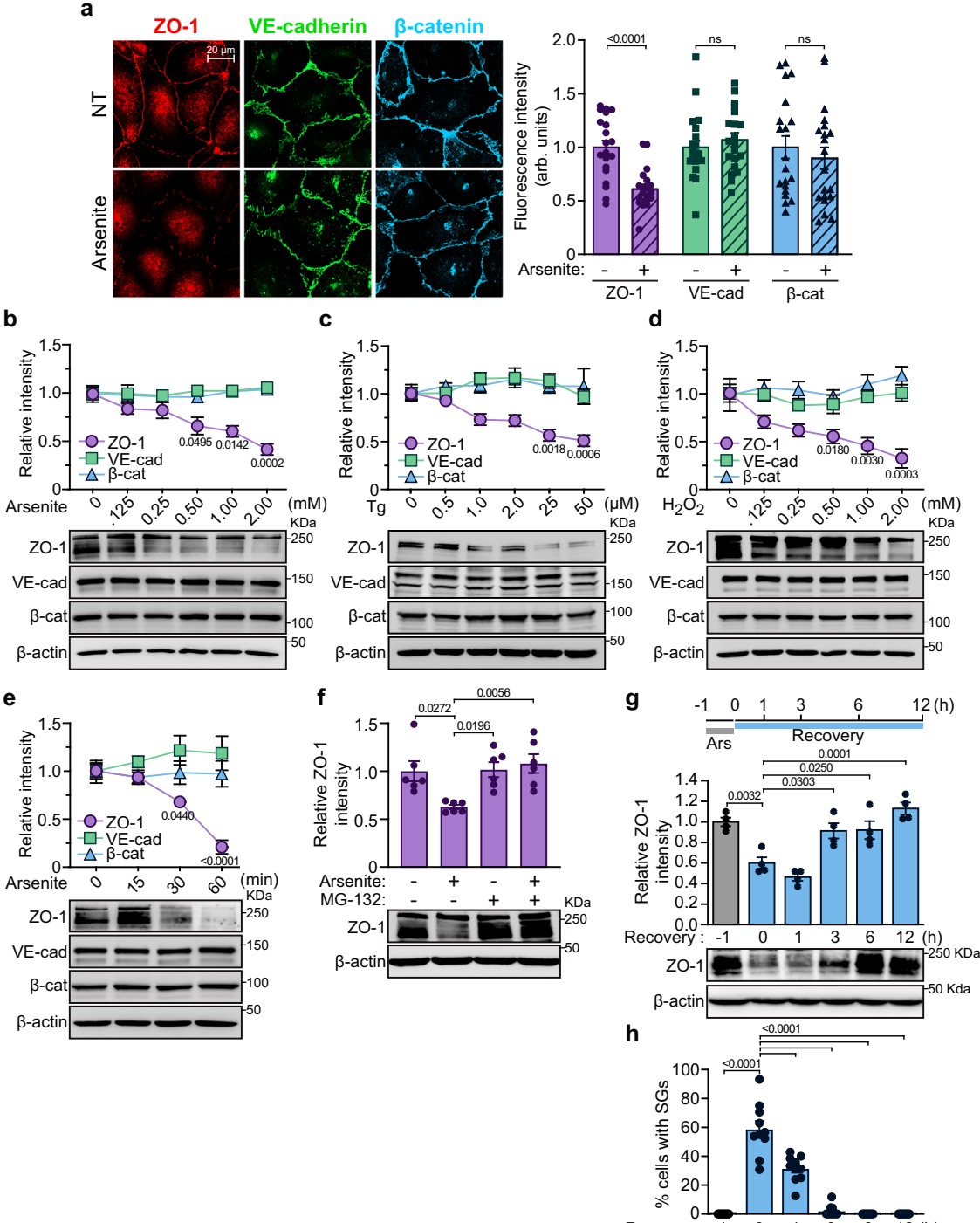

**Fig. 3 | Stress-inducing treatments decrease ZO-1 expression levels in ECs.**
**a** Arsenite treatment decreases ZO-1 localization at EC junctions without affecting the localization of VE-cadherin and β-catenin. The fluorescence intensity of ZO-1 (red), VE-cadherin (green) and β-catenin (blue) was measured in confluent cells in the absence (NT) or presence of sodium arsenite (500 μM; 30 min). The fluorescence intensity of each condition is normalized to the average intensity of the control (n = 20 cell junctions per condition). Unpaired two-tailed Student's *t* test. **b, c, d** Effects of increasing concentrations of arsenite (30 min; n = 3) (**b**), thapsigargin (2 h; n = 3) (**c**) or $H_2O_2$ (30 min; n = 3) (**d**) on expression levels of ZO-1, VE-cadherin and β-catenin in HUVECs (**b** and **d**) or BAECs (**c**). Protein expression levels were determined by immunoblotting and β-actin serves as a loading control. Comparisons of basal expression levels versus treated conditions were performed for each. **e** Kinetics of the effects of arsenite treatment (500 μM) on expression levels of ZO-1, VE-cadherin and β-catenin in HUVECs. Protein expression levels were determined by immunoblotting and β-actin serves as a loading control (n = 4 independent experiments). **f** Effect of the proteasome inhibitor MG132 (40 μM; 60 min) on arsenite-induced (500 μM; 30 min) downregulation of ZO-1 in HUVECs. ZO-1 expression levels were determined by immunoblotting and β-actin serves as a loading control (n = 6 independent experiments). Effects of a recovery period post-arsenite treatment (1 h; 500 μM) on ZO-1 levels (n = 4 independent experiments) (**g**) and percentage of HUVECs with SGs (n = 10 fields of view per condition and at least 15 cells/field) (**h**). **b–f** Each point of the line graphs refers to the quantification of protein levels relative to β-actin. **b–h** Groups were compared using one-way ANOVA followed by Bonferroni's multiple comparison tests; ns not significant. Data are presented as mean values ± SEM. Source data are provided in the Source Data file.

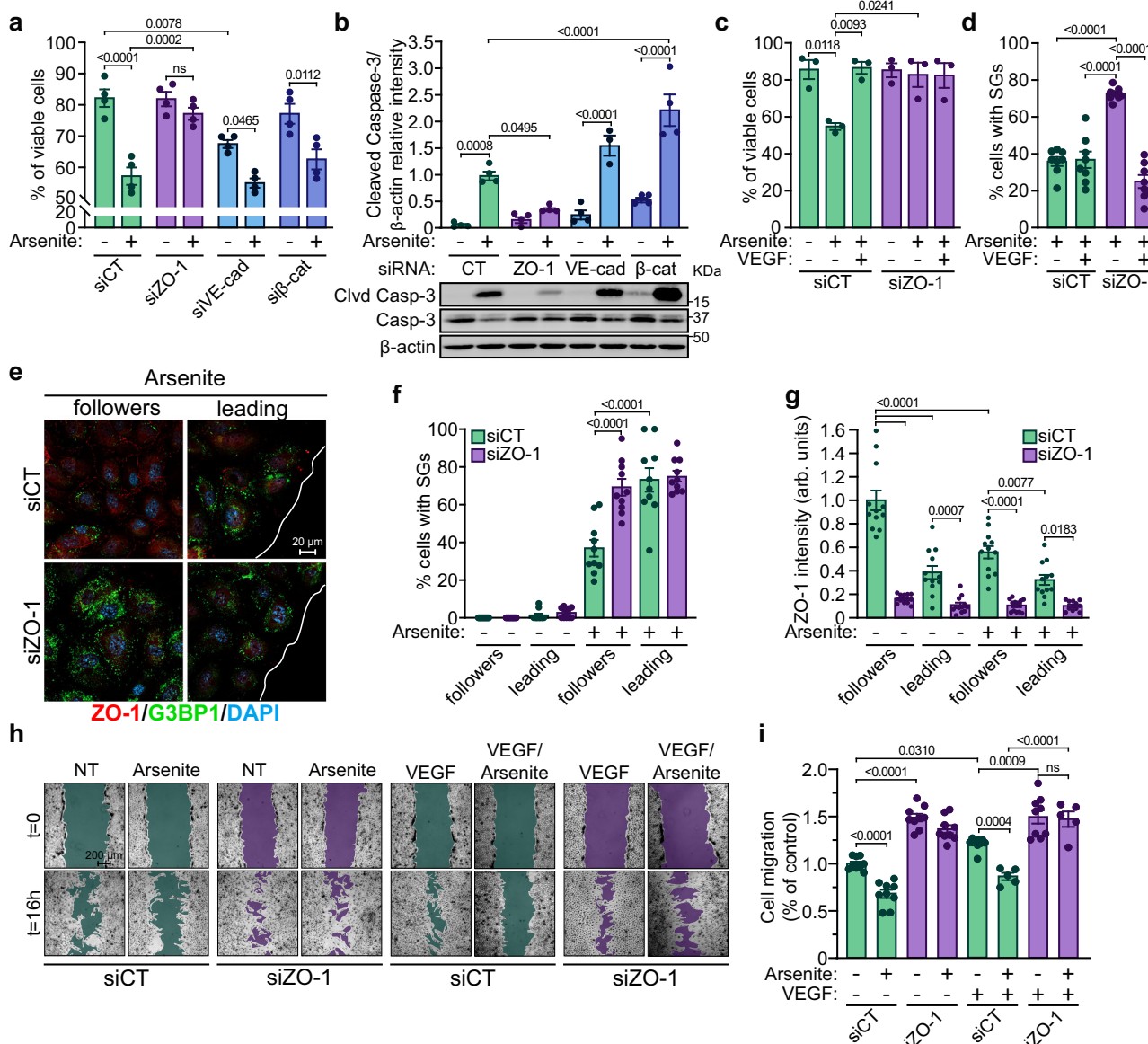

**Fig. 4 | ZO-1 depletion increases viability and protects ECs against cellular stress. a** ZO-1 depletion protects ECs against cell death induced by arsenite treatment. Cell viability, measured by Trypan blue exclusion, of HUVECs transfected with siRNA against ZO-1, VE-cadherin or β-catenin and treated or not with arsenite (25 µM; 12 h) (n = 4 independent experiments). **b** Downregulation of ZO-1 decreases caspase-3 cleavage. Immunoblot analysis of cleaved caspase-3 levels in siCT, siZO-1, siβ-catenin and siVE-cadherin transfected BAECs with siRNA and treated or not with arsenite (50 µM; 6 h). Histogram referring to the quantification of cleaved caspase-3 levels relative to β-actin (siVE-cadherin + arsenite n = 3; all other conditions n = 4). Efficacy of siRNA-mediated knockdowns is shown in Supplemental Fig. S3a. **c** Effect of VEGF treatment (40 ng/ml) on cell viability, measured by Trypan blue exclusion, of siCT- or siZO-1-transfected HUVECs in presence of sodium arsenite (25 µM; 12 h) (n = 3 independent experiments). **d** Effect of VEGF treatment (40 ng/ml; 1 h) on the induction of G3BP1-positive SGs by sodium arsenite (500 µM for 30 min) in siCT- or siZO-1-transfected HUVECs (n = 8 fields of view per condition and at least 25 cells/field). **e, f, g** Effects of arsenite treatment on migrating ECs. Immunofluorescence analysis of SG formation and of ZO-1 levels in siCT or siZO-1 transfected HUVECs treated with arsenite (500 µM) or with arsenite and subjected to a wound healing

migration assay for 30 min. **e** Images of immunofluorescence staining of G3BP1 and ZO-1 of cells in 1st and 2nd rows (leading) and in 3rd to 7th rows (followers) relative to the edge of the wound (white line) are shown. Nuclei were stained with DAPI. **f** The percentage of cells with SGs was calculated in leading and follower migrating cells as shown in (**e**) (n = 10 fields of view per condition and at least 15 cells/field). **g** ZO-1 fluorescence intensity was measured in leading and follower migrating cells as shown in (**e**) and normalized in each condition to the average intensity of the control (n = 12 fields of view per condition and at least 12 cells/field). **h,** Images of wound healing migration assays of siCT- or siZO-1-transfected BAECs treated with sodium arsenite (10 µM) and/or VEGF (40 ng/ml). Images were taken immediately after wounding of the cell monolayer (t = 0 h) and after 16 h of migration (t = 16 h). **i** Quantification of the percentage of wound closure at 16 h of migration, normalized to the control, is shown (siCT, siCT + VEGF, siZO-1, siZO-1 + VEGF, siCT + VEGF, siZO-1 + VEGF n = 9 migration areas per condition; siCT + VEGF + arsenite, siZO-1 + VEGF + arsenite n = 5 migration areas per condition). **a–i** Groups were compared using one-way ANOVA followed by Bonferroni's multiple comparison tests; ns not significant. Data are presented as mean values ± SEM. Source data are provided in the Source Data file.

(Fig. 1b). YB-1 is a member of the Y-box factor family that includes YB-3, another well-known ZO-1-interacting protein[40,54]. We also identified YB-3, along with YB-1, in the ZO-1 interactome (Fig. 1b). We therefore examined the levels of YB-1 and YB-3 in ECs treated with arsenite for

6 h. Consistent with its role in SG formation, YB-1 levels increased in ECs treated with arsenite while the levels of YB-3—which is not involved in SGs—and ZO-1 decreased (Fig. 5a). In addition, the expression levels of YB-1 and phosphorylation of YB-1 at Ser102, a

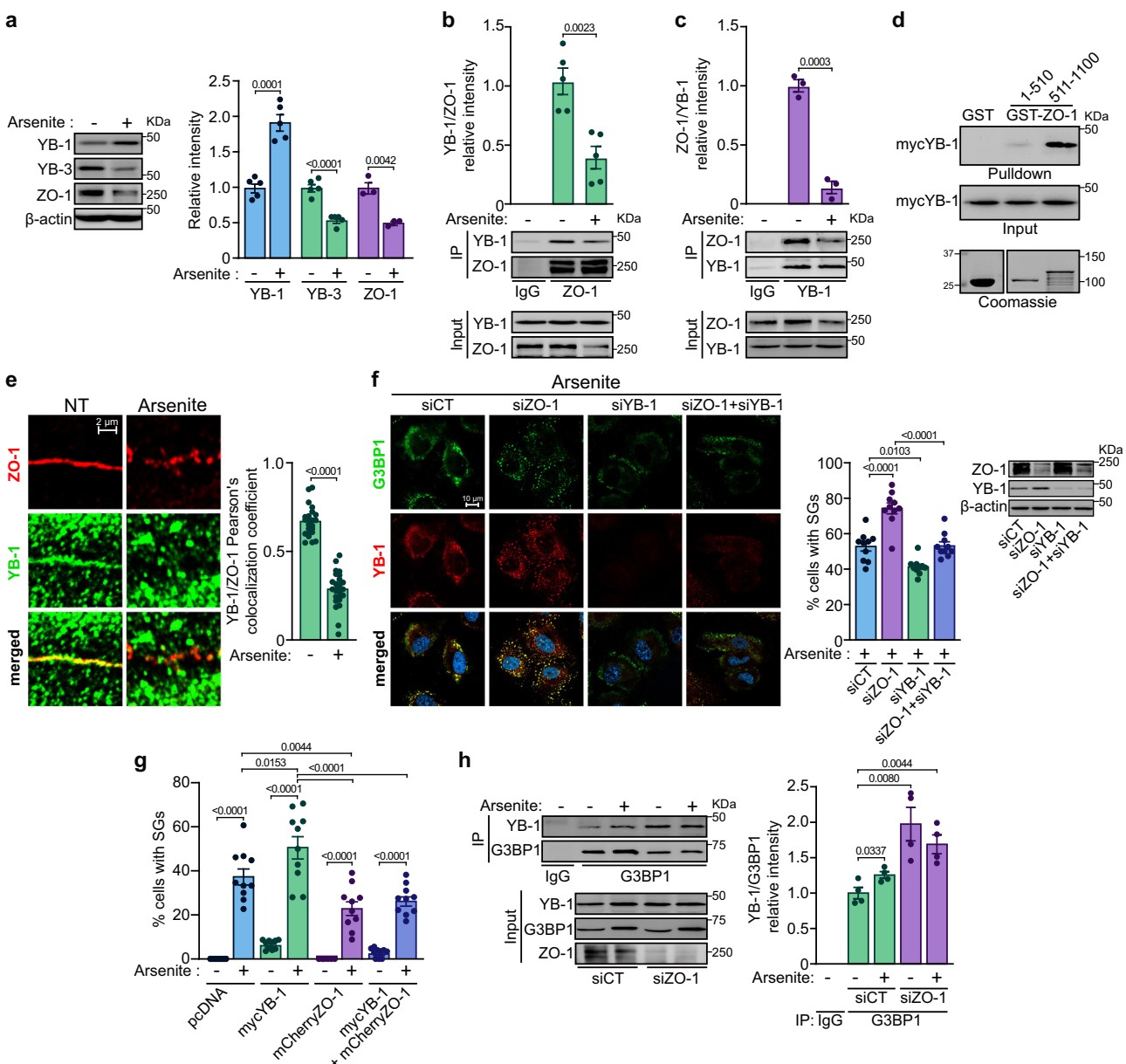

marker of its nuclear shuttling and transcriptional activity[55], increased in ZO-1-downregulated ECs (Supplemental Fig. S4a). In turn, the overexpression of ZO-1 in ECs resulted in decreased levels of YB-1 and pYB-1 (Supplemental Fig. S4b).

Subsequently, we examined the effect of arsenite treatment on the association between ZO-1 and YB-1 in ECs. YB-1 and ZO-1 coimmunoprecipitated from lysates of non-treated BAECs and this interaction was diminished in cells treated with arsenite (Fig. 5b, c). To further document the interaction between ZO-1 and YB-1, we performed pulldowns using ZO-1 fragments—residues 1-510 and 511-1100—fused to glutathione-S-transferase (GST) in lysates from BAECs expressing myc-tagged YB-1 (Supplemental Fig. S4c). We found that the fragment 511-1100 of ZO-1, which includes the SH3 and GUK domains, interacts preferentially with myc-YB-1 (Fig. 5d). We also show that YB-1 and ZO-1 partially colocalized at cell-cell junctions in untreated ECs and their colocalization was diminished in arsenite-treated ECs (Fig. 5e). Furthermore, in contrast to the interaction between YB-1 and G3BP1, the interaction between ZO-1 and YB-1 is not dependent on RNA since the treatment of the YB-1 IPs with RNAse or

benzonase did not disrupt the interaction (Supplemental Fig. S4d). Because we demonstrated that a prolonged arsenite treatment of HUVECs lowered ZO-1 protein levels in a proteasome-dependent manner (Fig. 3f), we then examined whether blocking ZO-1 degradation affected the arsenite-induced dissociation of the ZO-1/YB-1 interaction. As a result, treatment of ECs with MG132 did not prevent the dissociation of ZO-1 and YB-1 induced by low doses of arsenite (Supplemental Fig. S4e). This indicates that the dissociation of ZO-1 and YB-1 is initiated before ZO-1 degradation.

### YB-1 is required for ZO-1-induced SG formation

We examined the importance of YB-1 for SG formation in ZO-1-depleted ECs. ZO-1 depletion in arsenite-treated HUVECs boosted the formation of YB-1- and G3BP1-positive SGs (Fig. 5f). Downregulation of YB-1 in ECs diminished the percentage of cells containing SGs and the number of SGs per cell induced by the arsenite treatment. Furthermore, depletion of YB-1 hindered SG formation in ZO-1-depleted cells (Fig. 5f, Supplemental Fig. S4f). By contrast, overexpression of myc-tagged YB-1 in ECs drove SG formation in the absence or presence of

**Fig. 5 | The ZO-1/YB-1 interaction regulates SG formation. a** Effect of arsenite (50 μM; 6 h) on YB-1, YB-3 and ZO-1 protein levels in BAECs determined by immunoblot analysis. Histogram referring to the quantification of protein levels relative to β-actin. (YB-1 n = 5; YB-3 n = 5; ZO-1 n = 3 independent experiments). Unpaired two-tailed Student's *t* test. Arsenite treatment decreases the interaction between YB-1 and ZO-1. ZO-1 (**b**) or YB-1 (**c**) were immunoprecipitated (IP) from lysates of BAECs treated or not with arsenite (500 μM; 30 min). Non-immune IgG serves as control for non-specific co-immunoprecipitation. Levels of immunoprecipitated proteins were determined by immunoblot using the indicated antibodies. Histograms show the quantification of the ratio of YB-1 levels relative to ZO-1 (**b**, n = 5 independent experiments) or of ZO-1 levels relative to YB-1 (**c**, n = 3 independent experiments) present in the immunoprecipitates. Unpaired two-tailed Student's *t* test. **d** GST pulldown assay of YB-1 using ZO-1 fragments (aa. 1-510 and aa. 511-1100) fused to GST (schematic representation of domain organizations of ZO-1 is shown in Supplemental Fig. S4c). Lysates of myc-YB-1 expressing BAECs were incubated with ~50 pM of either GST alone, GST-ZO-1(1-510) or GST-ZO-1(511-1100) prebound to glutathione agarose beads. Association of myc-YB-1 was revealed by anti-myc immunoblotting. Input shows the quantity of protein used for GST pulldown. **e** Immunofluorescence staining images show the colocalization (yellow) of YB-1 (green) and ZO-1 (red), using antibodies against YB-1 or ZO-1, in HUVECs in the presence or absence of arsenite (500 μM; 30 min). The measure of colocation was determined by Pearson's colocalization coefficient of YB-1/ZO-1 on cell junctions of HUVEC (CT n = 23; CT + arsenite n = 25 cell junctions per condition). Unpaired two-

tailed Student's *t* test. **f** Decreased SG formation in YB-1 depleted cells. Images of immunofluorescence staining, and its measurement as a histogram, of SGs using antibodies against G3BP1 or against YB-1 in siCT, siZO-1, siYB-1 and combination of siZO-1 plus siYB-1 transfected HUVECs in presence of arsenite (500 μM; 30 min). Nuclei are stained with DAPI. The percentage of cells with SGs was calculated in n = 10 fields of view per condition and at least 25 cells/field. One-way ANOVA followed by Bonferroni's multiple comparison tests. The downregulation efficiency of siRNA was confirmed by immunoblot. **g** Effect of overexpression of YB-1, ZO-1 or both on SG formation in ECs. BAECs expressing mycYB-1, mCherryZO-1 or both were treated with arsenite (500 μM; 30 min) and the percentage of cells with SGs of n = 10 fields of view per condition and at least five transfected cells/field was quantified. One-way ANOVA followed by Bonferroni's multiple comparison tests. **h** Effect of arsenite on YB-1 and G3BP1 association in siCT or siZO-1 transfected HUVECs. G3BP1 was immunoprecipitated (IP) from lysates of BAECs transfected with siCT or siZO-1 and treated or not with arsenite (500 μM; 30 min). Non-immune IgG serves as control for non-specific co-immunoprecipitation. Levels of immunoprecipitated proteins were determined by immunoblot using the indicated antibodies. Histogram showing the quantification of the ratio of YB-1 levels relative to immunoprecipitated G3BP1. Note that arsenite treatment increases G3BP1 levels in lysates (n = 4 independent experiments). Unpaired two-tailed Student's *t* test. Data are presented as mean values ± SEM. Source data are provided in the Source Data file.

arsenite but this process was inhibited by the coexpression of mCherry-tagged ZO-1 (Fig. 5g). YB-1 and G3BP1 are both markers of SGs and their interaction generally reflects SG formation[40,41,43]. Thus, we examined the effect of arsenite treatment in ZO-1-downregulated ECs on the YB-1 and G3BP1 interaction. We show that arsenite treatment of ECs did increase the coimmunoprecipitation of YB-1 and G3BP1 and, interestingly, that the interaction between YB-1 and G3BP1 was augmented in untreated and ZO-1-downregulated ECs (Fig. 5h). However, this interaction was not further increased by arsenite treatment (Fig. 5h). Overall, these results suggest that YB-1 is necessary for the induction of SG formation in ECs where ZO-1 is downregulated and that the interplay between ZO-1 and YB-1 regulates the assembly of SGs in ECs.

## YB-1 is necessary for the cytoprotective effects of ZO-1 downregulation

We have shown that downregulation of ZO-1 protected ECs against arsenite-induced cell death (Fig. 4a, b) and enhanced cell migration (Fig. 4h, i). Hence, we examined the role of YB-1 in these cytoprotective effects. Downregulation of YB-1 alone did not affect the induction of cell death by arsenite treatment (Fig. 6a). However, in contrast to ZO-1-depleted cells that were protected from arsenite-induced death, the simultaneous downregulation of both YB-1 and ZO-1 abolished the protective effects provided by the downregulation of ZO-1 alone (Fig. 6a). Similarly, in wound-healing migration assays, downregulation of YB-1 alone inhibited the migration of untreated ECs, which is in sharp contrast to the increased migration seen in ZO-1-depleted ECs (Fig. 6b). Thus, downregulation of YB-1 resulted in more potent inhibition of migration by arsenite compared to control-treated ECs. While arsenite treatment did not inhibit the migration of ZO-1-downregulated ECs, arsenite treatment inhibited the migration of cells in which both ZO-1 and YB-1 were depleted (Fig. 6b). Moreover, the downregulation of YB-1 in ECs resulted in lowered expression of BCL-XL and G3BP1, two known transcriptional targets of YB-1. By contrast, depletion of ZO-1 increased the expression of YB-1, BCL-XL, and G3BP1 (Fig. 6c, Supplemental Fig. S4a). These results demonstrate that YB-1 is necessary for the protective effects against stress that are induced by the downregulation of ZO-1 in ECs. SG formation in response to stress is associated with a pause of translation, which is primarily mediated by the phosphorylation of eIF2α on Ser51. Arsenite treatment of HUVECs induced an increase in eIF2α phosphorylation on

Ser51 and this increase was significantly greater in ZO-1-downregulated cells. Interestingly, YB-1 depletion did not affect phospho-eIF2α levels in ECs that were also depleted of ZO-1 (Fig. 6d). This suggests that, in contrast to its role in SG formation, YB-1 is nonessential for the increase in the phospho-eIF2α levels caused by the downregulation of ZO-1.

## Regulation of ZO-1 and YB-1 in the developing retinal vasculature

To assess whether ZO-1 levels in ECs are regulated during angiogenesis in vivo, we examined the vasculature of mouse retinas at postnatal day 7 (P7), when the outward growth of the vasculature is yet incomplete. We found that ZO-1 levels were lowered in ECs located at the vascular front of the retina, which includes tip ECs at the extremities, compared to ECs located in the plexus (median portion) of the retinal vasculature (Fig. 7a). In particular, we found that the vascular front of retinas contained markedly more ECs with YB-1- and G3BP1-positive granules than the ECs located in the plexus of the retinal vasculature (Fig. 7b, Supplemental Fig. S5a).

Next, we reanalyzed our publicly available single-cell RNA sequencing (scRNA-seq) data from ECs isolated from P6 mouse retinas to determine the mRNA levels of ZO-1 and YB-1 in the EC subtypes[56]. The uniform manifold approximation and projection (UMAP) plot of Pecam1+;Cldn5+ EC clusters at P6 reveals the presence of arterial ECs (Unc5b+;Bmx+), capillary ECs (Mfsd2a^high;Slc22a8^high), venous ECs (Hmgn2^high;Ptgis^high), tip ECs (Kcne3+;Angpt2+), and proliferative ECs (Mki67+;Birc5+) (Fig. 7c)[56]. The expression levels of marker genes reveal the enrichment of Esm1 and Angpt2 in the tip EC cluster and of Top2a and Mki67 in the proliferative EC cluster. As previously reported, tip ECs have a low proliferative status (Fig. 7d)[14,56]. Gene expression analysis revealed that Tjp1 and Ybx1 mRNAs are differentially expressed in EC subtypes (Fig. 7e). In plexus-forming ECs, namely arterial ECs and capillary ECs, the expression levels of Tjp1 were elevated, whereas those of Ybx1 were low (Fig. 7e). However, venous ECs exhibited a higher expression of Ybx1 than of Tjp1. Pearson's correlation analysis indicated a strong to modest association between the expression of Tjp1 and Ybx1 in arterial ECs, venous ECs, and capillary ECs with scores of 0.59, 0.24, and 0.23, respectively. In line with previous findings related to ZO-1, proliferative ECs exhibited low Tjp1 expression but high Ybx1 levels, which is associated with a low Pearson's correlation score of 0.06[11]. In contrast to the ZO-1 protein levels seen at the vascular front, mRNA levels were elevated in tip ECs. However, Ybx1 mRNA

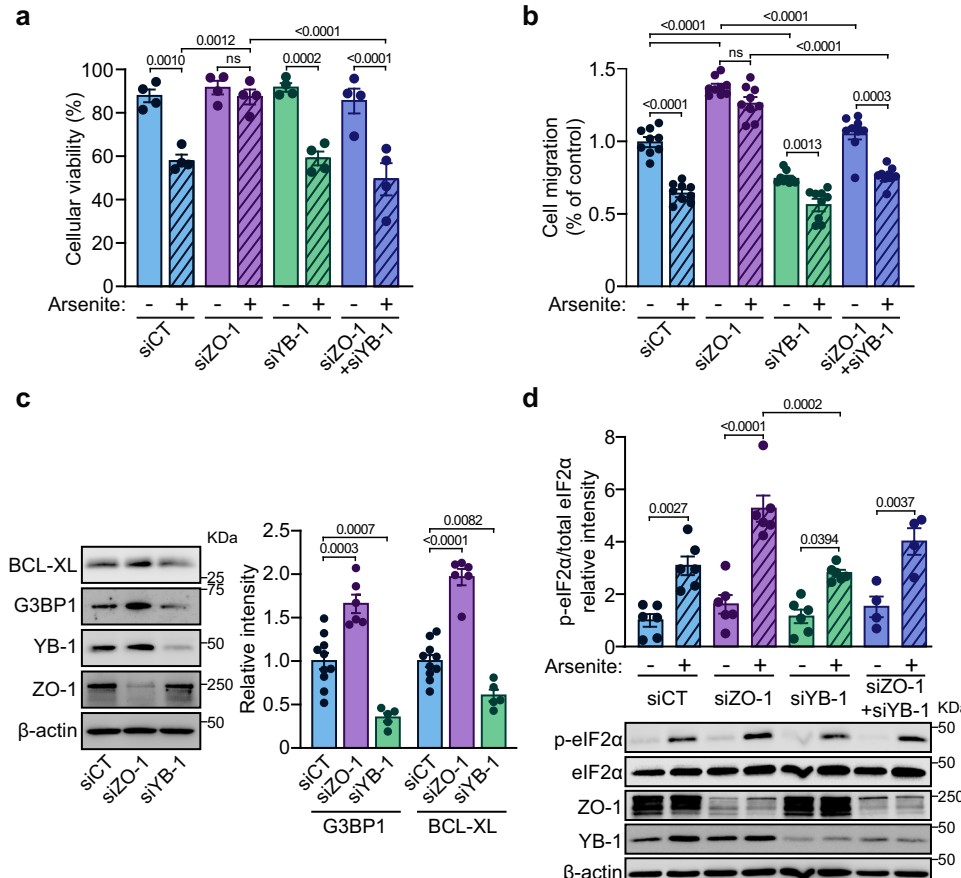

**Fig. 6 | YB-1 is necessary for the cytoprotective effects of ZO-1 downregulation.** **a** Cytoprotection induced by the absence of ZO-1 is not maintained in the absence of YB-1. Cell viability, measured by Trypan blue exclusion, of HUVECs transfected with siRNA against CT, ZO-1, YB-1 or ZO-1 and YB-1 and treated or not with arsenite (25 μM; 12 h) (n = 4 independent experiments). **b** EC migration induced by the absence of ZO-1 is not maintained in the absence of YB-1. Quantification of the percentage of wound closure at 16 h of migration, normalized to the control, of BAECs transfected with siCT, siZO-1, siYB-1 or ZO-1 and YB-1 treated or not with arsenite (10 μM) (n = 9 fields of view per condition). **c** Increased transcriptomic targets of YB-1 in the absence of ZO-1. Immunoblot analysis of BCL-XL or G3BP1 levels in siCT, siZO-1 and siYB-1 transfected HUVECs with siRNA. Histogram

referring to the quantification of BCL-XL or G3BP1 levels relative to β-actin (siCT n = 10, siZO-1 n = 6, siYB-1 n = 5 independent experiments). **d** ZO-1 influences translation initiation during EC stress independently of YB-1. Immunoblot analysis of serine 51 phosphorylation of eIF2α levels in siCT, siZO-1, siYB-1 or siZO-1 and siYB-1 transfected HUVECs and treated or not with arsenite (500 μM; 30 min). Histogram referring to the quantification of serine 51 phosphorylation of eIF2α relative to total eIF2α (siCT, siCT + arsenite, siZO-1, siZO-1 + arsenite, siYB-1, siYB-1 + arsenite n = 6; siZO-1 + siYB-1, siZO-1 + siYB-1 + arsenite n = 4 independent experiments). **a–d** Groups were compared using one-way ANOVA followed by Bonferroni's multiple comparison tests; ns not significant. Data are presented as mean values ± SEM. Source data are provided in the Source Data file.

levels were also elevated in line with the increase in YB-1-positive granules in ECs at the vascular front (Fig. 7b). Nevertheless, we found no linear correlation in the expression levels of *Tjp1* and *Ybx1* in tip ECs (Pearson's correlation score of 0.05). This suggests that, within each tip EC, the expression of *Tjp1* and *Ybx1* does not correlate, which agrees with the protein expression patterns of ZO-1 and YB-1 observed at the vascular front (Fig. 7a, b). Finally, the mRNA levels of claudin 5 (*Cldn5*) followed those of *Tjp1*, except for tip ECs, where the levels were low (Fig. 7e). These findings are in line with the results showing that the presence of YB-1 granules in ECs at the vascular front is associated with low ZO-1 expression.

### ZO-1 deficiency in ECs of mice increases YB-1-positive granule formation and inhibits retinal angiogenesis

To determine the importance of endothelial ZO-1 in retinal angiogenesis in mice, we generated an EC-specific and tamoxifen (TAM)-inducible ZO-1 knockout mouse (ZO-1^EC-KO;Pdgfb-iCreER;Tjp1^fl/fl). ZO-1 deletion in ECs of the developing retinal vasculature of newborn ZO-1^EC-KO mice was achieved by TAM treatment at P3, and the vasculature of the retinas was analyzed at P7. TAM-treated Pdgfb-iCreER (Cre+) and Tjp1^fl/fl mice (CTL) were used as controls. TAM treatment induced a

marked reduction in ZO-1 protein levels in ECs isolated from the lungs of ZO-1^EC-KO (Supplemental Fig. S5b) and in ECs of retinas from ZO-1^EC-KO mice compared to CTL mice (Fig. 8a). Analysis of the retinal vasculature at P7 revealed that the deletion of ZO-1 in ECs during retinal vascular development inhibited the growth of the vascular plexus (Fig. 8b, c). No difference was observed in the retinal vasculatures of Cre+ and CTL mice (Fig. 8c). Detailed analyses of the remodeling plexus revealed increases in both the density and disorganization (lacunarity) of the vascular pattern of ZO-1^EC-KO retinas (Fig. 8d, e). A decreased number of endpoints was also observed at the vascular front (Fig. 8b, f). Tip ECs in the retinas of ZO-1^EC-KO mice had shorter and fewer filopodia compared to controls (Fig. 8g, h, i). Furthermore, the filopodia of ZO-1^EC-KO mice were less oriented in the direction of the tip ECs, suggesting defects in cell polarization (Fig. 8g, j), which could negatively affect the well-organized spatial and temporal development of the retinal vasculature. Finally, increased proliferation of ECs located in the remodeling plexus of the retinas of ZO-1^EC-KO mice was observed by phospho-histone H3 staining and is likely to be responsible for the increase in vascular density. This increased proliferation was not observed in retinal ECs present at the vascular front (Fig. 8k). Notably, ECs in the plexus and at the front of the developing retinal

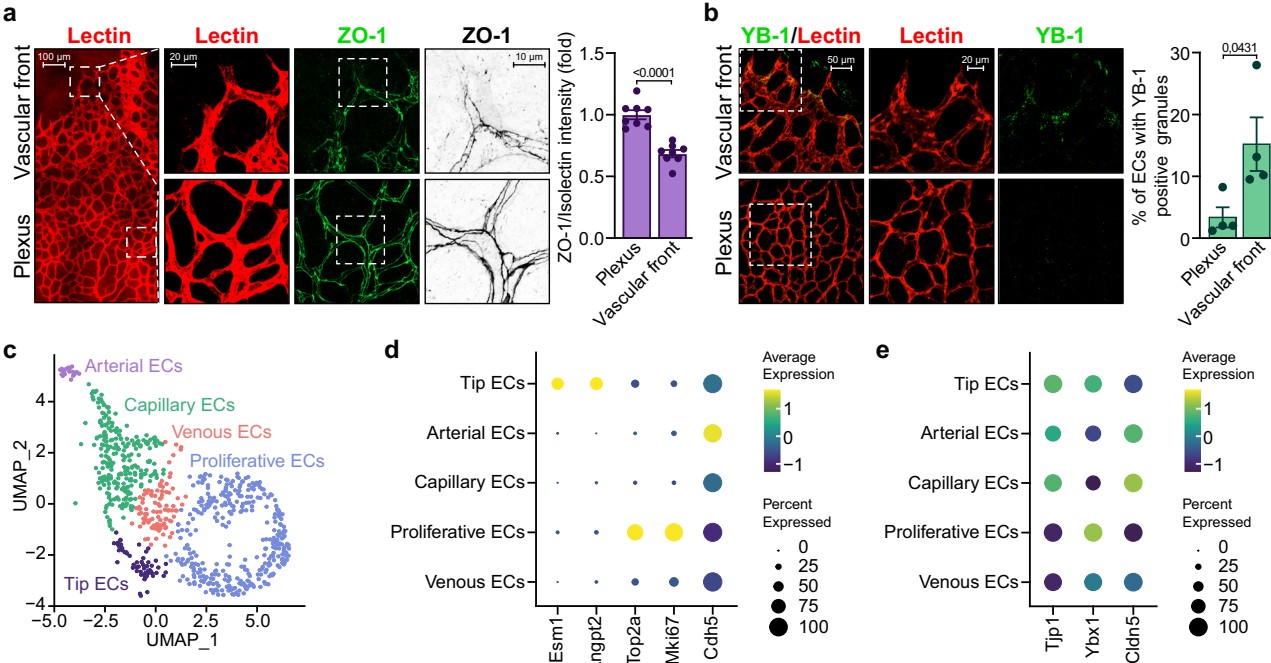

**Fig. 7 | Decreased ZO-1 expression and increased YB-1-positive granules in ECs at the vascular front of developing retinal blood vessels. a** Decreased ZO-1 levels in ECs at the vascular front of the developing retinal vasculature of mice. Immunofluorescence analysis of ZO-1 levels in ECs of the vascular front (edge) and plexus (central) portions of the developing retinal vasculature (P7) of mice. Vascular ECs of the retinas were stained with isolectin B4 (red) and with antibodies against ZO-1 (green). For clarity, right panels show the ZO-1 staining in black and white with intensities inverted. The histogram shows the quantification of images shown in (**a**). The results are presented as a ratio of the ZO-1/isolectin B4 intensity measured in n = 8 retinas (plexus or vascular front) from four mice. **b**, Increased YB-1-positive granules in ECs of the vascular front portion of the developing retinal vasculature. YB-1-positive granules were stained by immunofluorescence in ECs at the vascular front and in the plexus portion of the developing retinal vasculature (P7) of mice

(green). ECs of the retinas were stained with isolectin B4 (red). The histogram presents the quantification of YB-1-positive granules in retinal ECs shown in (**b**). The percentage of isolectin B4-stained cells with YB-1 granules was calculated at the vascular front and in the plexus of the retinal vasculature (n = 4 retinas; one retina per mouse was used). **c** Analysis of Single-cell RNA-sequencing data of ECs from mouse retinas at P6. Uniform Manifold Approximation and Projection (UMAP) plot of EC subclusters (Tip ECs, Arterial ECs, Capillaries, Proliferative ECs, Venous ECs) from P6 retinas generated in ref. 56. **d** Dot plot of expression level and frequency among EC subclusters of marker genes (*Esm1*, *Angpt2*; Tip EC markers) (*Top2a*, *Mki67*; Proliferative EC markers) (*Cdh5*, EC marker) and of (**e**) selected genes *Tjp1* (ZO-1), *Ybx1* (YB-1) and *Cldn5*. (**a**, **b**) Groups were compared by unpaired two-tailed Student's *t* test. Data are presented as mean values ± SEM. Source data are provided in the Source Data file.

vasculature of ZO-1[EC-KO] mice had significantly more YB-1- and G3BP1-positive granules (G3BP1 is not shown) than ECs in the retinas of control mice (Fig. 8l). These results suggest that by disturbing the balance of ZO-1 in ZO-1[EC-KO] mice, the angiogenic program is altered, thereby resulting in increased YB-1-positive granules and enhanced EC proliferation in the remodeling plexus. Overall, these results corroborate the model whereby decreased ZO-1 expression in ECs promotes proliferation, SG formation, and cytoprotection during angiogenesis. These effects are, at least in part, dependent on YB-1.

## Discussion

The cell junction protein ZO-1 links transmembrane proteins with cortical actin cytoskeletal proteins and is thus instrumental in assembling cell junctions[7]. Herein, we found that, in addition to its roles at cell junctions and its YB-3-dependent function in transcription, ZO-1 has a major effect on the fate of ECs under stress and during angiogenesis. We demonstrated that ZO-1 is a negative regulator of SG formation through its association with YB-1 and that ZO-1/YB-1 interactions endow cytoprotection to ECs against stress. Moreover, we confirmed that ZO-1 deficiency in ECs of developing vessels in the mouse retina results in an increased number of SGs and disrupts retinal angiogenesis.

By identifying the VEGF-regulated ZO-1 interactome in ECs, we found that ZO-1 interacts with RBPs and, in particular, with proteins that are known to be components of SGs. This suggests that ZO-1 may act as a regulator of SG assembly. Downregulation of ZO-1 led to

increased SG formation following stress and exposure of ECs to stress decreased ZO-1 levels. Furthermore, ECs with reduced ZO-1 expression were more resistant to arsenite-induced cell death. This suggests that ECs lower ZO-1 expression to promote SG formation and protect themselves from stress. It is known that VEGF treatment of ECs lowers ZO-1 expression levels[52,57] (Supplemental Fig. S3e). However, we found that VEGF stimulation did not induce SG formation, indicating that the anti-apoptotic signaling mechanisms activated by VEGF are different from the ones activated by ZO-1 signaling to SGs in response to stress. It is well-established that the phosphoinositide 3-kinase/protein kinase B (PI3K/Akt) pathway is primarily responsible for the pro-survival and the anti-apoptotic effects of VEGF on ECs[58,59]. Moreover, both VEGF and ZO-1 downregulation increase EC proliferation. Interestingly, it was proposed that an increase in the rate of cell proliferation contributes to SG assembly[60]. Hence, it is tempting to suggest that, in addition to differing for the regulation of SG formation, the signaling mechanisms employed by ZO-1 and VEGF may also differ in regulating EC proliferation. We also confirmed that actively migrating ECs at the leading edge in wound closure assays expressed less ZO-1 and exhibited more SGs than cells located at the back. This is consistent with our in vivo results, which revealed that the numbers of SG-like structures (YB-1- and G3BP1-positive granules) are greater in ECs located at the front of the developing retinal vasculature, where low levels of ZO-1 protein expression are observed. Furthermore, scRNA-seq analyses of ECs from the retinas of mice at P6 showed that the mRNA levels of ZO-1 and YB-1 were inversely correlated in EC subtypes[56]. Proliferative ECs

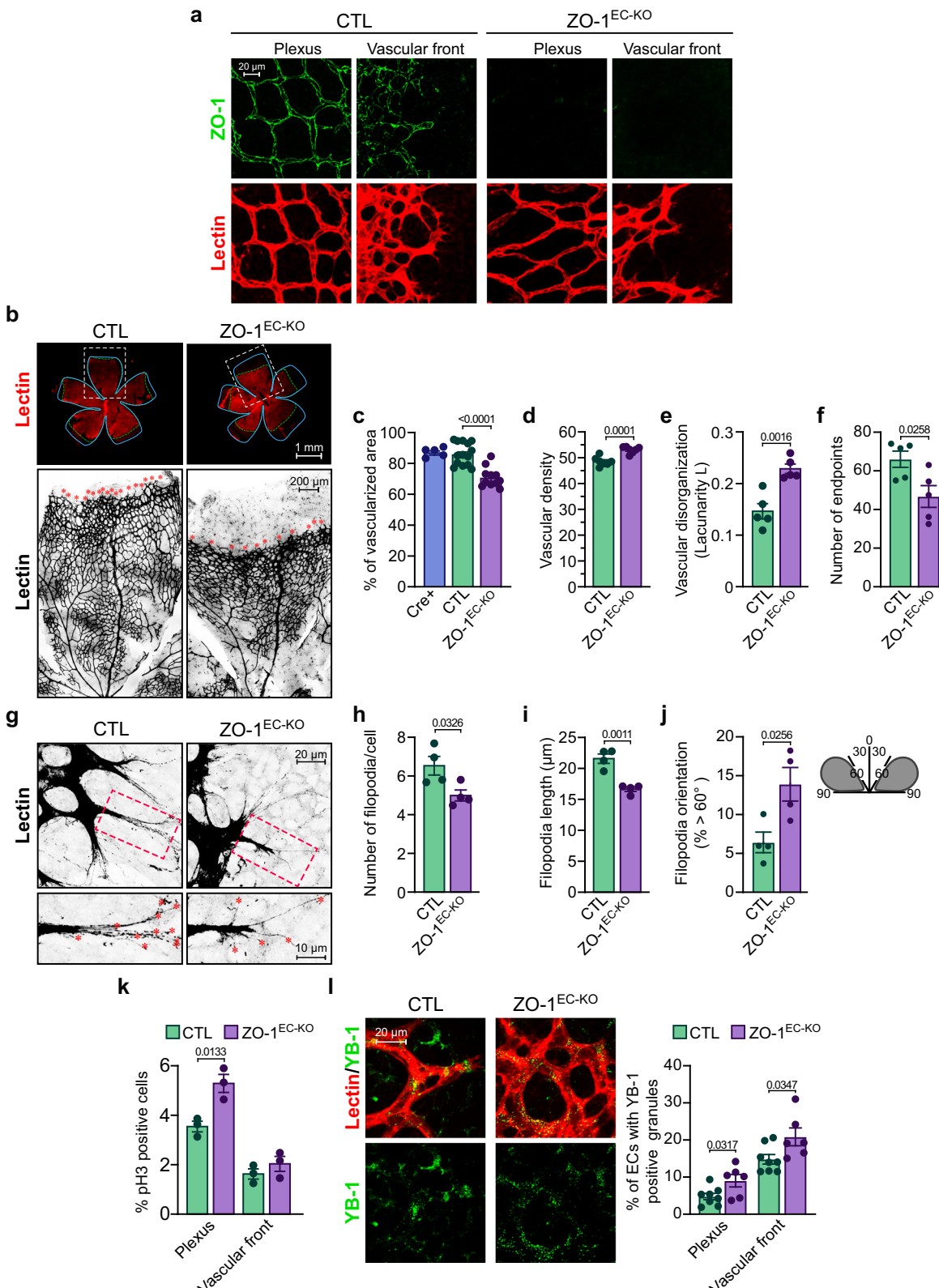

displayed low ZO-1 mRNA expression but high YB-1 expression, whereas arterial ECs and capillary ECs showed the opposite expression pattern. These observations are consistent with past findings on ZO-1, revealing that proliferative ECs have low ZO-1 but high YB-1 levels[11,61,62]. In agreement with the increases in SGs observed at the vascular front, we found that YB-1 mRNA expression levels are elevated in tip ECs. The scRNA-seq data show that tip ECs have elevated levels of ZO-1 mRNA.

However, the low correlation between the expression of *Tjp1* and *Ybx1* within each tip EC is in line with the protein expression patterns of ZO-1 and YB-1 observed at the vascular front (Fig. 7a). Thus, the protein and mRNA levels of ZO-1 may be differentially regulated in tip ECs.

The induction of SG formation and the cytoprotective effects mediated by ZO-1 downregulation were dependent on the expression of YB-1, an RBP involved in SG assembly, which we found to be a

**Fig. 8 | Deletion of ZO-1 in ECs of newborn mice increases YB-1 granules formation and induces defective retinal angiogenesis. a** ZO-1 is deleted in retinal ECs of ZO-1$^{EC-KO}$ mice at P7. Tamoxifen (TAM)-inducible and EC-specific ZO-1 knockout mice (ZO-1$^{EC-KO}$, Pdgfb-iCreER;Tjp1$^{fl/fl}$) and control mice (CTL, Tjp1$^{fl/fl}$) were treated with TAM at post-partum (P) day 3. The vasculature of the retinas was analyzed at P7. ZO-1 expression in ECs of the developing retinal vasculature was examined by immunofluorescence in ZO-1$^{EC-KO}$ and CTL mice. ZO-1 staining is shown in green and ECs were stained with isolectin B4 (red). **b** Reduced vascular area in retinas of ZO-1$^{EC-KO}$ at P7. The vascularized area of the retina is determined by isolectin B4 staining (red). **c** Quantification of the percentage of vascularized area in retinas from ZO-1$^{EC-KO}$ (n = 11), CTL (n = 14) and TAM-treated Pdfgb-iCreER (Cre+; n = 5) mice was determined by the ratio of the vascular area (dashed green line) to the total area of the retina (cyan line) shown in (**b**). **d** Increased vascular density in the ZO-1$^{EC-KO}$ mice. Quantification of vascular density in ZO-1$^{EC-KO}$ (n = 7) and CTL (n = 7) retinas was determined by measuring the area of isolectin B4 staining over the total vascularized area. **e** Increased disorganization of the retinal vascular patterning in ZO-1$^{EC-KO}$. Organization of the vascular pattern of retinas from ZO-1$^{EC-KO}$

(n = 5) and CTL (n = 5) mice was analyzed by determining the index of lacunarity Λ using the AngioTool software. Higher lacunarity is indicative of lower pattern organization. **f** Decreased number of retinal vessel endpoints in ZO-1$^{EC-KO}$ mice. Endpoints at the vascular front were counted in isolectin B4 stained CTL (n = 5) and ZO-1$^{EC-KO}$ (n = 5) retinas. Vessel endpoints are marked by a red asterisk in (**b**). **g, h, i** Altered filopodia in Tip ECs of ZO-1$^{EC-KO}$ mice. Filopodia on Tip ECs were counted (**h**) and measured (**i**) in isolectin B4-stained CTL (n = 4) and ZO-1$^{EC-KO}$ (n = 4) retinas. **j** Orientation relative to Tip ECs were counted in isolectin B4 stained CTL (n = 4) and ZO-1$^{EC-KO}$ (n = 4) retinas. **k** Increased proliferation of ECs in the vascular plexus in retinas of ZO-1$^{EC-KO}$ mice. EC proliferation was measured by phospho-histone-H3 staining in isolectin B4-stained ZO-1$^{EC-KO}$ (n = 3) and CTL (n = 3) retinas. **l** Increased YB-1-positive granules in ECs of the retinal vasculature of ZO-1$^{EC-KO}$ mice. Percentage of ECs with YB-1-positive granules were counted in isolectin B4-stained CTL (n = 8) and ZO-1$^{EC-KO}$ (n = 6) retinas. (**c–f, h–l**) Groups were compared by unpaired two-tailed Student's *t* test. One retina per mouse was used throughout. Data are presented as mean values ± SEM. Source data are provided in the Source Data file.

hitherto undocumented ZO-1-associated protein[40–42]. Herein, we demonstrated in vivo that inducible ZO-1 deficiency results in increased SG formation in ECs of the developing retinal vasculature of newborn mice. In addition, the inducible ZO-1$^{EC-KO}$ mouse model generated helped us realize the importance of ZO-1 in angiogenesis. Deletion of ZO-1 in ECs during the growth of the retinal microvasculature resulted in increased EC proliferation, vascular density, and pattern disorganization but a decreased number of vascular endpoints and a reduced number, length, and directedness of filopodia on tip ECs. Moreover, it resulted in augmented SG formation. Efficacious angiogenesis in vivo requires the perfect coordination of several complex mechanisms that are finely regulated, such as proliferation, migration, cell communication, and the formation and breakdown of cell junctions[12,13]. ZO-1 is likely required to coordinate these mechanisms, which are modulated by VEGF and involved in the formation of a functional vasculature. Our detailed analyses of ZO-1$^{EC-KO}$ retinas indicate that the spatiotemporal regulation of ZO-1 levels in ECs is required for proper angiogenesis, and suggests that SG formation, which is actively controlled by the ZO-1/YB-1 interaction, participates in this process. While it is unclear whether stress conditions prevail in ECs during normal vascular development, this study reveals the presence of YB-1- and G3PB1-positive granules in ECs of angiogenic vessels. Interestingly, a recent study revealed that ECs located in the aortic arch of mice exhibit SGs, most probably because of their exposure to high oscillatory shear stress[63].

YB-1 is part of the Y-box family of proteins that also includes YB-2 and YB-3. YB-3 is a known ZO-1-associated protein (aka ZONAB), whose transcriptional activity is regulated by ZO-1 and involved in the contact-mediated inhibition of proliferation[25–27]. The identification of another Y-box protein associated with ZO-1 suggests that the cold shock domains present in the Y-box family of proteins may be involved in this association[54,64]. In addition, it is known that the SH3 domain of ZO-1 binds to YB-3[25]. We now show that the SH3 domain of ZO-1 is also likely involved in the interaction with YB-1 (Fig. 5d). In agreement with previous studies, our results show that YB-1 acts as a positive regulator of SG formation in ECs in part through the increase in G3BP1 expression (Fig. 6c)[40,43]. Our results also suggest that disruption of the ZO-1 and YB-1 association induces signals in ECs that promote the interaction between YB-1 and other constituents of SGs, namely G3BP1, which may contribute to SG formation (Fig. 5h). However, in cancer cells, YB-1 was found to be a negative regulator of SG assembly through its mRNA-unwinding activity[65,66]. To further minimize cellular damage during stress, most of the mRNA translation is attenuated in conjunction with SG formation[67,68]. Furthermore, our results reveal that in arsenite-treated cells, ZO-1 depletion also inhibits translation initiation, which was evidenced

by the elevated phosphorylation of eIF2α at serine 51 (Fig. 6d). However, some stress-responsive mRNAs containing upstream open-reading frames (uORFs), such as ATF4 mRNA, are translationally activated under stress conditions[68–70]. YB-1 has been implicated in the stress-specific translation of some mRNAs involved in the angiogenic process, including HIF1α and VEGF[64,71]. Hence, ZO-1, in association with YB-1 or other RBPs involved in translation, may be implicated in stress-triggered mechanisms of translational regulation. The identification of numerous RBPs in the ZO-1 interactome, with many of the associations regulated by VEGF, highlights the multifunctional role of ZO-1 in cells. It is possible that some of these RBPs also participate, in conjunction with YB-1, to the formation of SGs. In addition, RBPs may be involved in the liquid-liquid phase separation (LLPS) process leading to SG formation, and ZO-1 was implicated in LLPS during tight junction organization and mechanosensation to flow[34,72–75].

Herein, we provide evidence that SG formation in ECs is involved in angiogenesis, and demonstrate that the expression levels of ZO-1 regulate this process. ZO-1 levels in cells are known to be actively regulated and influence cellular proliferation and differentiation as well as the invasiveness potential of malignant cells[11,25,26]. The ZO-1 interactome revealed an association between ZO-1 and YB-1 that influences SG assembly. Upon stress, the association between ZO-1 and YB-1 is disrupted and ZO-1 levels are reduced, which in turn promotes SG formation in a YB-1-dependent manner, ensuring efficacious angiogenesis. Furthermore, the ZO-1$^{EC-KO}$ mouse line demonstrated the role of ZO-1 as an orchestrator of EC proliferation, migration, cell junction integrity, and now SG formation during angiogenesis. In summary, the interaction between ZO-1 and YB-1 regulates SG formation and reveals a role for ZO-1 in controlling the fate of ECs under stress, which contributes to the regulation of angiogenesis.

## Methods
### Cell culture and treatments
Bovine aortic endothelial cells (BAECs) obtained from VEC Technologies (Rensselaer, NY, USA) were cultured in Dulbecco Modified Eagle Medium (DMEM) (ThermoFisher Scientific, Waltham, MA, USA) supplemented with 10% fetal bovine serum (Cytiva-HyClone, Global Life Sciences Solutions Marlborough, MA, USA), 2 mM L-glutamine, 100 U/ml penicillin, and 100 μg/ml streptomycin (ThermoFisher Scientific, Waltham, MA, USA). Human umbilical vein endothelial cells (HUVECs) obtained from VEC Technologies (Rensselaer, NY, USA) were cultured in M199 media supplemented with 20% FBS, 2 mM L-glutamine, 100 U/ml penicillin, and 100 μg/ml streptomycin and Endothelial Cell Growth Supplement (ECGS, Corning Life Sciences, Corning, NY, USA). Human hepatoma cells (HepG2) were obtained from ATCC (Manassas, VA, USA) and cultured in DMEM supplemented with 10% fetal bovine

serum, 2 mM L-glutamine, 100 U/ml penicillin, and 100 μg/ml streptomycin. TeloHuman aortic endothelial cells (TeloHAEC) were obtained from ATCC and cultured in Vascular Basal Cell Medium (VBCM, ATCC) supplemented with Endothelial Cell Growth Kit-VEGF (ATCC) and 100 U/ml penicillin, and 100 μg/ml streptomycin.

Recombinant human VEGF-A165 was obtained from R&D System (Minneapolis, MI, USA). Sodium arsenite (NaAsO₂), hydrogen peroxide $H_2O_2$ and MG-132 were from Millipore-Sigma (Burlington, MA, USA) and Thapsigargin was from Alomone labs (Jerusalem, Israel). Before each treatment, BAECs and HUVECs were serum starved for 6 h followed by a treatment. The doses and the duration of treatments are specified in the figure legends.

## Cell migration and viability assays

Cell migration was measured in a wound healing assay. Briefly, BAECs plated in 6-well plates were transfected with siRNAs and allowed to reach 90% confluency over 48 h. Scratches on the monolayer were performed with a 200-μl pipette tip. Images of the wounded area were taken using a Zeiss Axio-Observer Z1 epifluorescent microscope (Zeiss Group, Oberkochen, Germany) (2.5×) at the beginning (t = 0) and after 16 h of migration. Wound closure was measured using the draw spline contour tool on Zen Blue.

Cell viability was determined by Trypan blue exclusion. Viable cells were manually counted using a hemacytometer or a Countess III cell counter (ThermoFisher Scientific, Waltham, MA, USA).

## Plasmids and transfections

ZO-1, YB-1, β-catenin and VE-cadherin small interfering RNA (siRNA) as well as non-silencing control siRNA were obtained from Horizon Discoveries (Cambridge, UK). siCT AUG AAC GUG AAU UGC UCA AUU, siVE-cadherin (bovine) ACA AAG AAC UGG ACA GAG AUU, siVE-cadherin (human) GCA CAU UGA UGA AGA GAA A, siZO-1 (bovine) GCA GAG AGG AAG AGA GAA UUU, siZO-1 #1 (human) UGG AAA UGA UGU UGG AAU A (unless specified, siZO-1 #1 is used in human cells throughout), siZO-1 #2 (human) GCA AAG ACA UUG AUA GAA AUU, siβ-catenin (human) CCA CUA AUG UCC AGC GUU U, siCldn5 (human) CCA CCG GCG ACU ACG ACA AUU, siYB-1 (bovine) CAG CAG AAC UAC CAG AAU A, siYB-1 (human) CGG CAA UGA AGA AGA UAA A. BAECs and HUVECs were transfected with plasmids or siRNAs using Lipofectamine 2000 (ThermoFisher Scientific, Waltham, MA, USA) according to manufacturer's instructions.

## Antibodies

The primary antibodies used for immunoblots (IB) or immunofluorescence (IF) experiments were: rabbit polyclonal Anti-ZO-1 (61-7300, 1:50 IF), mouse monoclonal anti-ZO-1 (33-9100, 1:1000 IB, 1:100 IF), rabbit polyclonal anti-ZONAB (40-2800, 1:250 IB), rabbit monoclonal anti-Claudin-5 (MA5-32614, 1:1000 IB) antibodies were purchased from ThermoFisher Scientific, USA. Mouse monoclonal anti-BrdU (5292, 1:100 IF), rabbit polyclonal anti-ZO-2 (2847, 1:1000 IB), rabbit monoclonal anti-MYC-Tag (2278S, 1:1000 IB), rabbit monoclonal anti-phospho-YB1 (Ser102) (2900S, 1:1000 IB), rabbit monoclonal anti-caspase-3 (9665T, 1:1000 IB), mouse monoclonal anti-β-Actin (3700S, 1:10 000 IB), rabbit polyclonal anti-YB1 (D299) (4202S, 1:25 IF), rabbit monoclonal anti-YB1 (D2A11) (9744S, 1:1000 IB) were purchased from New England Biolabs, Ipswich MA, USA. Mouse monoclonal anti-G3BP1 (611126, 1:1000 IB, 1:100 IF), mouse monoclonal anti-β-catenin (610153, 1:1000 IB), mouse monoclonal anti-γ-catenin (JUP) (610253, 1:1000 IB) were purchased from BD Biosciences, San Jose, CA, USA. Goat polyclonal anti-VE-cadherin (AF938, 1:1000 IB) was purchased from R&D Systems, Minneapolis, MN, USA. Mouse monoclonal anti-EMAP II/AIMP1 (sc-393228, 1:1000 IB), mouse monoclonal anti-FUS/TLS (sc-47711, 1:1000 IB), mouse monoclonal anti-Ribosomal Protein L23a (sc-517097, 1:1000 IB) were purchased from Santa Cruz Biotechnology, CA, USA.

## Immunoblotting and immunoprecipitation

Cells were solubilized with a lysis buffer containing 1% Nonidet P-40, 0.1% sodium dodecyl sulfate (SDS), 0.1% deoxycholic acid, 50 mM Tris (pH 7.4), 0.1 mM EGTA, 0.1 mM EDTA, 50 mM NaCl, 20 mM NaF, 1 mM $Na_4P_2O_7$ and 1 mM $Na_3VO_4$. Lysate was incubated for 30 min at 4 °C, centrifuged at 14,000 g for 10 min and boiled in SDS sample buffer. For the immunoprecipitation, BAECs were treated (with VEGF or arsenite) and were lysed in the lysis buffer with NaCl adjusted to 75 mM. The lysates were centrifuged at 20,000 g for 10 min and 1 mg of proteins was used for immunoprecipitation with 1 μg of antibody overnight. Immunoglobulin G (IgG) control IPs were used as a negative control. Immunocomplexes were incubated with protein A sepharose beads (Millipore-Sigma, Burlington, MA, USA) for 1 h, washed in lysis buffer and boiled in SDS sample buffer. The samples were separated by SDS-polyacrylamide gel electrophoresis, transferred onto a nitrocellulose membrane (Hybond-ECL, Cytiva Global Life Sciences Solutions Marlborough, MA, USA), and immunoblotted. Detection was performed using HRP-coupled antibodies from Jackson ImmunoResearch Laboratories: Peroxidase AffiniPure Goat Anti-Mouse IgG (H + L) (115-035-146, 1:5000), Peroxidase AffiniPure Donkey Anti-Rabbit IgG (H + L) (711-035-152, 1:5000), Peroxidase AffiniPure Donkey Anti-Goat IgG (H + L) (705-035-003, 1:5000) and an Image Quant LAS4000 chemiluminescence-based detection system (enhanced chemiluminescence) (Cytiva Global Life Sciences Solutions Marlborough, MA, USA). Uncropped versions of immunoblots are provided in Source Data file.

## Glutathione S-transferase pulldowns

GST-tagged ZO-1 fragments (residues 1-510 and residues 511-1100) cloned into the pGEX-KG vector were a kind gift from Dr. S. Meloche (IRIC, Montreal, Canada)[76]. GST fusion proteins were expressed in BL21 bacteria and purified by affinity chromatography on Glutathione-Agarose beads (Millipore-Sigma, G4510). Briefly, BL21 were sonicated and lysed in STE buffer (7.5 mM Tris-HCl pH 8, 150 mM NaCl, 3 mM EDTA, 5 mM MgCl₂, protease inhibitor cocktail) supplemented with 1% Triton X-100 for 30 min at 4 °C. Lysates were centrifuged for 10 min at 6500 g and supernatant was incubated with the Glutathione-Agarose beads for 2 h at 4 °C. The levels of recombinant proteins were evaluated using a Coomassie-stained SDS gels. Beads previously conjugated to recombinant proteins were used for the pull-down assays. BAECs overexpressing myc-YB1 were lysed in RIPA modified buffer (1% Triton X-100, 0.2% NP-40, 50 mM Tris-HCl pH 7.5, 110 mM NaCl, 1 mM EDTA, 1 mM EGTA, 20 mM sodium fluoride, 1 mM sodium pyrophosphate, 1 mM orthovanadate, and protease inhibitor cocktail) and 500 μg of proteins were incubated for 4 h at 4 °C with 30 μl of coated agarose beads (50% slurry) corresponding to ~50 pM of GST, GST-ZO-1 (aa. 1-510), GST-ZO-1 (aa. 511–1100). The beads were washed three times in washing buffer (50 mM Tris-Cl pH 7.4, 125 mM NaCl, 1 mM EDTA, and 1 mM EGTA). The beads were eluted by boiling in SDS-sample buffer and bound myc-YB-1 was revealed by immunoblotting.

## Immunofluorescence

BAECs and HUVECs were transfected and then cultured on 0.1% gelatin-coated coverslips. Cells were washed with cold PBS and fixed for 20 min in 4% paraformaldehyde (PFA) and permeabilized with 0.3% Triton X-100. Cells were rinsed with PBS and blocked with 1% BSA for 1 h at room temperature. After blocking, cells were incubated with primary antibodies G3BP1, ZO-1, YB-1 for 2 h at room temperature in 0.1% BSA in PBS. Bound primary antibodies were visualized after 1 h of incubation using Alexa Fluor 488-conjugated Goat anti-Rabbit IgG (H + L) (A-11008, 1:100), Alexa Fluor 488-conjugated Goat anti-Mouse IgG (H + L) (A-11001, 1:100), Alexa Fluor 488-conjugated F(ab')2-Goat anti-Mouse IgG (H + L) (A48286TR, 1:100), Alexa Fluor 488-conjugated Donkey anti-Goat IgG (H + L) (A11055, 1:100), Alexa Fluor 488 Donkey anti-Rabbit IgG (H + L) (A21206, 1:100), Alexa Fluor 568 Donkey anti-

Rabbit IgG (H + L) (A10042, 1:100), Alexa Fluor 568 Donkey anti-Mouse IgG (H + L) (A10037, 1:100), Alexa Fluor 647 Donkey anti-Rabbit IgG (H + L) (A31573, 1:100) (ThermoFisher Scientific, Waltham, MA, USA). Coverslips were mounted using Fluoromount (Millipore-Sigma, Burlington, MA, USA) and observed using an LSM800 Zeiss confocal laser-scanning microscope (Zeiss Group, Oberkochen, Germany). Samples were viewed with a 63×/1.5 zoom oil objective. Images were assembled via ImageJ and Photoshop CC (Adobe Systems). Colocalization was analyzed by determining the Pearson colocalization coefficient using the Zen software (Zeiss).

## BrdU immunostaining

BAECs were transfected and then cultured on 0.1% gelatin-coated coverslips for 24 h in 10% FBS. Cells were incubated with 0.03 mg/ml BrdU at 37 °C for 30 min and then fixed with 70% ethanol for 5 min. After three washes with PBS, cells were denatured with 1.5 m HCl for 30 min at room temperature and then incubated with 1% BSA to block nonspecific staining for 60 min. Cells were incubated with BrdU antibody overnight at 4 °C, washed three times with PBS and then cells were incubated with Alexa-Fluor 568-labeled goat anti-mouse for 1 h. DAPI was used to stain the nuclei.

## Single-cell RNA-sequencing

We reanalyzed our publicly available scRNA-sequencing data from P6 mouse retinas (GSE175895) as we previously described using tools from Seurat R-package[56,77]. Uniform manifold approximation and projection (UMAP) plots was used for the visualization of EC subtypes. Dotplot of normalized transcript abundances was used to characterize the expression of genes of interest across EC subtypes. EC clusters were identified by the presence of *Cldn5* and *Pecam1* expression. Subsequently, EC subtypes were identified as Arterial ECs ($Unc5b^+;Bmx^+$), Capillaries ($Mfsd2a^{high};Slc22a8^{high}$), Venous ECs ($Hmgn2^{high};Ptgis^{high}$), Tip ECs ($Kcne3^+;Angpt2^+$), and Proliferative ECs ($Mki67^+;Birc5^+$).

## Mice

All animal studies were approved by the Animal Care Committee of the University of Montreal in agreement with the guidelines established by the Canadian Council on Animal Care. Wildtype C57BL/6J mice were purchased from The Jackson Laboratory (Bar Harbor, ME, USA). Male and female mice were used throughout the study. All animals were housed under controlled conditions with an ambient temperature set at 22 °C and relative humidity ranging from 35% to 50% under a 12/12-h light/dark cycle with unrestricted access to food and water throughout the duration of the experiments. Since deletion of ZO-1 induces lethality during mouse embryonic development[6], we have generated a new mouse model that enables us to induce endothelial ZO-1 deletion in mice postnatally upon 4-hydroxytamoxifen (TAM) treatment. To generate these mice, Tjp1[tm2a](KOMP)Wtsi (MGI:98759) C57BL/6 N mice were bred with B6.Cg-Tg(ACTFLPe)9205Dym/J (MGI:2448985) to delete the En2SA-LacZ-neo cassette between the Frt sites, thereby creating the B6-Tjp1[tm2a] line. For experimental purpose, B6-Tjp1[tm2a] were crossed with Pdgfb-iCreER mice, which were generously provided by Dr. Marcus Fruttiger (University College London)[78], to generate Pdgfb-iCreER;Tjp1[fl/fl] mice. Pdgfb-iCreER;Tjp1[fl/fl] were viable and fertile with no obvious defects. CreER induction and ZO-1 deletion in endothelia of newborn mice was achieved by intraperitoneal injection of TAM (20 µg/mouse in 2 µL methanol; i.p.) at P3. Retinal vasculature was analyzed at P7.

## Retina Immunostaining

Dissection and whole mount staining of postnatal retinas of mice at stage P7 were performed as described previously[11]. Retinas were fixed for 2 h on ice in 4% PFA. Dissected retinas were blocked overnight in 1% BSA, 0.3% Triton X-100 in PBS. For isolectin B4 staining, retinas were equilibrated with Pblec buffer containing 1 mm $CaCl_2$, 1 mm $MgCl_2$, 1%

Triton X-100 in PBS (pH 6.8) and then stained with Rhodamine conjugated Lectin I (dilution 1:100) overnight at 4 °C. For G3BP1, YB-1 or ZO-1 staining, retinas were incubated with mouse anti-G3BP1 (dilution 1:100), rabbit anti-ZO-1 (dilution 1:100) and rabbit anti-YB-1 (dilution 1:50) in blocking buffer 2 h at 4 °C. After primary incubation, retinas were labeled with Alexa-Fluor 488-labeled goat anti-mouse or anti-rabbit (dilution 1:100). Stained retinas were flat mounted using Fluoromount G (Electron Microscopy Sciences, Hatfield, PA). Z-stack confocal imaging was performed on Zeiss LSM800 confocal laser-scanning microscope using a 63× oil objective and a 2.5× digital zoom. All quantifications were performed on z-stack confocal images. Images were analyzed using Fiji software (NIH, Bethesda, MD, USA) or the built-in tools in Zen software (Zeiss). For the evaluation of lacunarity (higher values for lacunarity dimension (Λ) indicate that the microvascular network has low pattern organization), images of the retina were saved in tiff format and analyzed using AngioTool software[79].

## IP-MS sample preparation

For MS experiments, BAECS cultured in 10 cm dishes were treated with VEGF (40 ng/ml) for 10 min. Cell treatment was done in three independent experiments, and then lysed in 1% Nonidet P-40, 0.1% sodium dodecyl sulfate (SDS), 0.1% deoxycholic acid, 50 mM Tris (pH 7.4), 75 mM NaCl, 0.1 mM EGTA, 0.1 mM EDTA, 20 mM NaF, 1 mM sodium $Na_4P_2O_7$ and 1 mM $Na_3VO_4$. Lysates were centrifuged at 20,000 $g$ for 10 min and 1 mg of proteins were then used for immunoprecipitation with 1 µg of ZO-1 antibody overnight. Immunoglobulin G (IgG) control IPs were used as a negative control. Immunocomplexes were incubated with protein A agarose beads for 1 h, washed three times with lysis buffer and then three times detergent-free lysis buffer. Then, immunoprecipitated proteins were eluted in Urea 8 M, 50 mM Tris (pH 7.4), and proteases and phosphatases inhibitors. Beads were incubated with 50 µl elution buffer at room temperature for 30 min with frequent agitation. Eluted proteins were reduced at 37 °C using dithiothreitol (DTT) for 1 h and alkylated by iodoacetamide for 60 min at room temperature in the dark. The mixture was digested using trypsin (ratio enzyme/total protein of 1:50) followed by incubation at 37 °C overnight. The tryptic digestion was quenched by adding 1% TFA (trifluoroacetic acid).

## LC-MS/MS analysis

Peptides were re-solubilized under agitation for 15 min in 21 µL of 1% ACN / 1% formic acid. The LC column was a PicoFrit fused silica capillary column (17 cm × 75 µm i.d; New Objective, Woburn, MA), self-packed with C-18 reverse-phase material (Jupiter 5 µm particles, 300 Å pore size; Phenomenex, Torrance, CA) using a high-pressure packing cell. This column was installed on the Easy-nLC II system (Proxeon Biosystems, Odense, Denmark) and coupled to the LTQ Orbitrap Velos (ThermoFisher Scientific, Bremen, Germany) equipped with a Proxeon nanoelectrospray Flex ion source. The buffers used for chromatography were 0.2% formic acid (buffer A) and 100% ACN / 0.2% formic acid (buffer B). Peptides were loaded on-column at a flow rate of 600 nL/min and eluted with a 2 slopes gradient at a flow rate of 250 nL/min. Solvent B first increased from 1 to 40% in 110 min and then from 40 to 80% B in 50 min.

The mass resolution for a full MS scan was set to 60,000 (at m/z 400) and lock masses were used to improve mass accuracy. Mass over charge ratio range was from 375 to 1800 for MS scanning with a target value of 1,000,000 charges and from ~1/3 of parent m/z ratio to 1800 for MS[n] scanning in the linear ion trap analyzer with a target value of 10,000 charges. The data-dependent scan events used a maximum ion fill time of 100 ms and target ions already selected for MS/MS were dynamically excluded for 30 s after two repeat counts. Nanospray and S-lens voltages were set to 1.5 kV and 50 V, respectively. The normalized collision energy used was 27 with an activation q of 0.25 and activation time of 10 ms. Capillary temperature was 250 °C.

## MS data processing

Raw mass spectrometry data were processed using the MaxQuant software (version 1.5.3.17). Database searching was performed using the Andromeda search engine (version 1.5.3.17) integrated into Max-Quant against the bovine UniProt database and against the human UniProt database. MaxQuant default parameters were used with the exception of minimum ratio count and LFQ minimum ratio count set to 1. Enzyme specificity was set to trypsin and up to two missed clea-vages was allowed. Cysteine carbamidomethylation (C) was set as fixed modification and oxidation (M) and phosphorylation (STY) were set as variable modification. The minimum required peptide length was 6 amino acids. Mass tolerances for precursor ions and fragment ions were set to 20 ppm and 0.5 Da, respectively. The "matching between runs" algorithm in MaxQuant was enabled. The false discovery rate (FDR) was estimated by searching against the databases with the reversed sequences. For protein and peptide identification, the max-imum FDR was set to 1%. Three independent biological replicates and two technical replicates were performed. Correlations between the biological replicates are shown in Supplemental Fig. S1b. For proteins quantification, LFQ intensity values from biological and technical replicates that represent protein abundance were used for statistical analysis. Protein identification in at least two biological replicates was required for further analysis. Also, at least two peptides of a protein must be identified to be considered as a potential ZO-1 interacting protein. LFQ intensities across different samples were first normalized according to the intensities of the bait protein ZO-1 in each sample in order to have equal amounts of ZO-1 in each replicate. Then, intensities of ZO-1 interactors across different samples were adjusted according to the normalized ZO-1 intensities in each sample. Normalized LFQ intensities were used for determination of specific protein–protein interactions.

Significant interactors were determined by performing a statis-tical analysis of the bait IPs of each condition versus IgG control IPs. Datasets were log2 transformed and using Perseus tools, we imputed normal distributed values for missing values using a normal distribu-tion with width of 0.3 and a downshift of the mean by 1.8 compared to distribution of all LFQ intensities. Then, we performed a student's $t$ test-based comparisons of bait IPs versus IgG control IPs to identify significant interactors with false discovery threshold set at 0.05. Permutation-based FDR method in Perseus was used to perform mul-tiple testing corrections. We calculated the average intensities of ZO-1 interacting proteins between the replicates and the treated/untreated ratio for each protein was determined. To determine the interactors that are modulated by each treatment, we compared statistically using a student's $t$ test the bait IPs of each condition versus control condi-tion. Only interactors with more than 2-fold change compared to control condition were considered as affected by VEGF treatment. Interactors that are statistically influenced by VEGF treatment with a p value < 0.05 were considered significantly modulated by VEGF. We used CRAPome Web interface (www.crapome.org) to confirm the specificity of interactions identified in our affinity purification experi-ments. From the CRAPome database, we selected agarose beads as affinity support for control experiments similar to our experimental condition. As shown in Supplementary Data 3, interactors were cross-referenced with this CRAPome list. Only 3 out of 125 ZO-1 interactors (TUBB, HSPB1 and SNRPD1) had an average spectral count above 15. This indicates the high specificity of all the interactions identified in our AP-MS. However, these three proteins were not removed from the ZO-1 interactome because their interaction with ZO-1 was significantly above our statistical threshold and influenced by VEGF treatment.

## Proteomics data analysis

Gene ontology annotations for biological processes were obtained from the Gene Ontology integrated in STRING database (version 11.0). Only GO annotations that were significantly enriched with a p-value of less than 0.05 were used in the analysis. To generate protein interac-tions network, STRING interactions database was used. The published or informatic-predicted interactions were first determined using standard STRING-defined confidence (medium confidence 0.4). Pro-tein enrichment in the protein interaction network was manually annotated based on the GO biological process enrichment. The obtained STRING network data were imported into the Cytoscape software for visualization[80]. For clustering analysis, The Markov Clus-ter Algorithm (MCL) algorithm with clusterMaker2 plugin in Cytoscape was used to identify functional protein clusters within the networks[81,82].

## Statistics and reproducibility

Statistical analyses were performed with Prism 5 software (GraphPad, San Diego, CA, USA). Data are presented as mean values ± SEM. Unless specified, otherwise all experiments were repeated at least three times with similar results.

## Reporting summary

Further information on research design is available in the Nature Portfolio Reporting Summary linked to this article.

## Data availability

The mass spectrometry proteomics data have been deposited to the ProteomeXchange Consortium via the PRIDE partner repository with the dataset identifier PXD029332[83]. The scRNA-sequencing data from P6 mouse retinas can be accessed as GSE175895. STRING database version 11 was used to create all networks containing 24'584'628 pro-teins from 5090 organisms with 3'123'056'667 interactions which can be accessed via https://version-11-0.string-db.org/. Source data are provided with this paper.

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

## Acknowledgements

This work was funded by operating grants from the Cancer Research Society of Canada (#25151 and #937718) and from the Canadian Institutes of Health Research (CIHR) (PJT – 142180 and PJT – 180376) to J.-P.G. Y.E.B. is supported by Fonds de Recherche du Québec—Santé (FRQS). L.H., A.D., I.T. and E.L. are supported by FRQS Junior 1, Junior 2, Senior and Merit Investigator awards, respectively.

## Author contributions

Y.E.B. and R.C. designed and performed the experiments, analyzed the data, prepared figures, and wrote the manuscript; J.C., V.G.L. G.C. and C.D. performed experiments, analyzed data and revised the manuscript; A.C., E.L., S.A., J.S.J., I.T., L.H., and A.D. provided key expertise and revised the manuscript; J.-P.G. obtained funding, designed, and supervised the experiments, analyzed the data, prepared figures, and wrote the manuscript.

## Competing interests

The authors declare no competing interests.
