## [Peer Review File · Nature Communications]

ZO-1 interacts with YB-1 in endothelial cells to regulate stress granule formation during angiogenesisReviewers' comments:

Reviewer #1 (Remarks to the Author):

The study by Bakkuori presents the ZO1 interactome which identifies ZO1 in a network of RNA-binding proteins implicated in stress granule (SG) formation and function. ZO1 expression correlates negatively with formation of SGs, which is observed also in sparse endothelial cells. Enhanced SG formation in ZO1-deficient endothelial cell cultures is accompanied by reduced accumulation of cleaved caspase 3 and enhanced migration in response to arsenate treatment. The negative regulatory effect of ZO1 on SG formation is dependent on Y-box binding protein-1 (YB-1). In the developing retina, ZO1 expression is lower in the growing front compared to the remodelling part of the vascular plexus. Depletion of ZO1 from endothelial cells results in delayed retina outgrowth paralleled by increased endothelial proliferation. This is an interesting and well-presented study with careful biochemical analyses and novel information. However, there are some inconsistencies between the in vitro and in vivo models, moreover, the different roles of acute and chronic SGs during physiological processes (retina vascular development) versus pathologies need to be made clearer.

1. Acute and chronic SGd differ in composition and effects on cell fate. Importantly, acute SGs have a pro-survival function while chronic SGs have a pro-death function. In the introduction, the authors describe how cellular stress is imposed by various pathologies and generation of ROS, oxidized LDL, hypoxia and turbulent blood flow (lines 74-87). The treatment of cells in vitro with arsenate is readily classified as toxic, acute stress. However, it is less obvious that growth of a vascular plexus represents an acute stress condition. The authors need to give a better background to acute and chronic SGs and describe what is known about cellular stress and SG formation during physiological processes. What is the role of ZO1 during vascular growth and remodelling in inflammatory processes?
2. The authors perform immunostaining for ZO1 and show lower expression after normalization to Isolectin intensity, in the vascular front (Fig 6c and d). The retinal vascular front is characterized by VEGF-driven proliferation and migration of endothelial cells. In vitro, however, VEGF does not seem to affect ZO1 levels (see Fig 1c). Is VEGF-induced endothelial migration and proliferation accompanied by changes in the expression levels of ZO1?
3. In order to get a more quantitative estimate of ZO1 expression in different vessel subtypes and in different conditions, it would be useful to interrogate published data bases on the expression levels of ZO1 in different vessel subtypes in regions in the eye. See for example Zarkada et al., Dev Cell 2021 PMID: 34273276.
4. In the absence of ZO1 expression, the authors describe reduced vascular outgrowth (Fig. 7) compared to the Cre-negative control mice. It is well established that Cre expression can be toxic which can be detrimental and stall vascular outgrowth. See Brash et al. 2020, PMID: 32635822. The authors are strongly recommended to use Cre+ wildtype mice as a control.

5. The in vitro and in vivo effects of deletion of ZO1 do not match, possibly due to the detrimental effects of Cre (see above). In vitro, stress-induced SG formation results in decreased migration which is rescued when cells are depleted for ZO1 (Figure 4f, g). In contrast, in vivo, retinal vessel outgrowth is delayed in the ZO1-deficient pups, which appears to be dependent on decreased migration as proliferation is actually increased (Fig 7). How do the authors explain these opposing effects of ZO1 depletion in vitro and in vivo? Moreover, Tornevaca et al. PMID: 25753039 show that wound healing is delayed in endothelial cells treated with ZO1 siRNA, which is in contrast to the data shown here where siZO1 enhances endothelial migration. This needs to be sorted out.

6. The authors show that endothelial proliferation is increased in the ZO1-deficient retina vasculature, judged from the number of pH3 positive cells compared to that for wildtype retina vessels (Fig 7g). Still the vascular density does not seem to be affected judged from the images in Fig 7b. Please explain.

7. In ZO1-deficient conditions, SG formation is increased, paralleled by cytoprotection. Can the authors discuss whether the cytoprotective effect of ZO1 deficiency could be dependent on transcription of YB1 target genes rather than being solely through binding of YB1 to GPBP1 and SG formation?

Minor

8. The authors study the developing retina. It would be preferable to use the well-established designations “vascular front” instead of “distal” or “extremities” of the vasculature and “remodelling plexus” instead of “median”.

Reviewer #2 (Remarks to the Author):

In this manuscript, the authors describe a new mechanism by which ZO-1, a membrane protein involved in the regulation of intercellular junctions, controls the assembly of stress granules through an interaction with YB-1, an mRNA-Binding proteins. In a previous report (Balda et al., JCB, 2003), an interaction between YB-3, another member of the Y-box family was already proposed. Starting with the identification of ZO-1 interactome, which reveals the preferential interaction of ZO-1 with proteins involved in RNA processing, the authors pay a particular attention to a putative role of ZO-1 in the regulation of stress granule assembly. In arsenite-stressed endothelial cells, ZO-1 is partly degraded and its interaction with YB-1 is decreased. A decreased interaction between YB-1 and ZO-1 then promotes stress granule assembly since the authors considered YB-1 to be a positive regulator of stress granules assembly. In the model proposed by the authors, ZO-1 “sequesters” YB-1 that would be subsequently released after stress to promote stress granule assembly. YB-1, after its release by ZO-1, can notably interact with G3BP-1, a protein known to promote SG assembly. Finally, the proposed model is put to the test in a more physiological system in endothelial cells of the retina.

The relationship between membrane proteins and RNA-processing proteins is very interesting topic. The results can reveal whether cells regulate their translational response to stress based on intercellular contacts. However, I do not think that the data presented in this manuscript supports the proposed model by which ZO1 would control stress granule assembly based on an interaction with YB-1.

Here are my specific concerns:

1) In the manuscript, when the expression of ZO-1 is decreased, SG assembly is promoted after arsenite stress in endothelial cells. However ZO-1 is known to limit cell proliferation based on contact inhibition between endothelial or epithelial cells (Balda, J cell biology, 2003). The formation of mRNA-rich stress granules follows the dissociation of polysomes due the inhibition of translation initiation. In proliferating cells, there are more polysomes whose dissociation after stress is more likely to form stress granules. Even if the same experiments were performed with VE-Cad and beta-catenin as controls (Fig. 2), the proliferation status or (and) mRNA translation should be measured under their experimental conditions (in addition at least two siRNAs have to be used to discard potential non specific effects). In the wound healing assays (Fig. 4), the author reports that the occurrence of SGs is higher in leading cells than in followers. After wound, the rate of proliferation increases which promotes stress granule assembly (as show previously in epithelial cells (Desforges, MBOC, 2013). ZO1 is known to control cell proliferation (Balda, J cell biology, 2003 and). Downregulation ZO1 may thus promote SG assembly by increasing the proliferation rate of followers. This hypothesis should be also considered.

2) Why the authors focus their attention on YB-1 is not very clear but it may be due to the previous study on the YB-3:ZO1 system (YB-3 is very similar to YB-1 and there should be a comment about whether the proposed interplay between YB-1 with ZO-1 works also for YB-3). The authors show that silencing YB-1 counteracts the effect of silencing ZO-1 on SG assembly (Fig. 5). However, YB-1 is associated to cancer and elevated YB-1 level promotes high cancer cell proliferation rates. Silencing YB-1 expression may thus prevent an increased proliferation rate which may occur after silencing ZO1 expression. Therefore, this experiment does not tell whether silencing YB-1 specifically prevents the increased level of stress granule assembly in siZO1-treated cells.

3) As suggested in the manuscript, ZO-1 may prevent YB-1 from interacting with G3BP-1. What are the expression levels of ZO1 in endothelial cells compared to those of YB-1, which is an abundant protein? . If the authors propose that ZO1 can sequester YB-1 (it is not clear to me), a significant fraction of YB-1 under normal condition should colocalize with ZO1 at cell:cell junction. However, YB-1 is diffuse in the cytoplasm under physiological condition (figure 5 (I did not see the localization of YB-1 with ZO1 at cell:cell interface under normal condition)). The level of sequestration of YB-1 by ZO-1 should be estimated by measuring the fraction of YB-1 in the cytoplasm and associated with membranes before and after treatment with 0.5 mM arsenite for 30 min.

An increase of YB-1 level was observed after arsenite treatment (fig.5a) but it is misleading because this results was obtained after cells were exposed to 50 μ M for 6 h (leaving times for changes in protein expression) while 0.5 mM arsenite for 30 min was used in stress granule assays.

4) In the proposed model, YB-1 interacts with ZO-1 under normal condition but to lesser extent after stress which is somehow logic as ZO1 is partly degraded (figure 3). It is important to show whether the decreased interactions depend on ZO1 degradation (by using MG132). In addition, the author indicates that ZO1 has been identified as an RNA-binding protein. They should also show whether the interactions between YB-1 and ZO1 is RNA dependent with RNase treatment and compare the results with other mRNA binding proteins as controls. Moreover, if there is a direct interaction, it would be interesting to identify which domains are necessary for this interaction to take place.

5) ZO1 is not found in stress granules even if pull down did show a partial interaction (even reduced) between YB-1 and ZO1 after arsenite stress. As YB-1 is associated to stress granules, we may expect to partly detect ZO1 in SGs. May be that, as indicated in the manuscript, YB-1 no longer interacts with ZO1 because of an interaction with G3BP-1 or the degradation of ZO-1. To my knowledge, there is no report on a possible direct interaction between YB-1 and G3BP-1 (mRNA dependent?). If yes, the author should indicate this point. The authors should also elaborate on this hypothesis. We do not know whether the reported interaction between YB-1 and G3BP-1, if confirmed, can promote or inhibit stress granule assembly.

6) In the introduction, YB-1 is presented as a positive regulator of stress granule assembly. While it has been proposed that YB-1 may promote stress granule assembly by increasing the expression of G3BP-1 (Somasekharan et al., *Journal of Cell Biology*, 2015), several genome-wide analyzes did not show that YB-1 is critical for stress granule assembly (Wheeler, *Nature Methods*, 2020 and Youn, *Molecular Cell*, 2018). YB-1 could even be a negative regulator of stress granule assembly (Ripin, *RNA*, 2022, Budkina, *Nucleic acids research*, 2021). Of course, the role of YB-1 in stress granules can be contextual and cell type dependent but these results should be mentioned in the text. In this manuscript, the authors show that the % of cells with stress granules increases after siYB-1 treatment but, as I already mentioned, the proliferation rate of the cells may have decreased. Besides, even if these results are controversial (see Lyons, *Nucleic acids research*, 2016), YB-1 may in specific cases increase the expression level of G3BP1 which may provide an explanation for a positive regulation of SGs by YB-1. It would be interesting for the authors to probe whether G3BP-1 expression decreases after silencing YB-1 in their endothelial cells.

In summary, the subject of this manuscript is very interesting. However, the proposed model needs to be supported by stronger experimental evidences.

Reviewer #3 (Remarks to the Author):

The work by Gratton and co-workers represents a comprehensive and detailed analysis of the influence of ZO1 protein levels on the formation of so-called stress granules in endothelial cells. In short, the authors show that down-regulation of ZO-1 enhances the formation of SGs under stress conditions (i.e. arsenite treatment), thereby protecting cells from cell death. This function is independent of other junctional components such as VE-cadherin or β -Catenin. At the molecular level, the authors show that formation of SGs require the protein YB-1 downstream of ZO-1 and show that stress leads to a dissociation of YB-1 from ZO-1.

Overall, the study is carefully executed and the data are of high quality. Because of the wealth of data, the arguments are not always easy to follow. A graphical abstract detailing the proposed mechanism of ZO-1 and YB-1 function would be helpful.

Major point:

YB-1 is proposed to be the key factor as YB-1 kd prevents SG formation downstream of ZO-1. This hypothesis should be tested by overexpression of YB-1 under normal, stress and ZO-1 overexpression conditions.

The authors propose two stress induced mechanisms. 1) downregulation of ZO-1 and 2) dissociation of YB-1. Is the dissociation of the proteins simply do the loss of the binding partner (ZO-1) or are these two distinct molecular mechanisms?

Along these lines:

Was YB-1 detected as a protein down-regulated upon VEGF-A treatment in the proteomic analysis? This is not mentioned in the text.

Is YB-1 localized to cell junctions in confluent cells?

Minor points:

Figure 2:

To show that ZO-1 regulates SG formation independent of junctions the authors analyze VE-cad and β -cat knockdowns. It would also be important to knock down tight junction proteins, e.g. Cldn5 or Esam, which may increase cytoplasmic ZO-1 levels.

In Figure 2f, the cell junction becomes merely visible. Is ZO-1 specifically removed from EC junctions. Does VE-cadherin still show normal distribution?

Figure 4:

The authors perform scratch-wound assays and suggest that down-regulation of ZO-1 during EC migration protects angiogenic ECs from stress. Down-regulation of junctional components is a prerequisite for angiogenic sprouting. In this context, the authors observe SGs only upon arsenite treatment. Therefore, the author's interpretation (lines 212f) appears unwarranted.

Figure 5:

In Figure 5a, the authors show that YB-1 levels increase upon 30'-minute arsenite treatment. This is surprisingly fast. How is this upregulation achieved? De novo protein synthesis?

Figure 7:

The EC specific knock-out of ZO-1 leads to strong defects in the retinal vasculature. However, the authors do not present evidence that this is due to misregulation of SGs.

Reviewer #4 (Remarks to the Author):

Reviewer's comments:

The study described in the manuscript by Bakkouri et al showed that interaction of tight junction protein ZO-1 with YB-1 controls stress granule formation of endothelial cells (ECs) during angiogenesis. First, the authors performed classical proteomics-based AP-MS approach to identify VEGF-induced interactors of ZO-1 in ECs. Through standard bioinformatics analysis of ZO-1 interactors with or without VEGF treatment, they showed ZO-1 interacts with YB-1 and FUS, which are associated with stress granule formation, and establish a novel link between ZO-1 and stress granules. Authors further showed that association of ECs ZO-1 with YB-1 negatively regulates the formation of stress granules through series of experiments. Furthermore, using EC-specific TAM-inducible ZO-1 knockout mice, they showed increased formation of stress granules in ZO-1 deficient ECs in the retina of new-born mice and retinal angiogenesis defects in the ZO-1 knockout mice.

Overall, the manuscript is well-written, and the obtained results are presented logically. While the experimental designs are acceptable, they are based on routine and conventional strategies (such as AP-MS). The major conclusions are somewhat sufficiently supported by experimental evidence. Although the study adds new knowledge to the field with regards to the new role of ZO-1 in stress granule formulation, conceptual advance, and significance of the work in the broader context is unclear. I have the following major concerns regarding the methodology employed, results obtained and possible additional experiments that should be addressed before recommending this work for publication in Nature Communications.

1. The authors carry out AP-MS to capture interacting partners. This method is known to generate a lot of non-specific interactors particularly in the label-free format it is being carried out. For identifying specific interactors with high confidence especially under perturbation such as VEGF treatment, a proximity biotinylation method with recent advancements (e.g., Santos-Barriopedro et al, Nat Commun 2021) is recommended. This is particularly useful to identify the dynamic interactome of a junctional protein such as ZO-1 whose immediate interactome could change depending on the cellular localization.
2. BAEC of bovine origin was chosen for interactome study while HUMVE was used for other experiments. Authors should explain this.
3. The authors have picked up a few interactors for subsequent validation. What is the rationale for selecting such candidates (Fig. 1c)? Are these proteins the top hits (statistically significant or functional relevance or just cherry picking)? Since YB-1 and G3BP1 were picked up for subsequent studies why they have not been included in the Western blot validation shown in Fig. 1c.? These two proteins should be included in this Western validation. And also significantly enriched proteins including YB-1, G3BP1 and other relevant proteins such as stress granule proteins should be labelled in Fig. 1a.
4. It is not clear why authors have selected VEGF treatment for the first pull-down experiment. Also, the authors used VEGF treatment for 10 minutes. Does 10 minutes of VEGF treatment alter localization of ZO-1? How did authors arrive at this stimulation time? There seems to be no follow up with regards to the relevance of VEGF in the later sections of the manuscript. The reasons for all these should clearly be stated.
5. It is also recommended to perform this interactome analysis under arsenite treatment, which is directly related to stress granule formation (or stress tolerance) as authors mainly used arsenite treatment for cell-line based validation with regards to stress granule assembly.
6. The ZO family of proteins include ZO-1, ZO-2, and ZO-3. Do the other proteins in the family have roles similar to ZO-1 in influencing stress granule formation? Is there any compensatory mechanism during loss of ZO-1? Authors can comment on this in the discussion.
7. For shortlisting the interacting proteins, the authors have performed statistical analysis. Were any attempts made to filter off non-specific interactors or false positive interactors? It is useful to check against the CRAPome database (<https://reprint-apms.org/?q=chooseworkflow>) to make sure the interactors reported are indeed confident.
8. The authors report that ZO-1 shares interactions with RBPs involved in stress granule formation. However, later they showed that ZO-1 levels negatively regulate stress granule formation and loss of ZO-

1 triggers stress granule assembly. This seems a bit confusing as to the nature of interaction between ZO-1 and the RBPs and the relevance of such interaction in a general context.

9. While there is much discussion of ZO-1 and YB-1, the co-localization of ZO-1 and YB-1 in-vivo is not shown (not only based on Co-IP). Authors need to show co-localization of ZO-1 and YB-1 to confirm this interaction. This is especially important when ZO-1 is primarily a junctional protein and YB-1 is a cytoplasmic or nuclear.

10. It will be useful to assess if any domains within ZO-1 is required for binding to YB-1? Authors can express ZO-1 without such domain and check the effect of stress granule assembly/disassembly?

11. Since ZO-1 is a tight junction protein, authors should elaborate the relationship between tight junction and stress granule formation if any. Is there any cross talk?

12. The relevance of ZO-1 influenced stress granule formation is unclear. Authors should emphasize and elaborate the importance of stress granule formation and how this can be significant in the context of disease or processes such as angiogenesis.

13. Line 219-221 “consistent with its role in stress granule formation, YB-1 protein levels were increased in ECs treated with arsenite while the levels of ZONAB and of ZO-1 were decreased (Fig. 5a)”. YB-1 level does not seem to be increased in the presence of Arsenite (Fig.5b and c input). How was input normalized? Loading control should be included to avoid confusion.

14. In the experiments discussing on migratory potential, it seems that decreased ZO-1 while promoting stress granule formation also increases migration during arsenite treatment. The authors concluded that section with “downregulation of ZO-1 promotes stress granule formation which in turn protects ECs from exogenous stresses that may occur during angiogenesis”. How is this a protective effect when decreased ZO-1 amplifies the migratory potential during angiogenesis? Isn't that detrimental?

15. Was there any stress applied to ZO-1 deficient mice that triggered stress granule formation or is it only triggered by loss of ZO-1. If stress was not applied, how does it align with previous in vitro results?

16. No follow up was made on post stress recovery with regards to stress granule levels and ZO-1 expression being bounced back. Rescue experiments should be performed to assess the specific relationship of ZO-1 with stress granule assembly. Also, any experiments to overexpress ZO to correlate to reduce stress granule or disassembly of stress granules is needed to establish a direct role.

17. The relevance of angiogenesis to this whole study is unclear. Angiogenesis is described once in a while and the relevance of VEGF in the interactome performed is unclear as there is no follow up. Any other process other than angiogenesis is relevant?

18. Signalling mechanism on how stress granule formation is triggered by ZO-1 deficiency is missing. The assembly of stress granule is largely driven by liquid-liquid phase separation (LLPS). Any protein modifications on stress granule proteins in ZO-1 deficient ECs that could serve to either disrupt critical multivalent interactions or provide new contact surfaces that drive LLPS?

Minor

1. Line 75: Include citation(s)
2. VE-cad western-blot for figure 3a showed one band, figure 3b showed 2 bands. Authors need to comment on this.
3. Line 355. Include complete company information
4. Line 359, 368. Company location is not listed. It should be consistent throughout the manuscript.
5. In material and method. Mass Spectrometry acquisition parameters should be listed
6. Line 506. p-value (typo)
7. Line 512, Include citations for Cytoscape and MCL algorithm

Reviewer #1 (Remarks to the Author):

The study by Bakkouri presents the ZO1 interactome which identifies ZO1 in a network of RNA-binding proteins implicated in stress granule (SG) formation and function. ZO1 expression correlates negatively with formation of SGs, which is observed also in sparse endothelial cells. Enhanced SG formation in ZO1-deficient endothelial cell cultures is accompanied by reduced accumulation of cleaved caspase 3 and enhanced migration in response to arsenate treatment. The negative regulatory effect of ZO1 on SG formation is dependent on Y-box binding protein-1 (YB-1). In the developing retina, ZO1 expression is lower in the growing front compared to the remodeling part of the vascular plexus. Depletion of ZO1 from endothelial cells results in delayed retina outgrowth paralleled by increased endothelial proliferation. This is an interesting and well-presented study with careful biochemical analyses and novel information. However, there are some inconsistencies between the in vitro and in vivo models, moreover, the different roles of acute and chronic SGs during physiological processes (retina vascular development) versus pathologies need to be made clearer.

We thank reviewer #1 for finding our study interesting and acknowledging that it provides novel information. In this extensively revised manuscript, we have resolved the apparent inconsistencies between the *in vitro* and *in vivo* models. In particular, we have better defined the implication of ZO-1 and stress granules in vascular development.

1. Acute and chronic SGs differ in composition and effects on cell fate. Importantly, acute SGs have a pro-survival function while chronic SGs have a pro-death function. In the introduction, the authors describe how cellular stress is imposed by various pathologies and generation of ROS, oxidized LDL, hypoxia and turbulent blood flow (lines 74-87). The treatment of cells in vitro with arsenate is readily classified as toxic, acute stress. However, it is less obvious that growth of a vascular plexus represents an acute stress condition. The authors need to give a better background to acute and chronic SGs and describe what is known about cellular stress and SG formation during physiological processes.

As per the reviewer's suggestions, we have added in the introduction some information on the physiological vs pathological roles of SGs and of other RNP granules (p. 4). Also, we agree with the reviewer that the roles of SGs in the developing plexus is less obvious than in ECs acutely treated in vitro with arsenite. However, it is clear that in both cases decreased ZO-1 levels in endothelial cells (ECs) (retinal and in vitro) induce YB-1 and G3BP1 positive granules. We cannot completely define the nature of these YB-1 positive RNP granules present in ECs at the extremities of growing retinal vessels in mice and if these are actually caused in response to stress during vascular development; however, to our knowledge, this is the first demonstration of the presence of RNP granules in ECs during angiogenesis. It is likely that these YB-1 positive granules have physiological roles during normal development of the vasculature and that by disturbing the balance of ZO-1 in EC-KO mice, the angiogenic program is altered thereby resulting in increased YB-1 positive granules and augmented EC proliferation in the remodeling plexus. This in turn leads to an increase in vascular density and disorganization, and ultimately a delay in the outgrowth of the vasculature. These points are now better explained in the revised discussion.

2. The authors perform immunostaining for ZO1 and show lower expression after normalization to Isolectin intensity, in the vascular front (Fig 6c and d). The retinal vascular front is characterized by VEGF-driven proliferation and migration of endothelial cells. *In vitro*, however, VEGF does not seem to affect ZO1 levels (see Fig 1c). Is VEGF-induced endothelial migration and proliferation accompanied by changes in the expression levels of ZO1?

We apologize for the confusion. As indicated in the Figure 1c legend, we monitored the effects of a short-term VEGF stimulation (40 ng/mL; 10 min) on the association of ZO-1 with interacting proteins. This condition was used for determining the VEGF-regulated ZO-1 interactome (Figure 1b). This short-term VEGF stimulation does not affect the expression levels of ZO-1, as monitored by western blotting (Figure 1c). It is already known that VEGF exposure induces downregulation of ZO-1 in ECs (Ghassemifar et al. Cell and Tissue Research 2006, PMID: 16163490 ; Wang et al. Mol Neurobiol. 2016, PMID: 26530694). We have now added Figure S2f which shows that long-term VEGF stimulation of ECs in culture decreases ZO-1 levels. Indeed, it takes 2 hours of treatment with these concentrations of VEGF (40ng/ml) before ZO-1 protein levels are significantly decreased (Fig. S2f).

The reviewer is correct in stating the important roles of VEGF at the retinal vascular front. As suggested by reviewers and to better correlate our *in vivo* and *in vitro* models, we performed VEGF-induced migration assays and monitored the impact of VEGF on ZO-1 expression level and SG formation in migrating ECs. Similar to the retinal vascular front, ECs at the leading edge of the migration front *in vitro* have reduced ZO-1 levels than following ECs. We now show that VEGF stimulation during EC migration resulted in a reduction of ZO-1 levels in follower cells and further decreased ZO-1 levels in leading control-transfected (Fig. 4e).

3. In order to get a more quantitative estimate of ZO1 expression in different vessel subtypes and in different conditions, it would be useful to interrogate published data bases on the expression levels of ZO1 in different vessel subtypes in regions in the eye. See for example Zarkada et al., Dev Cell 2021 PMID: 34273276.

Thank you for this suggestion. As recommended by the reviewer, we established a collaboration with the Dubrac laboratory to obtain and analyze their scRNA-seq data of retinal endothelial cells published in Zarkada et al. (Dev Cell 2021). We used these data to generate the new Figures 7c - e that now present mRNA expression of ZO-1 (*Tjp1*) and YB-1 (*Ybx1*) in Tip ECs, Arterial ECs, Capillaries, Proliferative ECs, and Venous ECs. Gene expression analysis revealed that *Tjp1* and *Ybx1* mRNAs are differentially expressed in EC subtypes (Fig. 7e). In plexus-forming ECs, namely Arterial ECs and Capillaries, expression levels of *Tjp1* were elevated while levels of *Ybx1* were low (Fig. 7e). However, Venous ECs showed higher expression of *Ybx1* compared to *Tjp1*. Pearson's correlation analysis indicated a strong to modest association between the expression of *Tjp1* and *Ybx1* in Arterial ECs, Venous ECs and Capillaries with a Pearson's correlation score of 0.59, 0.24 and 0.23, respectively. In line with previous findings related to ZO-1, proliferative ECs had low *Tjp1* expression but high *Ybx1* levels, which is associated with a low Pearson's correlation score of 0.06. In contrast to ZO-1 protein levels seen at the vascular front, mRNA levels were elevated in Tip ECs. However, *Ybx1* mRNA levels were also elevated in agreement with increased YB-1-positive granules in ECs at the vascular front. However, we found no linear correlation in the expression levels of *Tjp1* and *Ybx1* in Tip ECs (Pearson's correlation score of 0.05). This suggests that within each individual Tip EC the expression of *Tjp1* and *Ybx1* does not correlate,

which is similar to the protein expression patterns of ZO-1 and YB-1 observed at the vascular front (Fig. 7a, b). Finally, mRNA levels of the integral membrane tight junction protein Claudin 5 (*Cldn5*) followed those of *Tjp1* except in Tip ECs where levels were low (Fig. 7e).

4. In the absence of ZO1 expression, the authors describe reduced vascular outgrowth (Fig. 7) compared to the Cre-negative control mice. It is well established that Cre expression can be toxic which can be detrimental and stall vascular outgrowth. See Brash et al. 2020, PMID: 32635822. The authors are strongly recommended to use Cre+ wildtype mice as a control.

Thank you for suggesting this additional control. We agree that high Cre expression in ECs may have toxic effects, which could hinder vascular outgrowth, particularly at high Tamoxifen doses (ranging from 50 to 150 μg per mouse) (Brash et al. 2020). We now confirm that the effects we observed in ZO-1^{EC-KO} mice are not due to high Cre expression but a direct consequence of ZO-1 deletion of ZO-1 in ECs. Firstly, the dosage of Tamoxifen we used for this study was 20 μg per mouse, which is at a level at which no known toxicities to retinal vascular development have been reported and we confirmed that this dose was sufficient to induce Cre-dependent ZO-1 deletion in ECs (Figure 8a). In addition, as per the reviewer's suggestion, we now added Cre+ mice (TAM-treated Pdgfb-iCreER) as controls. The revised Figure 8c shows that no difference was observed in the retinal vasculatures of Cre+ and *Tjp1*^{fl/fl} mice, demonstrating that Cre expression in our model does not interfere with vascular development.

5. The in vitro and in vivo effects of deletion of ZO1 do not match, possibly due to the detrimental effects of Cre (see above). In vitro, stress-induced SG formation results in decreased migration which is rescued when cells are depleted for ZO1 (Figure 4f, g). In contrast, in vivo, retinal vessel outgrowth is delayed in the ZO1-deficient pups, which appears to be dependent on decreased migration as proliferation is actually increased (Fig 7). How do the authors explain these opposing effects of ZO1 depletion in vitro and in vivo? Moreover, Tornevaca et al. PMID: 25753039 show that wound healing is delayed in endothelial cells treated with ZO1 siRNA, which is in contrast to the data shown here where siZO1 enhances endothelial migration. This needs to be sorted out.

We are confident that the effects observed *in vivo* are in fact due to the deletion of ZO-1 and not caused by detrimental effects of Cre expression (see response to comment #4). While we agree that the *in vitro* and *in vivo* effects of ZO-1 deletion appeared to be contradictory, we have clarified our explanations and added new analyses that now clearly demonstrate that the *in vivo* and *in vitro* results complement each other.

First, it is important to consider that our *in vitro* studies investigate the cytoprotective effects of SG formation induced by the downregulation of ZO-1 in ECs. Hence, arsenite-induced stress did, as expected, inhibit the migration of ECs. We show in Figure 4f that ZO-1 depletion in ECs is protective against stress due to the increased formation of stress granules and restores migration in presence of arsenite to levels comparable to VEGF-stimulated ECs. Furthermore, Figure 6b adds to this by showing that this protective effect of ZO-1 downregulation on migration is dependent on YB-1-induced SG formation.

Second, the new Figure 8 now provides more detailed analyses of the altered retinal vasculature in ZO-1^{EC-KO} mice. It shows that deletion of ZO-1 in ECs *in vivo* increases retinal vascular density and disorganization (Figure 8d, e), decreases the number of vascular endpoints (8f), decreases the number, length and directedness of filopodia on Tip ECs (Figures 8h, i, j), and increases the

proliferation of retinal ECs. While it is not clear if stress conditions prevail in ECs during normal vascular development, this study reveals for the first time the presence of YB-1 and G3BP1 positive granules in ECs of angiogenic vessels. Efficacious angiogenesis *in vivo* requires a perfect coordination of several complex mechanisms that are finely regulated, such as proliferation, migration, cell communication, the formation and breakdown of cell junctions. Our new detailed analyses of the retinal vasculature clearly show that deletion of ZO-1 in ECs *in vivo* increases granule formation but also alters many of the angiogenic processes resulting in an overall inhibition of vascular growth in the retina. We cannot directly determine if migration of ZO-1^{EC-KO} ECs is increased *in vivo* but the increased disorganization and proliferation and reduced polarization of ZO-1 KO Tip-ECs surely contribute to the inhibition of vascular growth.

While it is true that Tornavaca et al. reported contradictory findings regarding migration, it is widely acknowledged in the literature that ZO-1 depletion enhances the mobility of many cell types, including tumor cells (Hyung Seok Kim et al., *Int. J. Mol. Sci*, 2023, PMID: 37240145). In fact, a decrease in ZO-1 levels is a recognized hallmark of epithelial to mesenchymal transition (EMT) of tumor cells which results in increased migratory potential (Mingyang Liu et al., *Clin Cancer Res*, 2018, PMID: 29615456). In addition, several other studies using ECs showed results similar to ours where decreased levels of ZO-1 by miRNA targeting ZO-1 result in increased migration (Y-L Hsu et al., *Oncogene*, 2017, PMID: 28436951; Weiyang Zhou et al., *Cancer cell*, 2014, PMID: 24735924).

6. The authors show that endothelial proliferation is increased in the ZO1-deficient retina vasculature, judged from the number of pH3 positive cells compared to that for wildtype retina vessels (Fig 7g). Still the vascular density does not seem to be affected judged from the images in Fig 7b. Please explain.

We have now refined the quantification of pH3-positive cells and we note that the increase in proliferation due to deletion of the *Tjp1* gene occurs only in the vascular plexus (Fig. 8k) and this translates into an increase in vascular density (Fig. 8d).

7. In ZO1-deficient conditions, SG formation is increased, paralleled by cytoprotection. Can the authors discuss whether the cytoprotective effect of ZO1 deficiency could be dependent on transcription of YB1 target genes rather than being solely through binding of YB1 to GPBP1 and SG formation?

Following the reviewer's comment, we investigated the transcriptional targets of YB-1 following ZO-1 siRNA transfection in ECs. We found that ZO-1 depletion leads to an increase in the expression of YB-1 and its transcriptional target BCL-XL (Fig. 6c). As expected, ZO-1 depletion also increases the expression of G3BP1 indicating SG formation. It is important to highlight that we don't refer to the interaction between YB-1 and G3BP1 as a mechanism of promoting or inhibiting SG assembly but rather as an indicator of SG presence since G3BP1 is a commonly used SG marker.

Minor

8. The authors study the developing retina. It would be preferable to use the well-established designations “vascular front” instead of “distal” or “extremities” of the vasculature and

“remodelling plexus” instead of “median”.

We agree with the reviewer. In our revised manuscript, we used “vascular front” instead of “distal” and “plexus” instead of “median” to describe the developing retinal vasculature.

Reviewer #2 (Remarks to the Author):

In this manuscript, the authors describe a new mechanism by which ZO-1, a membrane protein involved in the regulation of intercellular junctions, controls the assembly of stress granules through an interaction with YB-1, an mRNA-Binding proteins. In a previous report (Balda et al., JCB, 2003), an interaction between YB-3, another member of the Y-box family was already proposed. Starting with the identification of ZO-1 interactome, which reveals the preferential interaction of ZO-1 with proteins involved in RNA processing, the authors pay a particular attention to a putative role of ZO-1 in the regulation of stress granule assembly. In arsenite-stressed endothelial cells, ZO-1 is partly degraded and its interaction with YB-1 is decreased. A decreased interaction between YB-1 and ZO-1 then promotes stress granule assembly since the authors considered YB-1 to be a positive regulator of stress granules assembly. In the model proposed by the authors, ZO-1 “sequesters” YB-1 that would be subsequently released after stress to promote stress granule assembly. YB-1, after its release by ZO-1, can notably interact with G3BP-1, a protein known to promote SG assembly. Finally, the proposed model is put to the test in a more physiological system in endothelial cells of the retina.

The relationship between membrane proteins and RNA-processing proteins is very interesting topic. The results can reveal whether cells regulate their translational response to stress based on intercellular contacts. However, I do not think that the data presented in this manuscript supports the proposed model by which ZO1 would control stress granule assembly based on an interaction with YB-1.

We thank the reviewer for his insightful comments. We believe that our revised manuscript contains additional results supporting the proposed model by which ZO-1, through its interaction with YB-1, controls SG formation during angiogenesis.

Here are my specific concerns:

1) In the manuscript, when the expression of ZO-1 is decreased, SG assembly is promoted after arsenite stress in endothelial cells. However ZO-1 is known to limit cell proliferation based on contact inhibition between endothelial or epithelial cells (Balda, J cell biology, 2003). The formation of mRNA-rich stress granules follows the dissociation of polysomes due the inhibition of translation initiation. In proliferating cells, there are more polysomes whose dissociation after stress is more likely to form stress granules. Even if the same experiments were performed with VE-Cad and beta-catenin as controls (Fig. 2), the proliferation status or (and) mRNA translation should be measured under their experimental conditions (in addition at least two siRNAs have to be used to discard potential non specific effects).

This is an interesting point raised by the reviewer. In our revised manuscript, we show that in arsenite-treated cells, ZO-1 depletion inhibits translation initiation as indicated by elevated levels of phosphorylation of eIF2 α at serine 51 (Fig. 6d). As the reviewer mentioned, ZO-1 is well known to control cell proliferation, and previous work from our laboratory also showed that ZO-1 depletion increases EC proliferation (Chidiac et al. MCP 2016, PMID: 26846344). To address the reviewer’s comment, we measured SG formation in a proliferation status in which control and ZO-1 depleted ECs were treated with VEGF, a growth factor known to induce EC proliferation (Fig.

2c). It is important to note that in the absence of stress conditions, VEGF treatment alone is not able to induce SG formation (Fig. 2c), arguing against proliferation being the main driver of SG formation. However, in the presence of Arsenite, VEGF pre-treatment of ZO-1 depleted ECs significantly reduced SG formation (Fig. 2c). This observation was consistent with our findings in migrating cells (Fig. 4c, d), where cells at the leading edge demonstrated reduced SG formation in the presence of VEGF. These results show that VEGF-increased EC proliferation is not linked to SG formation and VEGF is cytoprotective in ECs in parallel to ZO-1 signaling to SGs.

Following the reviewer's recommendation to rule out nonspecific effects, we performed experiments with two distinct siRNAs targeting ZO-1. We observed a consistent increase in SG formation in response to arsenite treatment, as shown in Fig. S2a.

1 Cont'd): In the wound healing assays (Fig. 4), the author reports that the occurrence of SGs is higher in leading cells than in followers. After wound, the rate of proliferation increases which promotes stress granule assembly (as show previously in epithelial cells (Desforges, MBOC, 2013). ZO1 is known to control cell proliferation (Balda, J cell biology, 2003 and). Downregulation ZO1 may thus promote SG assembly by increasing the proliferation rate of followers. This hypothesis should be also considered.

We considered the hypothesis suggested by the reviewer. To this end, in the revised manuscript, we examined whether ZO-1 promotes SG assembly in follower and leading cells in response to VEGF treatment, which induces EC proliferation. Notably, reducing ZO-1 levels in absence of arsenite treatment enhances EC migration and proliferation without increasing SG formation. In addition, in arsenite-treated cells, VEGF pre-treatment did not increase SG formation in control follower ECs instead it markedly reduced SG formation in ZO-1-downregulated follower cells (Fig. 4c,d). VEGF pre-treatment also reduced SG formation in leading control and ZO-1 depleted ECs exposed to arsenite (Fig. 4d). Interestingly, VEGF stimulation resulted in a reduction of ZO-1 levels in control-transfected follower cells, but it did not further decrease ZO-1 levels in both leading control-transfected and siZO-1-transfected ECs (Fig. 4e). These results provide additional evidence that downregulation of ZO-1 promotes SG formation which in turn preserves cell migration independently of cell proliferation.

2) Why the authors focus their attention on YB-1 is not very clear but it may be due to the previous study on the YB-3:ZO1 system (YB-3 is very similar to YB-1 and there should be a comment about whether the proposed interplay between YB-1 with ZO-1 works also for YB-3. The authors show that silencing YB-1 counteracts the effect of silencing ZO-1 on SG assembly (Fig. 5). However, YB-1 is associated to cancer and elevated YB-1 level promotes high cancer cell proliferation rates. Silencing YB-1 expression may thus prevent an increased proliferation rate which may occur after silencing ZO1 expression. Therefore, this experiment does not tell whether silencing YB-1 specifically prevents the increased level of stress granule assembly in siZO1-treated cells.

We apologize for not making this clearer. In the revised version of the manuscript, we highlighted that both YB-3 (ZONAB) and YB-1 were identified within the ZO-1 interactome, thus providing a logical reason to focus our attention on YB-1 (p.12). In addition, our data from Figure 5a reveals that consistent with its function in SG formation, YB-1 protein levels were increased in response to arsenite treatment while the levels of YB-3 and ZO-1 were decreased. This indicates that YB-3

does not participate in SG formation and the interplay between YB-1 and ZO-1 is not applicable to YB-3.

We agree that YB-1 has been linked to cancer and silencing YB-1 may prevent an increase in cell proliferation but as discussed in the above comments, we added additional experiments showing that cell proliferation is not associated with SG formation (please refer to comment #1).

3) As suggested in the manuscript, ZO-1 may prevent YB-1 from interacting with G3BP-1. What are the expression levels of ZO1 in endothelial cells compared to those of YB-1, which is an abundant protein? If the authors propose that ZO1 can sequester YB-1 (it is not clear to me), a significant fraction of YB-1 under normal condition should colocalize with ZO1 at cell:cell junction. However, YB-1 is diffuse in the cytoplasm under physiological condition (figure 5 (I did not see the localization of YB-1 with ZO1 at cell:cell interface under normal condition)). The level of sequestration of YB-1 by ZO-1 should be estimated by measuring the fraction of YB-1 in the cytoplasm and associated with membranes before and after treatment with 0.5 mM arsenite for 30 min.

We thank the reviewer for his comment. First, we note that YB-1 expression levels are higher than ZO-1 in ECs based on published and unpublished RNA sequencing data from our laboratory (Smith T et al. Cell Mol Life Sci 2022). In our revised manuscript, we investigated the colocalization of YB-1 and ZO-1 at cell-cell junctions under physiological conditions. Figure 5d now shows that while ZO-1 is predominately localized to the membrane, YB-1 is mainly expressed in the cytoplasm but partially colocalizes with ZO-1 at cell-cell junctions. Following arsenite treatment, colocalization of YB-1/ZO-1 at cell-cell junctions significantly decreases (Fig.5b,c,d) mainly due to YB-1 re-colocalizing to SGs (Fig.5d, 3a, 4e). This provides additional evidence to support the sequestration of YB-1 by ZO-1 under normal conditions.

3 Cont'd) An increase of YB-1 level was observed after arsenite treatment (fig.5a) but it is misleading because this results was obtained after cells were exposed to 50 μ M for 6 h (leaving times for changes in protein expression) while 0.5 mM arsenite for 30 min was used in stress granule assays.

We apologize for the confusion. In Figure 5a, we treated ECs for 6h to investigate the effect of arsenite on YB-1, ZO-1 and ZONAB/YB-3 levels. We observed that YB-1 protein levels were increased in ECs treated with arsenite while the levels of ZONAB/YB-3 and ZO-1 were decreased, indicating that ZONAB/YB-3 is not involved in SG formation in response to arsenite. In contrast, in Figures 5b and 5c, the treatment with arsenite was for 30min to capture the dynamic of interaction between ZO-1 and YB-1 during SG assembly. In these experiments, we could not interpret the effect on YB-1 expression as 30min is too short to observe any changes in protein levels. We agree with the reviewer that this was confusing and more details were added in the text to clarify the difference in the treatment conditions used in Figure 5a.

4) In the proposed model, YB-1 interacts with ZO-1 under normal condition but to lesser extent after stress which is somehow logic as ZO1 is partly degraded (figure 3). It is important to show whether the decreased interactions depend on ZO1 degradation (by using MG132).

This is an important point. To address the reviewer's comment, we investigated the interaction between ZO-1 and YB-1 in the presence of the proteasome inhibitor MG132. In our revised manuscript, we show in Figure S4d that blocking the proteasome activity with MG132 did not prevent the arsenite-induced dissociation of ZO-1 and YB-1 indicating that the decreased interaction between ZO-1 and YB-1 is initiated prior to ZO-1 degradation. However, degradation of ZO-1 most probably participates in maintaining the cytoprotective effect of SGs.

4 Cont'd) In addition, the author indicates that ZO1 has been identified as an RNA-binding protein. They should also show whether the interactions between YB-1 and ZO1 is RNA dependent with RNase treatment and compare the results with other mRNA binding proteins as controls. Moreover, if there is a direct interaction, it would be interesting to identify which domains are necessary for this interaction to take place.

We thank the reviewer for pointing this out. To investigate whether the YB-1/ZO-1 interaction is RNA dependent, we conducted co-immunoprecipitation experiments of ZO-1 and YB-1 in Figure S4c followed by treatment with RNase A/T or Benzonase. In parallel, the effect of RNase on YB-1 and G3BP1 was also examined (positive control). We found that unlike the interaction between YB-1 and G3BP1 that was significantly decreased in response to RNase A/T treatment or Benzonase, the interaction between ZO-1 and YB-1 was not affected. This suggests that the association between ZO-1 and YB-1 occurs through a mechanism that is not dependent on the presence of RNA.

We agree that the identification of domains responsible for the interaction between ZO-1 and YB-1 is intriguing and we intend to explore this in more detail in future experiments, but we believe that this is out of the scope of our present study. However, based on the known domains of ZO-1 and YB-1, we added the following to the discussion of our revised manuscript p. 20: “The identification of another Y-box protein associated to ZO-1 suggests that the cold shock domains present in this family of proteins may be involved in the association with ZO-1. In addition, it is known that ZONAB binds to the SH3 domain of ZO-1. Given the sequence homology and similar SH3 domains present in all Y-box proteins, we may speculate that YB-1 binds to ZO-1 through its SH3 domain”.

5) ZO1 is not found in stress granules even if pull down did show a partial interaction (even reduced) between YB-1 and ZO1 after arsenite stress. As YB-1 is associated to stress granules, we may expect to partly detect ZO1 in SGs. May be that, as indicated in the manuscript, YB-1 no longer interacts with ZO1 because of an interaction with G3BP-1 or the degradation of ZO-1. To my knowledge, there is no report on a possible direct interaction between YB-1 and G3BP-1 (mRNA dependent?). If yes, the author should indicate this point. The authors should also elaborate on this hypothesis. We do not know whether the reported interaction between YB-1 and G3BP-1, if confirmed, can promote or inhibit stress granule assembly.

Our proposed mechanism is that ZO-1 interacts and sequesters YB-1 but following stress conditions, ZO-1 releases YB-1 enabling it to form SGs where it interacts with G3BP1. We did not observe ZO-1 in SGs and the partial interaction between ZO-1 and YB-1 after arsenite treatment suggests that following stress conditions, a portion of YB-1 goes to SGs and the remaining interaction between YB-1 and ZO-1 may be outside of SGs.

In line with the known role of G3BP1 in SG formation, we use the interaction between YB-1 to G3BP1 as an initiator of SGs presence without referring to this interaction as a potential mechanism. Furthermore, as mentioned above, our new results in Figure S4d show that YB-1/G3BP1 interaction decreases in the presence of RNase A/T or benzonase. This indicates that the interaction between YB-1 and G3BP1 is RNA-dependent.

*6) In the introduction, YB-1 is presented as a positive regulator of stress granule assembly. While it is has been proposed that YB-1 may promote stress granule assembly by increasing the expression of G3BP-1 (Somasekharan et al., Journal of Cell Biology, 2015), several genome-wide analyzes did not show that YB-1 is critical for stress granule assembly (Wheeler, Nature Methods, 2020 and Youn, Molecular Cell, 2018). YB-1 could even be a negative regulator of stress granule assembly (Ripin, RNA, 2022, Budkina , Nucleic acids research, 2021). Of course, the role of YB-1 in stress granules can be contextual and cell type dependent but these results should be mentioned in the text. In this manuscript, the authors show that the % of cells with stress granules ~~increases~~ (**decreases**) after siYB-1 treatment but, as I already mentioned, the proliferation rate of the cells may have decreased. Besides, even if these results are controversial (see Lyons, Nucleic acids research, 2016), YB-1 may in specific cases increase the expression level of G3BP1 which may provide an explanation for a positive regulation of SGs by YB-1. It would be interesting for the authors to probe whether G3BP-1 expression decreases after silencing YB-1 in their endothelial cells.*

We apologize for the confusion. As mentioned before, we used the interaction of YB-1 and G3BP1 as a biochemical indicator of SG formation. As suggested by the reviewer, we further investigated the effect of YB-1 overexpression or depletion on SG formation in ECs. We found that YB-1 overexpression increases SG formation in the absence or presence of arsenite (Fig. 5f) while the percentage of SGs decreases in YB-1 depleted cells under stress conditions (Fig. 5e). The expression of G3BP1 in YB-1 depleted cells was decreased without affecting ZO-1 levels (Fig. 6c). Conversely when ZO-1 is depleted, there is an increase in the expression of both YB-1 and G3BP1 (Fig. S4a, 6c). This shows that in our conditions in ECs, YB-1 acts as a positive regulator of SGs in part through the increase of G3BP1 expression. Articles by Ripin et Budkina are now cited (p. 20).

In summary, the subject of this manuscript is very interesting. However, the proposed model needs to be supported by stronger experimental evidence.

We thank the reviewer for providing helpful comments that have improved the quality of the study. In our revised manuscript, we have incorporated additional experiments to provide more robust evidence supporting our model.

Reviewer #3 (Remarks to the Author):

The work by Gratton and co-workers represents a comprehensive and detailed analysis of the influence of ZO1 protein levels on the formation of so-called stress granules in endothelial cells. In short, the authors show that down-regulation of ZO-1 enhances the formation of SGs under stress conditions (i.e. arsenite treatment), thereby protecting cells from cell death. This function is independent of other junctional components such as VE-cadherin or β -Catenin. At the molecular level, the authors show that formation of SGs require the protein YB-1 downstream of ZO-1 and show that stress leads to a dissociation of YB-1 from ZO-1.

Overall, the study is carefully executed, and the data are of high quality. Because of the wealth of data, the arguments are not always easy to follow. A graphical abstract detailing the proposed mechanism of ZO-1 and YB-1 function would be helpful.

We thank the reviewer for finding our study well-executed and of high quality. We are confident that the arguments in our revised manuscript are now more straightforward and clearer. Because of space limitations (the revised manuscript has now 8 figures), we decided not to include a graphical abstract, but we are ready to provide one if it is required.

Major point:

YB-1 is proposed to be the key factor as YB-1 kd prevents SG formation downstream of ZO-1. This hypothesis should be tested by overexpression of YB-1 under normal, stress and ZO-1 overexpression conditions.

This is a good point raised by the reviewer that is now addressed by overexpressing mycYB-1, mCherryZO-1, or both in ECs in absence or presence of arsenite as shown in Figure 5f and Figure S4f. We found that the overexpression of YB-1 increased SG formation under normal conditions which was further increased in the presence of arsenite. Interestingly, SG formation induced by YB-1 overexpression was inhibited by the overexpression of ZO-1. These findings further support our hypothesis by showing that overexpression of ZO-1 inhibits YB-1-induced SG formation.

The authors propose two stress induced mechanisms. 1) downregulation of ZO-1 and 2) dissociation of YB-1. Is the dissociation of the proteins simply do the loss of the binding partner (ZO-1) or are these two distinct molecular mechanisms?

This is an important point. As discussed in our response to reviewer 2 (comment #4), we examined whether the decreased interactions depend on ZO1 degradation. Hence, we investigated the interaction between ZO-1 and YB-1 in presence or absence of the proteasome inhibitor MG132. We found that treatment of ECs with MG132 did not prevent the dissociation of ZO-1 and YB-1 induced by arsenite (Fig. S4d). This suggests that stress-induced dissociation of ZO-1 and YB-1 is initiated prior to ZO-1 degradation. However, degradation of ZO-1 most probably participates in maintaining the cytoprotective effect of SGs.

Along these lines:

Was YB-1 detected as a protein down-regulated upon VEGF-A treatment in the proteomic analysis? This is not mentioned in the text.

Figure 1b shows that the interaction between YB-1 and ZO-1 is increased in response to VEGF treatment (YBX1 is colored in green). However, we cannot address whether VEGF affects YB-1 expression since our proteomic experiment was performed in ZO-1 immunoprecipitation samples and not in total lysates. Furthermore, VEGF treatment was only 10 minutes to monitor changes in protein-protein interaction. We added the following in the revised manuscript p. 12 : “Notably, we found that the interaction between YB-1 and ZO-1 increased significantly in response to VEGF treatment (Fig. 1b).”

Is YB-1 localized to cell junctions in confluent cells?

We thank the reviewer for this question. As mentioned in our response to reviewer #2 (comment #3), we now show that a fraction of YB-1 and ZO-1 colocalize at cell junctions and this colocalization significantly decreases in arsenite-treated cells (Fig. 5d).

Minor points:

Figure 2:

To show that ZO-1 regulates SG formation independent of junctions the authors analyze VE-cad and β -cat knockdowns. It would also be important to knock down tight junction proteins, e.g. Cldn5 or Esam, which may increase cytoplasmic ZO-1 levels

This is an important point. As suggested by the reviewer, we have examined in our revised manuscript the depletion of other tight junction proteins, Claudin-5 (Cldn5) and ZO-2 (Figure S2i, S2j). In contrast to ZO-1 downregulation, the depletion of Cldn5 or ZO-2 did not increase SG formation in response to arsenite. This strongly suggests that these effects are specific to ZO-1 and are independent of cell junctions.

In Figure 2f, the cell junction becomes merely visible. Is ZO-1 specifically removed from EC junctions. Does VE-cadherin still show normal distribution?

We thank the reviewer for pointing this out. To clarify this, we have quantified the localization of ZO-1, VE-Cadherin and β -catenin at the cell-cell junctions in the absence or presence of arsenite. As shown in the new Figure 3a, arsenite treatment resulted in a significant decrease in ZO-1 localization at EC junctions without affecting VE-Cadherin or β -catenin localization at cell junctions.

Figure 4:

The authors perform scratch-wound assays and suggest that down-regulation of ZO-1 during EC migration protects angiogenic ECs from stress. Down-regulation of junctional components is a prerequisite for angiogenic sprouting. In this context, the authors observe SGs only upon arsenite treatment. Therefore, the author’s interpretation (lines 212f) appears unwarranted.

We agree with the reviewer that our initial interpretation was overstated and therefore we toned down our conclusion in the revised manuscript p.11-12 to the following: “These results indicate that downregulation of ZO-1 promotes SG formation, which sustains cell migration in presence of

arsenite. However, VEGF-mediated EC cytoprotection and migration are not influenced by ZO-1 levels and are independent of SG formation.”.

Figure 5:

In Figure 5a, the authors show that YB-1 levels increase upon 30'-minute-arsenite treatment. This is surprisingly fast. How is this upregulation achieved? De novo protein synthesis?

This point was raised by all reviewers, we apologize for not being clear. In Figure 5a, we used a lower concentration (50 μ M) of arsenite for 6 hours (not 30min) treatment, which is a sufficient duration to increase YB-1 levels and downregulate ZO-1 protein expression. We clarified this in our revised manuscript (p. 44) and all doses and duration of treatments are indicated in figure legends.

Figure 7:

The EC specific knock-out of ZO-1 leads to strong defects in the retinal vasculature. However, the authors do not present evidence that this is due to misregulation of SGs.

We agree with the reviewer that we did not present direct evidence that retinal defects in ZO-1^{EC-KO} are due to misregulation of SGs. To address this issue, we performed a more detailed analyses of the altered retinal vasculature in ZO-1^{EC-KO} mice shown in the new Figure 8. It shows that deletion of ZO-1 in ECs in vivo increases retinal vascular density and disorganization (Figure 8d, e), decreases the number of vascular endpoints (8f), decreases the number, length and directedness of filopodia on Tip ECs (Figures 8h, i, j), and increases the proliferation of retinal ECs. While it is not clear if stress conditions prevail in ECs during normal vascular development, this study reveals for the first time the presence of YB-1 and G3PB1 positive granules in ECs of angiogenic vessels. Efficacious angiogenesis in vivo requires perfect coordination of several complex mechanisms that are finely regulated, such as proliferation, migration, cell communication, and the formation and breakdown of cell junctions. Our new detailed analyses of the retinal vasculature clearly show that deletion of ZO-1 in ECs in vivo increases granule formation but also alters many of the angiogenic processes resulting in an overall inhibition of vascular growth in the retina. Taken together, along with our extensive in vitro results, we can conclude that tight regulation of ZO-1 levels and of SG formation in ECs are required for proper angiogenesis.

Reviewer #4 (Remarks to the Author):

Reviewer's comments:

The study described in the manuscript by Bakkouri et al showed that interaction of tight junction protein ZO-1 with YB-1 controls stress granule formation of endothelial cells (ECs) during angiogenesis. First, the authors performed classical proteomics-based AP-MS approach to identify VEGF-induced interactors of ZO-1 in ECs. Through standard bioinformatics analysis of ZO-1 interactors with or without VEGF treatment, they showed ZO-1 interacts with YB-1 and FUS, which are associated with stress granule formation, and establish a novel link between ZO-1 and stress granules. Authors further showed that association of ECs ZO-1 with YB-1 negatively regulates the formation of stress granules through series of experiments. Furthermore, using EC-specific TAM-inducible ZO-1 knockout mice, they showed increased formation of stress granules in ZO-1 deficient ECs in the retina of new-born mice and retinal angiogenesis defects in the ZO-1 knockout mice.

Overall, the manuscript is well-written, and the obtained results are presented logically. While the experimental designs are acceptable, they are based on routine and conventional strategies (such as AP-MS). The major conclusions are somewhat sufficiently supported by experimental evidence. Although the study adds new knowledge to the field with regards to the new role of ZO-1 in stress granule formulation, conceptual advance, and significance of the work in the broader context is unclear. I have the following major concerns regarding the methodology employed, results obtained and possible additional experiments that should be addressed before recommending this work for publication in Nature Communications.

We thank the reviewer for the insightful review and the valuable recommendations that have significantly improved our manuscript.

1. The authors carry out AP-MS to capture interacting partners. This method is known to generate a lot of non-specific interactors particularly in the label-free format it is being carried out. For identifying specific interactors with high confidence especially under perturbation such as VEGF treatment, a proximity biotinylation method with recent advancements (e.g., Santos-Barriopedro et al, Nat Commun 2021) is recommended. This is particularly useful to identify the dynamic interactome of a junctional protein such as ZO-1 whose immediate interactome could change depending on the cellular localization.

We are well aware that there are now several techniques available to identify interacting partners, particularly when anticipating rapid changes in the interactome due to a particular treatment. However, the primary goal of this study was to uncover a new biological function for ZO-1 in ECs, which was achieved by performing AP-MS. Hence, we identified novel and known ZO-1 interactors but more importantly, novel biological processes in which ZO-1 is implicated. We are highly confident in the interactors that we identified, in particular YB-1, given the extensive confirmations and validations we carried out in our study.

2. BAEC of bovine origin was chosen for interactome study while HUMVE was used for other experiments. Authors should explain this.

BAECs were chosen to perform our proteomic experiments because of their reproducible and robust response to VEGF treatment. All experiments were replicated and validated in other cell lines including HUVEC (human umbilical vein endothelial cells), TeloHAEC (immortal human aortic endothelial cells), HepG2 (human liver cancer cell line) and BAEC (bovine aortic endothelial cells). We believe that this confirms the universality of our findings.

3. The authors have picked up a few interactors for subsequent validation. What is the rationale for selecting such candidates (Fig. 1c)? Are these proteins the top hits (statistically significant or functional relevance or just cherry picking)?

In order to validate ZO-1 interactors identified in our mass spectrometry, we selected known and novel ZO-1 interactors implicated in specific biological processes functionally relevant to our study. For instance, to confirm our MS results, we validated the previously identified interaction of ZO-1 with the junctional protein JUP (junction plakoglobin) and with the transcriptional factor ZONAB (Fig. 1c, Fig. S1c). In our revised manuscript p. 6, we added the following: “We confirmed the previously identified ZO-1/JUP and ZO-1/ZONAB interactions by IP and showed that VEGF treatment increased the association between ZO-1 and JUP (Fig. 1b, c).” Next, to validate the novel interaction between ZO-1 and RBPs, we selected AIMP1 and RPL23A, known RNA binding proteins (RBPs) with commercially available antibodies that allow us to examine the endogenous interaction with ZO-1 (Fig. 1c). More importantly, the novel interaction between ZO-1 and YB-1 was also validated.

Since YB-1 and G3BP1 were picked up for subsequent studies why they have not been included in the Western blot validation shown in Fig. 1c.? These two proteins should be included in this Western validation. And also significantly enriched proteins including YB-1, G3BP1 and other relevant proteins such as stress granule proteins should be labelled in Fig. 1a.

G3BP1 was not identified as an interactor of ZO-1 in our MS results, but since G3BP1 is a commonly used marker of SGs, we showed that arsenite treatment induces the interaction between YB-1 and G3BP1 indicating an increase SG formation (Fig. 5g). In Figure 1e, ZO-1 interactome that are part of the ribonucleoprotein complex and known to be associated with SGs were already highlighted in red (Fig. 1e).

4. It is not clear why authors have selected VEGF treatment for the first pull-down experiment. Also, the authors used VEGF treatment for 10 minutes. Does 10 minutes of VEGF treatment alter localization of ZO-1? How did authors arrive at this stimulation time? There seems to be no follow up with regards to the relevance of VEGF in the later sections of the manuscript. The reasons for all these should clearly be stated.

We employed VEGF treatment to examine alterations in the ZO-1 interactome in response to an angiogenic cue. We previously showed, in Chidiac et al Mol. Cell. Proteomics 2016, that VEGF treatment for 10 minutes is the optimal time point to activate signaling pathways downstream of the VEGF receptor-2 (VEGFR2) such as p-Akt, p-eNOS and p-ERK1/2 known to play a role in EC migration, survival, permeability and cell proliferation. In addition, we added results to document the importance of the YB-1/ZO-1 interaction in VEGF-mediated cytoprotective and migration of ECs (Fig. 2c, 4d, 4e, 4f).

5. It is also recommended to perform this interactome analysis under arsenite treatment, which is directly related to stress granule formation (or stress tolerance) as authors mainly used arsenite treatment for cell-line based validation with regards to stress granule assembly.

This is a valid point raised by the reviewer. Unfortunately, we found such an experiment to be complex and potentially unfeasible. Our data indicate that arsenite treatment markedly reduces ZO-1 expression levels (Fig. 3a to 3f), which would result in the impossibility of detecting the ZO-1 interactome and make it uninterpretable, consequently reducing our confidence in the outcomes.

6. The ZO family of proteins include ZO-1, ZO-2, and ZO-3. Do the other proteins in the family have roles similar to ZO-1 in influencing stress granule formation? Is there any compensatory mechanism during loss of ZO-1? Authors can comment on this in the discussion.

Thank you for this question. RNA sequencing data from our lab and others indicate that ZO-3 is not expressed in endothelial cells. We now show in figure S2j that ZO-2 downregulation did not increase SG formation in response to arsenite strongly suggesting that these effects are specific to ZO-1.

7. For shortlisting the interacting proteins, the authors have performed statistical analysis. Were any attempts made to filter off non-specific interactors or false positive interactors? It is useful to check against the CRAPome database (<https://reprint-apms.org/?q=chooseworkflow>) to make sure the interactors reported are indeed confident.

We have followed the reviewer's suggestions and used CRAPome Web interface to further investigate the specificity of our interactions identified in our affinity purification experiment. Our initial filtering of non-specific interactions was done by performing a statistical analysis of the bait immunoprecipitation of each condition versus IgG control immunoprecipitations. Isotype control antibody represented our internal negative control condition to filter non-specific interactors. The CRAPome database was then used to filter out additional non-specific interactors. This text is now added to the methods section of our revised manuscript p. 30-31: "We used CRAPome Web interface (www.crapome.org) to confirm the specificity of interactions identified in our affinity purification experiments. From the CRAPome database, we selected agarose beads as affinity support for control experiments similar to our experimental condition. As shown in Table S3, interactors were cross-referenced with this CRAPome list. Only 3 out of 125 ZO-1 interactors (TUBB, HSPB1 and SNRPD1) had an average spectral count above 15. This indicates the high specificity of all the interactions identified in our AP-MS. However, these three proteins were not removed from the ZO-1 interactome because their interaction with ZO-1 was significantly above our statistical threshold and influenced by VEGF treatment."

8. The authors report that ZO-1 shares interactions with RBPs involved in stress granule formation. However, later they showed that ZO-1 levels negatively regulate stress granule formation and loss of ZO-1 triggers stress granule assembly. This seems a bit confusing as to the nature of interaction between ZO-1 and the RBPs and the relevance of such interaction in a general context.

As previously stated, our study uncovered a novel mechanism by which ZO-1, a junctional protein, interacts with RBPs including YB-1 and modulates SG formation. *In vitro* and *in vivo* approaches using different molecular and biochemical experiments were used to specifically validate the YB-1 and ZO-1 interaction. As stated by other reviewers, the nature of the interaction between ZO-1 and RBPs is intriguing, and our study highlighted the relevance of this interaction in the context of blood vessel formation. We further elaborated on the impact and relevance of our findings in the revised manuscript which we hope is now less confusing.

9. While there is much discussion of ZO-1 and YB-1, the co-localization of ZO-1 and YB-1 in-vivo is not shown (not only based on Co-IP). Authors need to show co-localization of ZO-1 and YB-1 to confirm this interaction. This is especially important when ZO-1 is primarily a junctional protein and YB-1 is a cytoplasmic or nuclear.

As mentioned in our response to reviewers #1 and #2, we now show a fraction of YB-1 and ZO-1 colocalize at the cell junctions and this colocalization significantly decreases in arsenite-treated cells (Fig. 5d).

10. It will be useful to assess if any domains within ZO-1 is required for binding to YB-1? Authors can express ZO-1 without such domain and check the effect of stress granule assembly/disassembly?

Echoing our response to reviewer #2 (Comment #4), we added the following to the discussion of our revised manuscript p. 21: “The identification of another Y-box protein associated to ZO-1 suggests that the cold shock domains present in this family of proteins may be involved in the association with ZO-1. In addition, it is known that ZONAB binds to the SH3 domain of ZO-1. Given the sequence homology and similar SH3 domains present in all Y-box proteins, we may speculate that YB-1 binds to ZO-1 through its SH3 domain.”

11. Since ZO-1 is a tight junction protein, authors should elaborate the relationship between tight junction and stress granule formation if any. Is there any cross talk?

To our knowledge, this is the first study reporting a relationship between a tight junction protein, ZO-1, and stress granule formation. To confirm the relationship between ZO-1 and SG formation, we depleted other tight junction proteins, Claudin5 (Cldn5) or ZO-2 in ECs (Figure S2i). We found that, in contrast to ZO-1 depletion, the downregulation of Cldn5 or ZO-2 did not increase SG formation.

12. The relevance of ZO-1 influenced stress granule formation is unclear. Authors should emphasis and elaborate the importance of stress granule formation and how this can be significant in the context of disease or processes such as angiogenesis.

In the introduction of our revised manuscript, we elaborated on the importance of SGs formation and its impact in a specific biological context or disease. The following was added to the introduction p. 4-5: “Stress granules (SGs) are membranellar ribonucleoproteins (RNPs) conglomerates comprising 40S small ribosomal subunit, translation initiation factors, various RNA-binding proteins (RBPs) and translationally stalled mRNAs. Emerging evidence suggests

that SGs are dynamic membraneless cytoplasmic structures that form in response to acute or chronic stress. Their composition undergoes rapid remodeling influencing cell fate during the stress period. The stress-induced phosphorylation of the eukaryotic translation initiation factor 2 α (eIF2 α) attenuates global mRNA translation while promoting both acute and chronic SG assembly. SGs are generally thought to function as sites of mRNA triage, wherein individual mRNAs are dynamically sorted for storage during stress. Additionally, recent studies suggest that SGs function as modulators of innate immune response highlighting the multi-functional nature of SGs. SGs play a major role in homeostatic cellular stress responses whereas the alterations in their assembly and/or clearance are associated with a number of human pathologies including neurodegenerative and neoplastic diseases and viral infection”

13. Line 219-221 “consistent with its role in stress granule formation, YB-1 protein levels were increased in ECs treated with arsenite while the levels of ZONAB and of ZO-1 were decreased (Fig. 5a). YB-1 level does not seem to be increased in the presence of Arsenite (Fig.5b and c input). How was input normalized? Loading control should be included to avoid confusion.

We apologize for the confusion caused by the different doses and duration of arsenite treatment. In Figure 5a, we treated ECs for 6h in order to investigate the effect of arsenite on YB-1, ZO-1 and ZONAB/YB-3 levels. We observed that YB-1 protein levels were increased in ECs treated with arsenite while the levels of ZONAB/YB-3 and ZO-1 were decreased, indicating that ZONAB/YB-3 is not involved in SG formation in response to arsenite. In contrast, in Figures 5b and 5c, the treatment with arsenite was for 30min to capture the dynamic of interaction between ZO-1 and YB-1 during SG assembly. In these latter experiments, we can't interpret the effect on YB-1 expression as 30min is too short to observe any changes in protein levels. More details were added in the text to clarify the difference in the treatment conditions used in Figure 5a.

14. In the experiments discussing on migratory potential, it seems that decreased ZO-1 while promoting stress granule formation also increases migration during arsenite treatment. The authors concluded that section with “downregulation of ZO-1 promotes stress granule formation which in turn protects ECs from exogenous stresses that may occur during angiogenesis”. How is this a protective effect when decreased ZO-1 amplifies the migratory potential during angiogenesis? Isn't that detrimental?

This section was completely modified to include VEGF-stimulated migration in order to clarify the protective role of ZO-1 downregulation during cell migration. We also agree that our initial interpretation was overstated and therefore it was toned down in the revised manuscript p. 11-12 to the following: “These results indicate that downregulation of ZO-1 promotes SG formation, which sustains cell migration in presence of arsenite. However, VEGF-mediated EC cytoprotection and migration are not influenced by ZO-1 levels and are independent of SG formation.”.

15. Was there any stress applied to ZO-1 deficient mice that triggered stress granule formation or is it only triggered by loss of ZO-1. If stress was not applied, how does it align with previous in vitro results?

The correlation between our *in vitro* and *in vivo* experiments was also raised by other reviewers. Firstly, no external stress was applied to ZO-1 deficient mice and the increase in YB-1-positive granule formation is merely due to the loss of ZO-1 expression. The *in vitro* results correlate with the *in vivo* results because the percentage of YB-1 positive granules was higher at the vascular front of the retina which is associated with a decrease in ZO-1 expression (Fig. 7a, b). The new analyses performed in the retinal vasculature of ZO-1 deficient mice allowed us to observe an increase in vascular density, in endothelial cell proliferation and disorganization of ZO-1^{EC-KO} retinas (Fig. 8). Importantly, we also observed an increase in YB-1-positive granules in the vascular front and remodeling plexus of ZO-1^{EC-KO} retinas. However, it remains unclear whether the formation of YB-1- or G3BP1-positive granules in this context is a response to stress or a normal physiological mechanism used by ECs during angiogenesis.

16. No follow up was made on post stress recovery with regards to stress granule levels and ZO-1 expression being bounced back. Rescue experiments should be performed to assess the specific relationship of ZO-1 with stress granule assembly. Also, any experiments to overexpress ZO to correlate to reduce stress granule or disassembly of stress granules is needed to establish a direct role.

Although we agree that investigating post-stress recovery in regard to SGs formation and ZO-1 expression is of interest, we believe that this is out of the scope of our present study. Conducting such an experiment is complex since it demands determining the precise timing and dosages necessary for post-stress recovery, not to mention the ambiguity associated with defining what constitutes post-stress recovery. However, as shown in Figures 2d and 5f, we overexpressed ZO-1 in ECs and found that the overexpression of ZO-1 did decrease SG formation.

17. The relevance of angiogenesis to this whole study is unclear. Angiogenesis is described once in a while and the relevance of VEGF in the interactome performed is unclear as there is no follow up. Any other process other than angiogenesis is relevant?

This extensively revised manuscript better describes the angiogenic process (Introduction p. 3) and highlights the relevance of angiogenesis throughout. Furthermore, as mentioned above, we followed up on the VEGF-stimulated interactome by performing additional experiments that document the importance of the YB-1/ZO-1 interaction in VEGF-mediated cytoprotection and migration of ECs (Fig. 2c, 4d, 4e, 4f).

18. Signalling mechanism on how stress granule formation is triggered by ZO-1 deficiency is missing. The assembly of stress granule is largely driven by liquid-liquid phase separation (LLPS). Any protein modifications on stress granule proteins in ZO-1 deficient ECs that could serve to either disrupt critical multivalent interactions or provide new contact surfaces that drive LLPS?

This is an interesting point. Since ZO-1 has been shown to be involved in liquid-liquid phase separation during cell junction organization and mechanosensation to flow (Beutel et al., 2019, Cell 179, 923–936, Schwayer et al., 2019, Cell 179, 937–952). However, our proposed mechanism is that stress-induced ZO-1 downregulation releases YB-1 to drive SG formation. While it is possible that YB-1 or other RBPs may be involved in the liquid-liquid phase separation process

leading to SG formation (Liu et al. eLIFE 2021), we believe that ZO-1 is not directly implicated in LLPS related to SG formation. This is now added to the discussion.

Minor

1. Line 75: Include citation(s)

These citations are included.

2. VE-cad western-blot for figure 3a showed one band, figure 3b showed 2 bands. Authors need to comment on this.

HUVECs were used for Figure 3a while BAECs were used for Figure 3b. In addition, VE-cadherin appears as two bands on Western-blot, the appearance or not of the lower band is due to the time of exposure and expression levels.

3. Line 355. Include complete company information

Company information is included.

4. Line 359, 368. Company location is not listed. It should be consistent throughout the manuscript.

Company locations are now included.

5. In material and method. Mass Spectrometry acquisition parameters should be listed.

This information is included in our raw data submitted to the ProteomeXchange Consortium via the PRIDE partner repository with the dataset identifier PXD029332. We have now included it in the methods of our revised manuscript.

6. Line 506. p-value (typo)

We corrected this typo.

7. Line 512, Include citations for Cytoscape and MCL algorithm

We have provided appropriate references.

REVIEWER COMMENTS

Reviewer #1 (Remarks to the Author):

El Bakkouri et al have revised their paper “ZO-1 interacts with YB.1 in endothelial cells to regulate stress granule formation during angiogenesis”. The study is clearly improved, however still quite complex in particular in relation to VEGF-regulated endothelial proliferation. In the text to Fig. 4 (line 260-262) the authors conclude that “VEGF-mediated EC cytoprotection and migration are not influenced by ZO-1 levels and are independent on SG formation.” In Fig. 4, the authors do not analyze the protective effect of VEGF on EC apoptosis (which I presume is the definition of cytoprotection here). In their previous paper from 2016 (ref 11) they show that silencing ZO-1 results in enhanced ERK1/2 phosphorylation and EdU incorporation in response to VEGF, from a higher basal than in the control. In the abstract of the paper from 2016, they described “ZO-1 as part of a signaling node activated by VEGF, but not Ang-1, that specifically modulates EC proliferation during angiogenesis”. Moreover, ZO-1 is phosphorylated in response to VEGF (ref 11) and VEGF treatment leads to downregulation of ZO-1 expression levels (current study Fig. S2f). Now, without studying VEGF effects on cytoprotection in the current study, the authors uncouple the proliferative effect of VEGF from ZO-1. I cannot follow the authors reasoning here.

Moreover, on lines 260-262 the authors conclude that VEGF-induced migration is uncoupled from ZO-1 and stress granule, still in Fig. 4d, they show that SGs are reduced with VEGF stimulation (when ZO-1 levels are reduced). Arsenite-treatment with and without VEGF stimulation is not tested in the wound healing assay.

In Fig. 8, the authors show that endothelial loss of ZO-1 leads to aberrant vascular development with increased endothelial proliferation in the vascular plexus but maintained proliferation in the vascular front. This to me indicates that ZO-1 is required to balance VEGF-induced proliferation with morphogenesis (probably junction-dependent) to form a functional vasculature.

While I'm impressed by the efforts to improve their paper, the authors' thinking concerning the role of ZO-1 in VEGF biology is tangled and difficult to follow. I concur with what the authors write in the response to my first comment: “It is likely that these YB-1 positive granules have physiological roles during normal development of the vasculature and that by disturbing the balance of ZO-1 in EC-KO mice, the angiogenic program is altered thereby resulting in increased YB-1 positive granules and augmented EC proliferation in the remodeling plexus. This in turn leads to an increase in vascular density and disorganization, and ultimately a delay in the outgrowth of the vasculature.” It would improve the presentation and reading if the results were presented along these lines in a consistent manner.

Minor:

In the abstract, lines 52-53, the authors describe that deletion of ZO-1 leads to an “arrest in the growth of the retinal vasculature”. It is preferable to write “aberrant endothelial proliferation and arrest in the expansion of the retinal vasculature”

In the introduction, line 97, what is meant by “membranelles” here? Please modify this sentence, cannot be followed.

The description of the impact of ZO-1 on VEGF-induced proliferation and endothelial biology, is difficult to follow as described above, and on top of that, the language is sometimes not optimal. I suggest to have professional language editing of the text to improve further.

Reviewer #2 (Remarks to the Author):

The authors have done considerable work since their previous submission.

Some points have been clarified in response to my questions.

However, the fact that YB-1 can be sequestered by ZO-1 to prevent the interaction with G3BP1 and formation of stress granules is not documented.

As noted by the authors, YB-1 is significantly more expressed than ZO1 in endothelial cells. The authors provide an image showing that YB-1 colocalizes with ZO1 (Figure 5D). This is interesting for validating the interaction but no attempt to estimate the level of YB-1 sequestration was made. It could have been done by integrating the fluorescence along the junction between cells and that of the cytosol. From the image, we see that the presence of YB-1 in the cytosol does not seem altered. Furthermore, the enrichment of YB-1 along the junction between cells does not seem very marked. Most YB-1 is therefore likely ZO-1 free under control conditions.

In my view, they should indicate that there is no proof for sequestration but instead other mechanism may come into play. The interplay between ZO1 and YB-1 may control a signaling pathway that may control stress granule assembly or the RNA-dependent interaction of YB-1 with G3BP1. It may be also be due to the interaction of ZO1 with the other RBPS detected in the mass spectrometry analysis. Unless strong evidences are provided, I suggest to remove the “sequestration” model which is implicitly indicated in several sentences including in the abstract and discussion (for instance: “ Down regulation of ZO-1 increased the association of YB-1 and G3BP1”. “This association refrains YB-1 from engaging in SGs assembly”).

Other points:

1) Authors should indicate ZONAB as YB-3 in the manuscript all the time. The use of ZONAB is misleading.

2) Several questions I asked previously were related to the proliferation status of the endothelial cell in the absence or presence of ZO1 and under the various conditions tested. I would have preferred to have a direct measurement based on BRDU staining for example than using VEGF pretreatment.

I agree with the other data presented. The data indicating that ZO-1 interacts with many RBPs is very interesting.

Reviewer #3 (Remarks to the Author):

The authors have addressed some concerns raised by the reviewers. However, the model proposed by the authors is based on correlations and remains vague.

Figure 2:

One key finding of the paper is that downregulation of ZO1 promotes SG formation under stress. It is not clear to me why VEGF (VEGF-A121?), which leads to downregulation of junctional proteins (including VE-cad and ZO1), should prevent SG formation (Fig. 2c). This clearly shows that ZO1 downregulation is not sufficient for stress-induced SG formation. It also appears that VEGF treatment is cytoprotective while preventing SG formation.

The authors argue that SG induction by ZO1 depletion is independent of cell-cell junctions (Fig. S2.). However, simultaneous ZO1/VE-cad (or β -cat) knockdown enhances SG formation (Fig. 2e). Isn't this contradictory?

Figure 4:

Similar to Fig. 2c, VEGF-A treatment reduces ZO1 level while inhibiting SG formation.

In 4d, siCT in leading cells show SGs level similar to siZO1, however migration is reduced in scratch wound assays. At the same time VEGF-a treatment enhances cell migration irrespective of ZO-1 kd. This finding is in conflict with the in vivo data (Figures 7 and 8) showing migration defects in the retinas of

ZO1 ko, which are wild-type for VEGFa. Can the ZO1 ko phenotype be rescued by exogenous application of VEGF-A?

In the quantification of scratch wound assays (Fig. 4d and 3), it would be important to include an arsenite (-) control, to see whether SG formation and ZO1 are differentially regulated in leader and follower cells.

Figure 5:

The authors suggest that the binding (and dissociation) of ZO1 and YB-1 may represent the key regulatory mechanism of SG formation. The evidence presented is quite correlative.

Fig. 5a shows that long-term arsenite treatment leads to changes in expression of YB-1, YB-3 and ZO1. However, it is not clear how this long-term change impacts on SG formation.

Fig. 5d shows that YB-1 is lost from cell-cell junctions under arsenite treatment. While the data is convincing, it is difficult to conceive how this can explain the proposed mechanism, since YB-1 protein levels are much higher than ZO1 levels.

In Fig. 5f overexpression of YB-1 enhances while ZO1 overexpression reduces SG formation. Overexpression of both construct leads to an intermediate phenotype. It not clear, however this compensatory relationship is due to direct protein-protein interaction. By determining the interactive protein domain of ZO1 (SH3 as suggested by the authors?), the authors can test this hypothesis more directly, by using different mutant isoforms of ZO1 in rescue similar to those performed in Fig. 5f.

Reviewer #4 (Remarks to the Author):

Authors have addressed some of my concerns and comments on the original manuscript with support of additional experiments. However, the major suggested works such as the experiments related to interactome analysis under arsenite treatment (R4#5), assessing domain within ZO-1 (R4#10), post stress recovery (R4#16) and LLPS (R4#18), which are crucial to bring the manuscript to Nat Commun standard level have not been performed. Authors' reasoning on not performing such experiments such as out of scope and/or the complexity is not acceptable if this manuscript is considered for publication in Nat Commun.

Reviewer #1:

El Bakkouri et al have revised their paper “ZO-1 interacts with YB.1 in endothelial cells to regulate stress granule formation during angiogenesis”. The study is clearly improved, however still quite complex in particular in relation to VEGF-regulated endothelial proliferation. In the text to Fig. 4 (line 260-262) the authors conclude that “VEGF-mediated EC cytoprotection and migration are not influenced by ZO-1 levels and are independent on SG formation.” In Fig. 4, the authors do not analyze the protective effect of VEGF on EC apoptosis (which I presume is the definition of cytoprotection here). In their previous paper from 2016 (ref 11) they show that silencing ZO-1 results in enhanced ERK1/2 phosphorylation and EdU incorporation in response to VEGF, from a higher basal than in the control. In the abstract of the paper from 2016, they described “ZO-1 as part of a signaling node activated by VEGF, but not Ang-1, that specifically modulates EC proliferation during angiogenesis”. Moreover, ZO-1 is phosphorylated in response to VEGF (ref 11) and VEGF treatment leads to downregulation of ZO-1 expression levels (current study Fig. S2f). Now, without studying VEGF effects on cytoprotection in the current study, the authors uncouple the proliferative effect of VEGF from ZO-1. I cannot follow the authors reasoning here.

Thank you for acknowledging that our study was clearly improved. We totally agree with the reviewer's interpretation. The addition of the VEGF data was only meant to demonstrate that SG formation is not involved in the antiapoptotic effects of VEGF and to answer a reviewer's concerns on the possibility that cell proliferation drives SG formation. We apologize if it added confusion. It was not intended to mean that ZO-1 is not important for the effects of VEGF. As highlighted by the reviewer, it is clear that ZO-1 is involved in the proliferative effects of VEGF on endothelial cells.

Following the reviewer's suggestion, we examined VEGF effects on the viability of EC following arsenite treatment. As expected, we showed that a VEGF pre-treatment protects ECs against arsenite-induced cell death but reduces the number of SGs increased by arsenite treatment (Figure 4c, d). This contrasts with the cytoprotective effects of ZO-1 downregulation on arsenite-induced cell death, where SGs are increased. In ZO-1-downregulated ECs, VEGF pre-treatment still sustained the viability of cells but significantly decreased the arsenite-induced SG formation (Figure 4c, d). “These results suggest that the antiapoptotic effects of VEGF on ECs do not involve an increase in SG formation and differ from the protective effects of ZO-1 against stress, which involve the formation of SGs.”

Moreover, on lines 260-262 the authors conclude that VEGF-induced migration is uncoupled from ZO-1 and stress granule, still in Fig. 4d, they show that SGs are reduced with VEGF stimulation (when ZO-1 levels are reduced). Arsenite-treatment with and without VEGF stimulation is not tested in the wound healing assay.

We have removed the VEGF data from the former Figure 4d and 4e (new Figure 4f and 4g) since it was redundant with Figure 2c and created some confusion. Arsenite treatment of control-transfected ECs significantly attenuated their migration, which was restored in ZO-1-downregulated cells (Figure 4h). While VEGF treatment prevents arsenite-induced SG formation in ZO-1 downregulated cells, it maintains cell migration when ZO-1 is depleted (Figure 4d, h). Hence, the phrase on lines 260-262 of the previous version “However, VEGF-mediated EC cytoprotection and migration are not influenced by ZO-1 levels and are independent of SG formation.” was changed to: “These results indicate that the downregulation of ZO-1 promotes the

formation of SGs, thereby facilitating cell migration in the presence of arsenite, and that the formation of SGs is not involved in VEGF-stimulated migration or antiapoptotic effects.”

As required by the reviewer, we added arsenite treatment with and without VEGF stimulation into the wound healing assay (Figure 4h). These results show that VEGF-stimulated migration of ECs is inhibited by arsenite treatment and that the enhanced migration observed in ZO-1-downregulated is resistant to arsenite treatment.

In Fig. 8, the authors show that endothelial loss of ZO-1 leads to aberrant vascular development with increased endothelial proliferation in the vascular plexus but maintained proliferation in the vascular front. This to me indicates that ZO-1 is required to balance VEGF-induced proliferation with morphogenesis (probably junction-dependent) to form a functional vasculature.

We have incorporated the following in the discussion on page 21: “Efficacious angiogenesis *in vivo* requires the perfect coordination of several complex mechanisms that are finely regulated, such as proliferation, migration, cell communication, and the formation and breakdown of cell junctions. ZO-1 is likely required to coordinate these mechanisms, which are modulated by VEGF and involved in the formation of a functional vasculature.”

While I'm impressed by the efforts to improve their paper, the authors' thinking concerning the role of ZO-1 in VEGF biology is tangled and difficult to follow. I concur with what the authors write in the response to my first comment: “It is likely that these YB-1 positive granules have physiological roles during normal development of the vasculature and that by disturbing the balance of ZO-1 in EC-KO mice, the angiogenic program is altered thereby resulting in increased YB-1 positive granules and augmented EC proliferation in the remodeling plexus. This in turn leads to an increase in vascular density and disorganization, and ultimately a delay in the outgrowth of the vasculature.” It would improve the presentation and reading if the results were presented along these lines in a consistent manner.

We apologize for the confusion that the addition of the VEGF data might have created. As mentioned above, while we believe it still provides important information, we have removed some of the VEGF results from the former Figure 4d and 4e (new Figure 4e and 4f) since it was redundant with Figure 2c and it created some confusion. We are confident that the new data regarding VEGF-mediated cytoprotection (Figures 4c, d) requested by the reviewer, now makes our study less tangled and easy to follow. In addition, we have significantly revised and incorporated additional clarifications in the discussion (page 20): “However, we found that VEGF stimulation did not induce SG formation, indicating that the anti-apoptotic signaling mechanisms activated by VEGF are different from the ones activated by ZO-1 signaling to SGs in response to stress. It is well-established that the phosphoinositide 3-kinase/protein kinase B (PI3K/Akt) pathway is primarily responsible for the pro-survival and the anti-apoptotic effects of VEGF on ECs.”

Minor:

In the abstract, lines 52-53, the authors describe that deletion of ZO-1 leads to an “arrest in the growth of the retinal vasculature”. It is preferable to write “aberrant endothelial proliferation and arrest in the expansion of the retinal vasculature”

Thank you for this suggestion, we have modified the sentence in the abstract.

In the introduction, line 97, what is meant by “membranelles” here? Please modify this sentence, cannot be followed.

We apologize for this typo, “membranelles” was corrected for “membraneless”.

The description of the impact of ZO-1 on VEGF-induced proliferation and endothelial biology, is difficult to follow as described above, and on top of that, the language is sometimes not optimal. I suggest to have professional language editing of the text to improve further.

We appreciate the reviewer feedback and following the recommendations described above, we revised the text involving the impact of VEGF on EC biology and its role in SGs formation and improved the language used to describe the results. We substantially revised the manuscript, and it was proofread by a professional language editing service.

Reviewer #2:

The authors have done considerable work since their previous submission.

Some points have been clarified in response to my questions.

However, the fact that YB-1 can be sequestered by ZO-1 to prevent the interaction with G3BP1 and formation of stress granules is not documented.

As noted by the authors, YB-1 is significantly more expressed than ZO1 in endothelial cells. The authors provide an image showing that YB-1 colocalizes with ZO1 (Figure 5D). This is interesting for validating the interaction but no attempt to estimate the level of YB-1 sequestration was made. It could have been done by integrating the fluorescence along the junction between cells and that of the cytosol. From the image, we see that the presence of YB-1 in the cytosol does not seem altered. Furthermore, the enrichment of YB-1 along the junction between cells does not seem very marked. Most YB-1 is therefore likely ZO-1 free under control conditions.

In my view, they should indicate that there is no proof for sequestration but instead other mechanism may come into play. The interplay between ZO1 and YB-1 may control a signaling pathway that may control stress granule assembly or the RNA-dependent interaction of YB-1 with G3BP1. It may be also be due to the interaction of ZO1 with the other RBPS detected in the mass spectrometry analysis. Unless strong evidences are provided, I suggest to remove the “sequestration” model which is implicitly indicated in several sentences including in the abstract and discussion (for instance: “Down regulation of ZO-1 increased the association of YB-1 and G3BP1”. “This association refrains YB-1 from engaging in SGs assembly”).

Thank you for acknowledging that the manuscript was improved. We agree with the reviewer that the sequestration model may be an overinterpretation of the results related to the association between ZO-1 and YB-1. Therefore, we removed the proposed model of sequestration from the manuscript and now strictly focus on the disruption of the interaction. The revised conclusion for Figure 5 now states:

“This suggests that the interplay between ZO-1 and YB-1 controls the assembly of SGs and the downregulation of ZO-1 promotes the formation of SGs, and consequently the interaction between YB-1 and G3BP1 in response to stress.” (page 15, para. 2)

Also, we modified our discussion:

“It is possible that some of these RBPs also participate, in conjunction with YB-1, to the formation of SGs.” (page 23, para. 1)

“Upon stress, the association between ZO-1 and YB-1 is disrupted and ZO-1 levels are reduced, which in turn promotes SG formation in a YB-1-dependent manner, ensuring efficacious angiogenesis.” (page 23, para. 2)

Other points:

1) Authors should indicate ZONAB as YB-3 in the manuscript all the time. The use of ZONAB is misleading.

We now refer to ZONAB as YB-3 throughout the manuscript.

2) Several questions I asked previously were related to the proliferation status of the endothelial cell in the absence or presence of ZO1 and under the various conditions tested. I would have preferred to have a direct measurement based on BRDU staining for example than using VEGF pretreatment.

As mentioned above (reviewer #1), we have removed the VEGF results from the former Figure 4d and 4e (new Figure 4e, 4f) since it was redundant with Figure 2c and created some confusion. However, we kept the results using VEGF to demonstrate that cell proliferation does not drive SG formation.

As suggested by the reviewer, we now use a direct BrdU measurement to determine the proliferative status of cells containing SGs. The new Figure S3d shows that arsenite treatment decreased to similar levels the BrdU-positive cells in siCT- and si-ZO-1-transfected cells. This also indicates that cell proliferation is not the main factor driving SG formation in ZO-1 downregulated cells.

I agree with the other data presented. The data indicating that ZO-1 interacts with many RBPs is very interesting.

Thank you for this comment.

Reviewer #3:

The authors have addressed some concerns raised by the reviewers. However, the model proposed by the authors is based on correlations and remains vague.

Figure 2:

One key finding of the paper is that downregulation of ZO1 promotes SG formation under stress. It is not clear to me why VEGF (VEGF-A121?), which leads to downregulation of junctional proteins (including VE-cad and ZO1), should prevent SG formation (Fig. 2c). This clearly shows that ZO1 downregulation is not sufficient for stress-induced SG formation. It also appears that VEGF treatment is cytoprotective while preventing SG formation.

The addition of the VEGF data in the previous version (VEGF-A165, now mentioned in the methods) was meant to demonstrate that SG formation is not involved in the antiapoptotic effects of VEGF, in contrast to the protective effects induced by the downregulation of ZO-1. It is well-known that the anti-apoptotic effects of VEGF are mostly through the PI3K/Akt signaling pathway. Hence, VEGF pre-treatment attenuates the cytotoxic effects of arsenite and indirectly decreases SG formation even in ZO-1 downregulated cells. We now added new results showing that in ZO-1-downregulated ECs, VEGF pre-treatment sustained the viability of cells but significantly decreased the arsenite-induced SG formation (Figure 4c, d). We conclude: “These results suggest that the antiapoptotic effects of VEGF on ECs do not involve an increase in SG formation and differ from the protective effects of ZO-1 against stress, which involve the formation of SGs.”. Therefore, we believe that the cytoprotective effect of VEGF, which indirectly decrease SG formation, and the effects of ZO-1 downregulation, that induce SGs, are two distinct mechanisms that acts concurrently. (page 12, para. 2)

The authors argue that SG induction by ZO1 depletion is independent of cell-cell junctions (Fig. S2.). However, simultaneous ZO1/VE-cad (or β -cat) knockdown enhances SG formation (Fig. 2e). Isn't this contradictory?

Our point here is simply that when VE-cadherin or β -catenin is depleted, which disrupts cell-cell junctions in ECs, the downregulation of ZO-1 is still able to enhance SG formation in response to arsenite. This is similar to sparsely plated ECs (Figure S2j). Thus, the induction of SGs by the downregulation of ZO-1 is independent of the presence of VE-cadherin, β -catenin or cell-cell junctions.

Figure 4:

Similar to Fig. 2c, VEGF-A treatment reduces ZO1 level while inhibiting SG formation.

In 4d, siCT in leading cells show SGs level similar to siZO1, however migration is reduced in scratch wound assays. At the same time VEGF-a treatment enhances cell migration irrespective of ZO-1 kd. This finding is in conflict with the in vivo data (Figures 7 and 8) showing migration defects in the retinas of ZO1 ko, which are wild-type for VEGFa. Can the ZO1 ko phenotype be rescued by exogenous application of VEGF-A?

We apologize for the confusion. While we do not observe a difference in SG formation in arsenite-treated siCT and siZO-1 leading cells (ZO-1 levels are already low in leading cells), siZO-1 follower cells show a significant increase in SGs compared to siCT follower cells (Figure 4f). This explains why arsenite treatment decreases cell migration in siCT transfected ECs and not in siZO-1, taking in consideration that both leading and follower cells contribute to the migration measured

in a wound healing assays (Figure 4h). In addition, VEGF treatment enhances EC migration, but not irrespective of ZO-1 KD, since VEGF increases further the cell migration of ZO-1 downregulated cells compared to VEGF-stimulated siCT cells (Figure 4h).

We have addressed the apparent contradiction between the *in vivo* and *in vitro*. Figure 4h shows that ZO-1 depletion in ECs is protective against stress due to the increased formation of stress granules and restores migration in presence of arsenite to levels comparable to VEGF-stimulated ECs. Furthermore, Figure 6b adds to this by showing that this protective effect of ZO-1 downregulation on migration is dependent on YB-1-induced SG formation. In addition, our detailed analyses of the retinal vasculature clearly show that deletion of ZO-1 in ECs *in vivo* increases granule formation but also alters many of the angiogenic processes resulting in an overall inhibition of vascular growth in the retina. We cannot directly determine if migration of ZO-1^{EC-KO} ECs is increased *in vivo* but the increased disorganization and proliferation and reduced polarization of ZO-1 KO Tip-ECs surely contribute to the inhibition of vascular growth. Therefore, we believe that that the *in vivo* and *in vitro* results complement each other.

We have incorporated the following in the results: “These results suggest that by disturbing the balance of ZO-1 in ZO-1^{EC-KO} mice, the angiogenic program is altered, thereby resulting in increased YB-1-positive granules and enhanced EC proliferation in the remodeling plexus.” (page 19, para. 1)

We also added the following to the discussion: “Efficacious angiogenesis *in vivo* requires the perfect coordination of several complex mechanisms that are finely regulated, such as proliferation, migration, cell communication, and the formation and breakdown of cell junctions. ZO-1 is likely required to coordinate these mechanisms, which are modulated by VEGF and involved in the formation of a functional vasculature.” (page 21, para. 2)

ZO-1 KO phenotype cannot be rescued by exogenous application of VEGF-A. The injection of exogenous VEGF in the retinas will exacerbate damage the vascular growth and integrity leading to a pathological vessel leakage.

In the quantification of scratch wound assays (Fig. 4d and 3), it would be important to include an arsenite (-) control, to see whether SG formation and ZO1 are differentially regulated in leader and follower cells.

Thank you for this suggestion. As shown in the new Figure 4f, we do not observe SGs in leading or follower cells transfected with either siCT or siZO-1 in absence of arsenite treatment. Furthermore, in absence of arsenite, ZO-1 levels were lower in leading cells compared to follower cells (Figure 4g). Treatment with arsenite significantly decreased ZO-1 levels in follower cells whereas leading cells had already low ZO-1 levels.

Figure 5:

The authors suggest that the binding (and dissociation) of ZO1 and YB-1 may represent the key regulatory mechanism of SG formation. The evidence presented is quite correlative.

Fig. 5a shows that long-term arsenite treatment leads to changes in expression of YB-1, YB-3 and ZO1. However, it is not clear how this long-term change impacts on SG formation.

Fig. 5d shows that YB-1 is lost from cell-cell junctions under arsenite treatment. While the data is convincing, it is difficult to conceive how this can explain the proposed mechanism, since YB-1 protein levels are much higher than ZO1 levels.

We are convinced by our data showing that the dissociation of YB-1 and ZO-1 is the primary driver of SG formation. However, as mentioned above (Reviewer #2), we toned down the proposed mechanism that YB-1 can be sequestered by ZO-1 and prevent the formation of stress granules. Our study provides multiple evidence showing that the interplay between ZO-1 and YB-1 control the assembly of SGs.

The revised conclusion for Figure 5 now states:

“This suggests that the interplay between ZO-1 and YB-1 controls the assembly of SGs and the downregulation of ZO-1 promotes the formation of SGs, and consequently the interaction between YB-1 and G3BP1 in response to stress.” (page 15, para. 2)

Also, we modified our discussion:

“It is possible that some of these RBPs also participate, in conjunction with YB-1, to the formation of SGs.” (page 23, para. 1)

“Upon stress, the association between ZO-1 and YB-1 is disrupted and ZO-1 levels are reduced, which in turn promotes SG formation in a YB-1-dependent manner, ensuring efficacious angiogenesis.” (page 23, para. 2)

In Fig. 5f overexpression of YB-1 enhances while ZO1 overexpression reduces SG formation. Overexpression of both construct leads to an intermediate phenotype. It not clear, however this compensatory relationship is due to direct protein-protein interaction. By determining the interactive protein domain of ZO1 (SH3 as suggested by the authors?), the authors can test this hypothesis more directly, by using different mutant isoforms of ZO1 in rescue similar to those performed in Fig. 5f.

We performed pulldown experiments using recombinant GST-ZO-1 fragments (residues 1-510 and residues 511-1100) to document the protein-protein interaction between ZO-1 and YB-1. The new Figure 5d shows that the ZO-1 fragment 511-1100, which includes the SH3 and GUK domains, interacts with YB-1. (page 14, para. 2)

Based on our experience with ZO-1 constructs, any modifications to the ZO-1 structure disturbs greatly its function such as interaction with cytoskeleton, cell localization, etc. Thus, rescue experiments using different mutant isoforms of ZO-1 will be impossible to interpret. Of interest, ZO-2 downregulation does not induce SG formation.

Reviewer #4 (Remarks to the Author):

Authors have addressed some of my concerns and comments on the original manuscript with support of additional experiments. However, the major suggested works such as the experiments related to interactome analysis under arsenite treatment (R4#5), assessing domain within ZO-1 (R4#10), post stress recovery (R4#16) and LLPS (R4#18), which are crucial to bring the manuscript to Nat Commun standard level have not been performed. Authors' reasoning on not performing such experiments such as out of scope and/or the complexity is not acceptable if this manuscript is considered for publication in Nat Commun.

We respectfully disagree that defining the ZO-1 interactome under arsenite treatment (R4#5) or broadly investigating if LLPS is involved in driving SG formation (R4#18) are needed to attain Nat Commun standard level. Nonetheless, in this revised version, we have assessed domains within ZO-1 (R4#10) and analyzed post stress recovery (R4#16) as requested.

Assessing domain within ZO-1 (R4#10) – To assess the domains within ZO-1 that are involved in the interaction with YB-1, we performed pulldown experiments using recombinant GST-ZO-1 fragments (residues 1-510 and residues 511-1100) and now show that the ZO-1 fragment comprising residues 511-1100, which includes the SH3 and GUK domains of ZO-1 interacts with YB-1 (Figure 5d, Supplemental Figure S4c). These new data reinforce our initial hypothesis that the SH3 of ZO-1 is involved in the association with YB-1, which is similar to the association between ZO-1 and YB-3 (Discussion, page 22, para. 2).

Post stress recovery (R4#16) – As requested, we examined the formation of SGs and the levels of ZO-1 following a recovery period after 1 h of arsenite treatment. Page 11, 1st para.: “Next, we examined the presence of SGs and the levels of ZO-1 following a recovery period after 1 h of exposure to arsenite. Subsequent to arsenite removal, ZO-1 levels remained decreased for 1 h, and the number of ECs with SGs remained elevated (Figure 3g, h). After 3 h of recovery, ZO-1 levels began to return to normal, coinciding with an absence of cells positive for SGs (Figure 3g, h).”

Interactome analysis under arsenite treatment (R4#5) – In the initial phases of this study, prior to the identification of YB-1, we had attempted to examine the effect of arsenite treatment of ECs on the ZO-1 interactome by IP-MS. The reasoning for not using these results was not simply due to the complexity of the experiments. As mentioned in our previous response to the reviewer, arsenite treatment reduced the levels of the bait itself (ZO-1) by approximately by 5 to 14-folds, depending on the experiment. In addition, we observed a large variation between replicates in the effects of arsenite on the ZO-1 interactome as 50 to 98% of the ZO-1 interacting proteins were affected by the arsenite treatment. We tried different concentrations of arsenite and duration of treatment but the effects of arsenite treatment remained variable and sometimes disrupted all the interactions with ZO-1 rendering the data impossible to interpret. Nevertheless, we were able to identify YB-1 as the critical ZO-1 interactor for the regulation of SG formation. The addition of the ZO-1 interactome under arsenite treatment would not add much useful information.

LLPS (R4#18) – As we mentioned in our previous response to this point, we don't believe that ZO-1 is involved in the liquid-liquid phase separation process leading to SG formation. Since it is the ZO-1 deficiency that enhances SG formation in response to stress, it is difficult to conceive how LLPS would directly be activated by ZO-1. As for this previous comment by the reviewer: “Any

protein modifications on stress granule proteins in ZO-1 deficient ECs that could serve to either disrupt critical multivalent interactions or provide new contact surfaces that drive LLPS?"; it implies to broadly study the general LLPS mechanisms involved in the formation of SGs, which would not provide information on the role of ZO-1 in this process. We had included some text in the discussion on LLPS and SG formation in the previous version of the manuscript (now on page 23, para. 1).

REVIEWERS' COMMENTS

Reviewer #1 (Remarks to the Author):

The authors have improved their study in this second round of revision and have responded in full to my concerns. I have no further comments.

Reviewer #2 (Remarks to the Author):

In this new version of the manuscript, as I suggested, the authors withdrawn their claim about the sequestration of YB-1 by ZO-1, since ZO1 is much less expressed than YB-1. I appreciate it.

However, one of their results is that the interaction between ZO1 and YB-1 downregulates the interaction with G3BP1 (hard to imagine if there is plenty of free YB-1). In the discussion the authors mentioned that YB-1 increases the expression of G3BP1 level but no longer the interaction between G3BP1 and YB-1, while this results is still presented in the manuscript.

They can fix this point and be clearer in their conclusion.

The author performed BRDU staining analysis, as I suggested. They found that the percentage of BrdU positive cell increases significantly after ZO1 depletion. However, since 30 min arsenite treatment decreases similarity cell proliferation (% BrDu positive cells) in Si control and siZO1 cells, they considered that cell proliferation is not the main determinant.

I am very surprised about this conclusion. What matters most is the change in proliferation rate due to ZO1 silencing before the short 30 min arsenite treatment. Their results clearly prove the contrary. Cell proliferation could be a critical parameter, notably by promoting SG formation. Their conclusion is erroneous here.

Reviewer #4 (Remarks to the Author):

Authors have addressed almost all the concerns and comments on the revised manuscript with support of some additional experiments that helped to improve the manuscript.

Reviewer #1 (Remarks to the Author)

The authors have improved their study in this second round of revision and have responded in full to my concerns. I have no further comments.

We thank the reviewer for his insightful comments and suggestions, which have improved our manuscript.

Reviewer #2:

In this new version of the manuscript, as I suggested, the authors withdrawn their claim about the sequestration of YB-1 by ZO-1, since ZO1 is much less expressed than YB-1. I appreciate it. However, one of their results is that the interaction between ZO1 and YB-1 downregulates the interaction with G3BP1 (hard to imagine if there is plenty of free YB-1). In the discussion the authors mentioned that YB-1 increases the expression of G3BP1 level but no longer the interaction between G3BP1 and YB-1, while this result is still presented in the manuscript. They can fix this point and be clearer in their conclusion.

The reviewer refers to Figure 5h showing that the interaction between YB-1 and G3BP1 is increased in ZO-1-downregulated cells. As mentioned previously, here we use the interaction between YB-1 and G3BP1 solely as a biochemical marker of stress granule formation in response to arsenite treatment, not to argue in favor of the YB-1 sequestration model. Increased interaction between two known constituents of SGs, YB-1 and G3BP1, reflects increased SG formation. However, we cannot fully explain why the interaction between YB-1 and G3BP1 is increased in ZO-1 downregulated cells even in absence of arsenite and of SGs (Fig. 5h). We opted to keep this figure in the manuscript because we considered these results of interest, but we made our conclusions clearer.

We modified our description of Figure 5h in the results section (p.15-16): “YB-1 and G3BP1 are both markers of SGs and their interaction generally reflects SG formation^{40, 41, 43}. Thus, we examined the effect of arsenite treatment in ZO-1-downregulated ECs on the YB-1 and G3BP1 interaction. We show that arsenite treatment of ECs did increase the coimmunoprecipitation of YB-1 and G3BP1 and, interestingly, that the interaction between YB-1 and G3BP1 was augmented in untreated and ZO-1-downregulated ECs (Fig. 5h). However, this interaction was not further increased by arsenite treatment (Fig. 5h). Overall, these results suggest that YB-1 is necessary for the induction of SG formation in ECs where ZO-1 is downregulated and that the interplay between ZO-1 and YB-1 regulates the assembly of SGs in ECs.”

We also added the following in the discussion on p. 23: “Our results also suggest that disruption of the ZO-1 and YB-1 association induces signals in ECs that promote the interaction between YB-1 and other constituents of SGs, namely G3BP1, which may contribute to SG formation (Fig. 5h).”

The author performed BRDU staining analysis, as I suggested. They found that the percentage of BrdU positive cell increases significantly after ZO1 depletion. However, since 30 min arsenite treatment decreases similarity cell proliferation (% BrDu positive cells) in Si control and siZO1 cells, they considered that cell proliferation is not the main determinant.

I am very surprised about this conclusion. What matters most is the change in proliferation rate due to ZO1 silencing before the short 30 min arsenite treatment. Their results clearly prove the contrary. Cell proliferation could be a critical parameter, notably by promoting SG formation. Their conclusion is erroneous here.

We agree that the results obtained from the BrdU assay, as suggested by the reviewer, do not conclusively disprove (or prove) the role of cell proliferation in promoting SG formation (Supplemental Fig. S3d). Hence, we now limited our interpretation of Fig. S3d to the effects of arsenite treatment on EC proliferation in ZO-1-downregulated cells and now mention the following on p. 12:

“To examine the effects of arsenite treatment on EC proliferation induced by ZO-1 downregulation, we counted the number of proliferating cells by measuring bromodeoxyuridine (BrdU) incorporation in control and ZO-1-downregulated ECs in the absence or presence of arsenite. While the downregulation of ZO-1 increased cell proliferation in the control (untreated ECs), arsenite treatment decreased the number of BrdU-positive cells to similar levels to those in siCT- and siZO-1-transfected cells (Supplemental Fig. S3d). These results indicate that arsenite-induced stress inhibits proliferation equally in ZO-1-downregulated and control ECs.”

Additionally, we have now included in the discussion the reference mentioned by the Reviewer in the first round of evaluation that proposes, although based on indirect evidence, that an increase in the rate of cell proliferation contributes to SG assembly (Desforges et al., MBoC, 2013; Ref. 60). However, when considering the fact that VEGF is a strong inducer of EC proliferation that does not induce SG formation in ECs (Fig. 4d) we have limited our conclusion to “However, we found that VEGF stimulation did not induce SG formation, indicating that the anti-apoptotic signaling mechanisms activated by VEGF are different from the ones activated by ZO-1 signaling to SGs in response to stress. It is well-established that the phosphoinositide 3-kinase/protein kinase B (PI3K/Akt) pathway is primarily responsible for the pro-survival and the anti-apoptotic effects of VEGF on ECs^{58, 59}. Moreover, both VEGF and ZO-1 downregulation increase EC proliferation. Interestingly, it was proposed that an increase in the rate of cell proliferation contributes to SG assembly⁶⁰. Hence, it is tempting to suggest that, in addition to differing for the regulation of SG formation, the signaling mechanisms employed by ZO-1 and VEGF may also differ in regulating EC proliferation.” We believe that this conclusion leaves sufficient room for future investigations on the mechanisms involved in driving SG assembly.

Reviewer #4 (Remarks to the Author)

Authors have addressed almost all the concerns and comments on the revised manuscript with support of some additional experiments that helped to improve the manuscript.

Thank you for reviewing our manuscript.